# Oxygen-evolving photosystem II structures during S₁–S₂–S₃ transitions

Hongjie Li[1,12], Yoshiki Nakajima[1,12], Eriko Nango[2,3], Shigeki Owada[4], Daichi Yamada[5], Kana Hashimoto[1], Fangjia Luo[4], Rie Tanaka[3,6], Fusamichi Akita[1], Koji Kato[1], Jungmin Kang[3], Yasunori Saitoh[1], Shunpei Kishi[1], Huaxin Yu[1], Naoki Matsubara[1], Hajime Fujii[1], Michihiro Sugahara[4], Mamoru Suzuki[7], Tetsuya Masuda[8], Tetsunari Kimura[9], Tran Nguyen Thao[1], Shinichiro Yonekura[1], Long-Jiang Yu[1,10], Takehiko Tosha[3], Kensuke Tono[4], Yasumasa Joti[4], Takaki Hatsui[4], Makina Yabashi[4], Minoru Kubo[5], So Iwata[3,6], Hiroshi Isobe[1], Kizashi Yamaguchi[11], Michihiro Suga[1✉] & Jian-Ren Shen[1✉]

Photosystem II (PSII) catalyses the oxidation of water through a four-step cycle of $S_i$ states ($i = 0-4$) at the $Mn_4CaO_5$ cluster[1-3], during which an extra oxygen (O6) is incorporated at the $S_3$ state to form a possible dioxygen[4-7]. Structural changes of the metal cluster and its environment during the S-state transitions have been studied on the microsecond timescale. Here we use pump-probe serial femtosecond crystallography to reveal the structural dynamics of PSII from nanoseconds to milliseconds after illumination with one flash (1F) or two flashes (2F). $Y_Z$, a tyrosine residue that connects the reaction centre P680 and the $Mn_4CaO_5$ cluster, showed structural changes on a nanosecond timescale, as did its surrounding amino acid residues and water molecules, reflecting the fast transfer of electrons and protons after flash illumination. Notably, one water molecule emerged in the vicinity of Glu189 of the D1 subunit of PSII (D1-E189), and was bound to the $Ca^{2+}$ ion on a sub-microsecond timescale after 2F illumination. This water molecule disappeared later with the concomitant increase of O6, suggesting that it is the origin of O6. We also observed concerted movements of water molecules in the O1, O4 and Cl-1 channels and their surrounding amino acid residues to complete the sequence of electron transfer, proton release and substrate water delivery. These results provide crucial insights into the structural dynamics of PSII during S-state transitions as well as O–O bond formation.

Photosystem II (PSII) produces dioxygen by extracting electrons and protons from water, which takes place at the oxygen-evolving complex (OEC), an oxo-bridged $Mn_4CaO_5$ cluster with a shape that resembles a distorted chair[2,3,8]. The Mn atoms in the OEC accumulate oxidative power through a four-step cycle of $S_i$ states ($i = 0-4$) that is initiated by the light-driven excitation of P680, a reaction centre that is a complex of chlorophyll $a$ molecules[1] (Extended Data Fig. 1a)[1]. This is followed by a rapid charge separation that produces a pair of positive and negative charges on P680$^{•+}$/pheophytin$^{•-}$ (Pheo$^{•-}$) on a picosecond timescale[9,10]. The electron is transferred from Pheo$^{•-}$ to the primary and secondary plastoquinones $Q_A$ and $Q_B$ (Extended Data Fig. 1b). The P680$^{•+}$ is then reduced by a tyrosine residue (D1-Y161; $Y_Z$) located between P680 and the OEC, which is re-reduced by the OEC, pushing the OEC to a higher $S_i$ state[11]. In conjunction with the oxidation of the OEC, protons are released in a 1:0:1:2 stoichiometry for the $S_0-S_1$, $S_1-S_2$, $S_2-S_3$ and $S_3-(S_4)-S_0$ transitions[12-14], and two water molecules are split to produce a

dioxygen in the $S_3-(S_4)-S_0$ transition, after which the OEC returns to its most reduced $S_0$ state.

The water-splitting reaction requires a constant replenishment of water from the lumen, as well as the prompt elimination of the generated protons into the lumen. There are extensive hydrogen-bonding networks connecting the OEC with the lumen, and among these, the O1, O4 and Cl-1 channels are proposed to have essential roles in the water-splitting reaction[6,15-18] (Extended Data Fig. 1c). (Note that the first 56 water molecules are named following a previous report[18], and other water molecules are newly numbered; see Supplementary Table 1 for corresponding numbers in other studies). The O1 channel is a wide channel starting from a five-water cluster (W10, W20, W21, W22 and W23) that is located near O1 of the OEC (OEC-O1). This channel travels across a narrow area and ends at a giant cavity in which two glycerol molecules are found in the crystal structure[2,3] (Extended Data Fig. 1c). The wide O1 channel might give a high mobility of water within it, and is

[1]Research Institute for Interdisciplinary Science, Graduate School of Natural Science and Technology, Okayama University, Okayama, Japan. [2]Institute of Multidisciplinary Research for Advanced Materials, Tohoku University, Sendai, Japan. [3]RIKEN SPring-8 Center, Sayo, Japan. [4]Japan Synchrotron Radiation Research Institute, Sayo, Japan. [5]Department of Picobiology, Graduate School of Life Science, University of Hyogo, Kobe, Japan. [6]Department of Cell Biology, Graduate School of Medicine, Kyoto University, Kyoto, Japan. [7]Institute for Protein Research, Osaka University, Osaka, Japan. [8]Division of Food and Nutrition, Faculty of Agriculture, Ryukoku University, Otsu, Japan. [9]Department of Chemistry, Graduate School of Science, Kobe University, Kobe, Japan. [10]Key Laboratory of Photobiology, Institute of Botany, Chinese Academy of Sciences, Beijing, China. [11]Center for Quantum Information and Quantum Biology, Osaka University, Osaka, Japan. [12]These authors contributed equally: Hongjie Li, Yoshiki Nakajima. ✉e-mail: michisuga@okayama-u.ac.jp; shen@okayama-u.ac.jp

therefore considered as a potential water inlet pathway[6,17]. By contrast, the O4 channel is a shorter channel that starts at OEC-O4 and ends at a four- or five-water cluster (Extended Data Fig. 1c). The Cl-1 channel refers to a hydrogen-bonding network mediated by Cl-1, which spans from W1 to W4, continues through D1-D61 and further extends to an ionic gate comprising D1-E65, D1-R334 and D2-E312 (Extended Data Fig. 1c). Cl$^-$ ions are essential for the progression of PSII beyond the $S_2$ state[19–21], and the Cl-1 channel is thought to serve as a proton-release pathway in the $S_2$–$S_3$ transition[17,22,23].

Pump-probe time-resolved femtosecond crystallography (TR-SFX) has provided a lot of information about the intermediate S-state structures of PSII (refs. 4–7,17,24,25). However, time-resolved structures at shorter timescales during the $S_1$–$S_2$ and $S_2$–$S_3$ transitions are lacking, and thus the sequence of OEC oxidation, proton release, electron transfer and water delivery before O6 incorporation is unclear. Here we investigate the structural dynamics during the $S_1$–$S_2$ and $S_2$–$S_3$ transitions using the pump-probe TR-SFX method at delay times ($\Delta t$) of 20 ns to 5 ms (Extended Data Fig. 1d). We identify structural changes associated with electron transfer, proton release and water delivery at various regions, including $Q_A$–$Q_B$, $Y_Z$, the OEC and the O1, O4 and Cl-1 channels. Notably, we observe the presence of a water molecule close to Ca at initial stages of the $S_2$–$S_3$ transition. This water molecule subsequently disappears with the concomitant increase of the O6 electron density, suggesting that it is the origin of O6. Our findings provide spatial and time-resolved snapshots of the $S_1$–$S_2$–$S_3$ state transitions, which are important for the mechanism of O–O bond formation.

## Data quality

We obtained 14 datasets at resolutions ranging from 2.15 to 2.30 Å, with redundancy values higher than 100 even at the highest-resolution shells, after 1F or 2F (Extended Data Table 1). For all datasets, we calculated the $F_{obs}$ (1F($\Delta t1$)) – $F_{obs}$(Dark) and $F_{obs}$ (2F($\Delta t2$)) – $F_{obs}$(1F) isomorphous-difference density maps at 2.3-Å resolution. The $R_{iso}$ values between the intermediate and ground states ranged from 6% to 11% (Supplementary Table 2)—sufficiently low to allow the confident detection of subtle structural changes during $S_i$-state transitions. We observed substantial difference densities in the $Q_A$–$Q_B$ and OEC regions and in the proton and water channels at the electron donor side; their intensities are listed in Supplementary Table 3.

## Structural changes in the $Q_A$–Fe–$Q_B$ area

$Q_A$ and $Q_B$ are linked to the non-haem iron through hydrogen bonds with D2-H214 and D1-H215, forming an iron–quinone complex. The carbonyl oxygens of the $Q_A$ and $Q_B$ heads are also hydrogen-bonded to D2-F261 and to D1-F265/D1-S264, respectively (Fig. 1).

Large difference densities appear on the $Q_A$ side at $\Delta t1 = 20$ ns and $\Delta t1 = 200$ ns, become weak at $\Delta t1 = 1$ μs to $\Delta t1 = 200$ μs and vanish at $\Delta t1 = 5$ ms (Fig. 1a and Supplementary Video 1). These changes correspond to the formation of $Q_A^-$, the oxidation of $Q_A^-$ to $Q_A$ and the completion of $Q_A^-$ oxidation, respectively. The formation of $Q_A^-$ causes the counterclockwise rotation of its head group, concomitant with similar rotations or movements of D2-F261, D2-W253 and D2-H214, which surround $Q_A$ (Fig. 1a and Supplementary Video 1). The formation of $Q_A^-$ also induces a shift of the non-haem iron by about 0.2 Å towards $Q_A$. The pair of positive and negative difference densities around the non-haem iron is strongest at $\Delta t1 = 20$ ns and $\Delta t1 = 200$ ns, which is much faster than the time needed for the reduction of Fe$^{3+}$ by $Q_A^-$ (7 μs in refs. 26,27), indicating that the movement of the non-haem iron is caused not by its reduction but rather by the attraction of electropositive Fe$^{3+}$ to $Q_A^-$. The diminishing difference densities around $Q_A$ and the non-haem iron at $\Delta t1 = 1$ μs to $\Delta t1 = 200$ μs suggest that the attraction between the non-haem iron and $Q_A^-$ is decreased and that the electron on $Q_A^-$ is transferred to the non-haem iron (Fig. 1a and Supplementary Video 1).

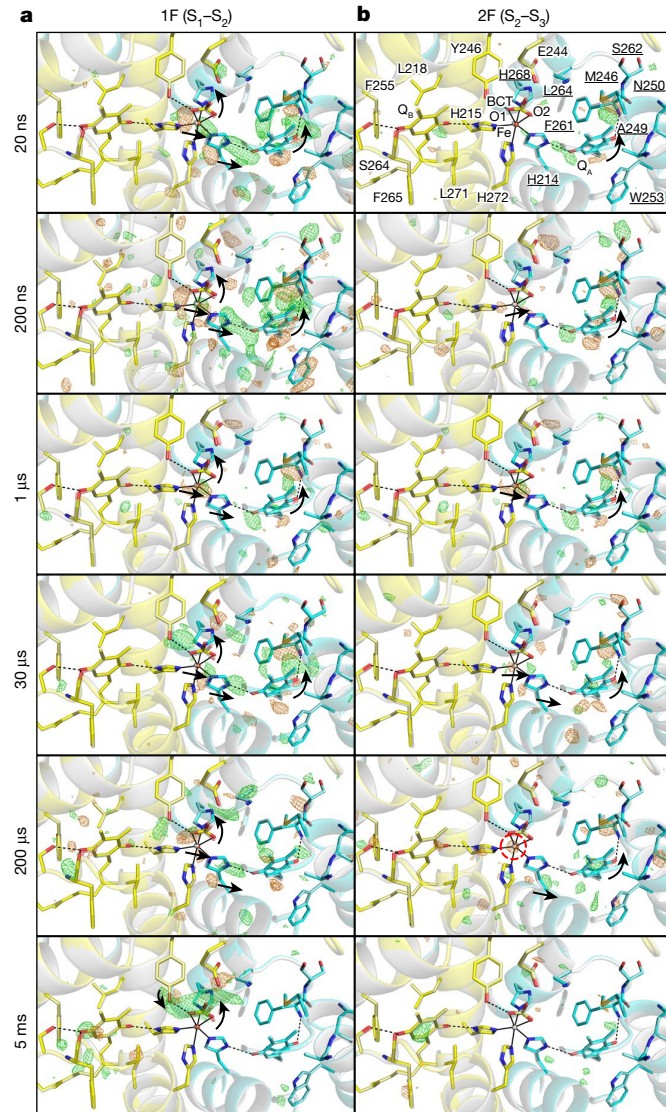

**Fig. 1 | Structural dynamics in the $Q_A$–$Q_B$ area during $S_1$–$S_2$–$S_3$ transitions. a,b**, Structures of PSII in the $Q_A$–$Q_B$ area are superposed with $F_{obs}$(1F) – $F_{obs}$(Dark) (**a**) and $F_{obs}$(2F) – $F_{obs}$(1F) (**b**) difference density maps contoured at +3.5$\sigma$ (green) and −3.5$\sigma$ (orange) from 20 ns to 5 ms. Ground-state models (dark in **a** and 1F in **b**) are depicted in grey, and the D1 and D2 proteins in the intermediate structures are shown in yellow and cyan, respectively. Residues of D1 and D2 are depicted without and with underlines, respectively. Hydrogen bonds are shown by black dotted lines. Black solid lines link the cofactors in PSII and their ligands. Black arrows indicate structural changes based on the refined models. The ordered and disordered atoms (non-haem iron in this figure and water molecules in the other figures) in the intermediate structures are encircled by cyan- and red-dotted lines, respectively. These nomenclature, hydrogen bonds, ligands of cofactors and black arrows are used in the other figures, unless otherwise stated.

By $\Delta t1 = 5$ ms, these difference densities disappear entirely, indicating the completion of the electron transfer together with the restoration of $Q_A$ and the non-haem iron.

The distances from two carbonyl oxygens of bicarbonate (BCT; BCT-O1 and BCT-O2) to the non-haem iron increase from 2.16 Å and 2.28 Å in the dark state to 2.43 Å and 2.44 Å after 5 ms of 1F (Fig. 1a, Extended Data Table 2 and Supplementary Video 1). These increases most likely reflect changes in the binding environment of BCT owing to the reduction of the non-haem iron. At $\Delta t1 = 5$ ms, a large positive difference density appears between BCT and D1-Y246 (Fig. 1a and Supplementary Video 1), consistent with our previous discovery[25], and

the distance between D1-Y246 and BCT-O1 decreases from 3.21 Å to 2.82 Å. At $\Delta t1 = 200$ μs to $\Delta t1 = 5$ ms, the $Q_B$ head shifts slightly, which might be a result of the movement of BCT or the partial reduction of $Q_B$ by $Q_A^-$ (ref. 28).

From $\Delta t2 = 20$ ns to $\Delta t2 = 30$ μs, the $Q_A$ head rotated in a counterclockwise direction, and this was accompanied by a movement of the non-haem iron towards $Q_A$—structural changes similar to those observed after 1F. However, the difference densities associated with these structural changes were much weaker after 2F (Fig. 1 and Supplementary Video 2). The non-haem iron remains $Fe^{2+}$ at 5 ms after photoreduction by 1F, because its re-oxidation by ferricyanide takes 20 s (ref. 27) (Extended Data Fig. 1d). For this reason, the electron of $Q_A^-$ does not travel to the non-haem iron, but rather travels directly to $Q_B$ after 2F, resulting in the absence of difference density on BCT and the appearance of positive difference density on the $Q_B$ head at $\Delta t2 = 5$ ms (ref. 4) (Fig. 1b and Supplementary Video 2). The non-haem iron becomes disordered at $\Delta t2 = 200$ μs but ordered by $\Delta t2 = 5$ ms, which is presumably related to electron transfer from $Q_A^-$ to $Q_B$ during this time.

## Structural changes around $Y_Z$

D1-V157, D1-F186, D1-I192 and D1-I290 lie between $Y_Z$ and $P_{D1}$ (P680 at the D1 side) (Extended Data Figs. 1b and 2), and $Y_Z$ is connected to the O1 channel through W4 and D1-Q165, to the Cl-1 channel through W7 and to the OEC through W3, W4 and W7 (Extended Data Fig. 1c). $Y_Z$ forms a short (low-barrier) hydrogen bond with D1-H190 (2.44 Å in the Protein Data Bank (PDB) under accession code 3WU2; ref. 2), through which the phenolic proton of $Y_Z^{\cdot+}$ migrates to D1-H190, forming $Y_Z^{\cdot}$–D1-H190$^+$ during the $S_i$-state transitions[11,29–31]. At $\Delta t1 = 20$ ns, two negative difference densities first appear adjacent to D1-Q165 and $Y_Z$, and at $\Delta t1 = 200$ ns, pairs of positive and negative difference densities appear over D1-Q165, $Y_Z$ and D1-F186, indicating their correlated movements towards P680 (Fig. 2a and Extended Data Fig. 2a,b). These movements might be in preparation for the subsequent electron transfer from $Y_Z$ to P680. Simultaneously with these movements, a positive difference density appears on the Mg atom of $P_{D1}$ (Fig. 2a and Supplementary Table 3), which might reflect the re-reduction of P680$^+$ by $Y_Z$.

D1-H190 moves away from $Y_Z$ at $\Delta t1 = 200$ ns, which, together with the movement of $Y_Z$ toward P680, causes the elongation of the hydrogen bond between $Y_Z$ and D1-H190 from 2.51 Å to 2.80 Å at $\Delta t1 = 200$ ns (Fig. 2a, Extended Data Fig. 3 and Supplementary Video 3). These changes suggest that $Y_Z$ is first oxidized by P680$^{\cdot+}$ and subsequently deprotonated, forming a $Y_Z^{\cdot}$/D1-H190$^+$ species, with the time constant of $Y_Z$ oxidation consistent with that reported for P680$^{\cdot+}$ reduction in the $S_1$–$S_2$ transition[31,32]. At $\Delta t1 = 1$ μs and $\Delta t1 = 30$ μs, difference densities on D1-Q165, $Y_Z$ and D1-H190 decrease, indicating that they have moved to their original locations. In addition, a strong negative difference density appears on W7, suggesting that W7 is disordered during this period (Fig. 2a and Supplementary Video 3). By $\Delta t1 = 200$ μs, all difference densities vanish at the $Y_Z$ area, indicating the restoration of all residues and water, and the $Y_Z$–D1-H190 distance returns to 2.53 Å (Fig. 2a, Extended Data Fig. 3, Extended Data Table 2 and Supplementary Video 3). The trajectories of the $Y_Z$ area at $\Delta t1 = 1$ μs and $\Delta t1 = 30$ μs correspond to the re-reduction and re-protonation of $Y_Z^{\cdot}$ to $Y_Z$, which completes by $\Delta t1 = 200$ μs, consistent with the 55–85-μs half-life of $Y_Z^{\cdot+}$ re-reduction by the OEC in the $S_1$–$S_2$ transition[33,34].

After 2F, difference densities start to appear only after $\Delta t2 = 200$ ns (Fig. 2b and Supplementary Movie 4). These lagged difference densities likely correspond to the slower and biphasic 50-ns and 280-ns components of the P680$^+$ decay in the $S_2$–$S_3$ transition[31,32]. This delay might arise from the reduced rate of electron transfer to P680$^{\cdot+}$ owing to the accumulation of a positive charge on the OEC. Difference densities on $Y_Z$ and D1-Q165 increase at $\Delta t2 = 1$ μs, and the $Y_Z$–D1-H190 distance increases slightly from $\Delta t2 = 0$ to $\Delta t2 = 1$ μs (Extended Data Fig. 3, Extended Data Table 2 and Supplementary Video 4). These

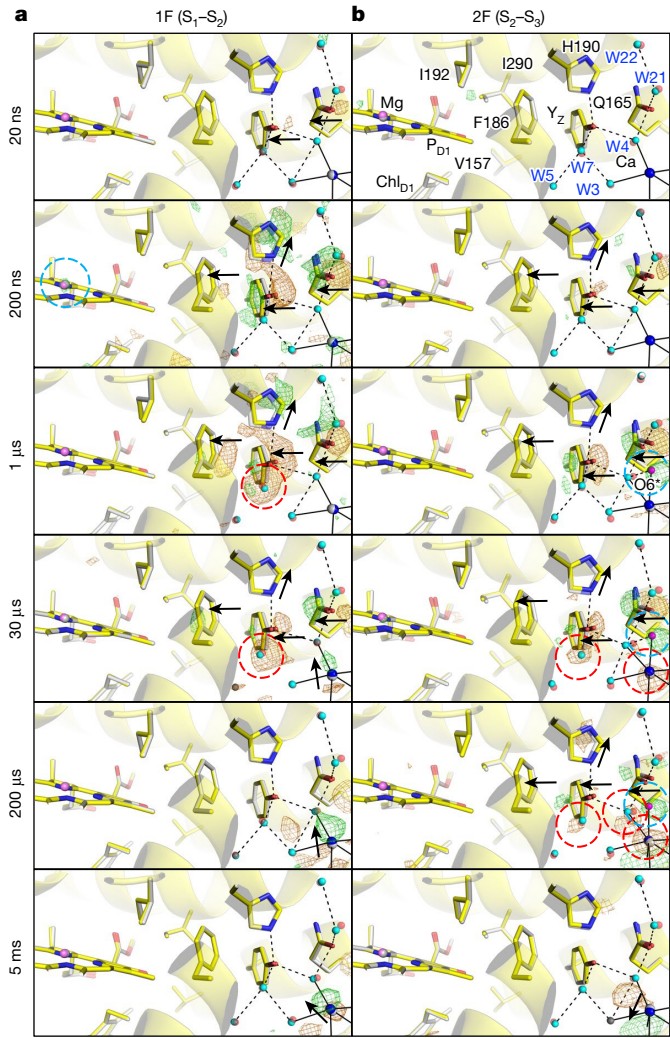

**Fig. 2 | Structural dynamics in the $Y_Z$ area during $S_1$–$S_2$–$S_3$ transitions.** **a**,**b**, Structures of PSII in the $Y_Z$ area are superposed with $F_{obs}$(1F) − $F_{obs}$(Dark) (**a**) and $F_{obs}$(2F) − $F_{obs}$(1F) (**b**) difference density maps contoured at +4.0σ (green) and −4.0σ (orange), with delay times from 20 ns to 5 ms. Water molecules at ground and intermediate states are depicted by red and cyan spheres, respectively. The Mg atom in $P_{D1}$ is shown as a grey sphere for the ground structure and a violet sphere for the intermediate structure. O6* and O6 (in the other figures) are depicted in magenta, and the Ca ion of the OEC is in blue. The same colour scheme is used in other figures, unless otherwise stated.

findings suggest the oxidation of $Y_Z$ and a potential proton transfer from $Y_Z$ to D1-H190 at $\Delta t2 = 1$ μs, which is similar to that observed at $\Delta t1 = 20$ ns–200 ns. At $\Delta t2 = 30$ μs, difference densities on $Y_Z$ and D1-Q165 decrease, indicating the re-reduction of $Y_Z^{\cdot}$ by the OEC. The difference density on $Y_Z$ becomes even weaker at $\Delta t2 = 200$ μs, but is still present (Fig. 2b and Supplementary Video 4), indicating that the reduction of $Y_Z$ is not yet complete, which is compatible with the half-life of 140–90 μs of $Y_Z$ reduction[33,34]. $Y_Z$ reduction is completed by 5 ms, and difference densities disappear at the $Y_Z$ area and all residues and water molecules are restored (Fig. 2b and Supplementary Video 4). The time-resolved redox states of $Y_Z$ after 1F and 2F that are described above are summarized in Extended Data Fig. 2c.

## Oxidation of the OEC during the $S_1$–$S_2$ transition

Notable positive difference densities first appear on Mn4 and subsequently cover all four Mn and one Ca ions at $\Delta t1 = 20$ ns–200 ns before OEC oxidation (Fig. 3a and Supplementary Video 3). Nevertheless,

metal–metal distances remain largely unchanged (Extended Data Fig. 3 and Extended Data Table 2), suggesting a possible charge rearrangement on the OEC triggered by the electrostatic effect of the oxidized $Y_Z^{•+}/Y_Z$. At $\Delta t1 = 1\,\mu s$, difference densities on Mn1–Mn3 and Ca vanish, whereas that on Mn4 continues (Fig. 3a and Supplementary Video 3). At $\Delta t1 = 30\,\mu s$, paired negative and positive difference densities appear on two sides of Ca, indicating that Mn4 and Ca move outwards from the OEC, causing an increase in the Mn4–Ca distance from 3.83 Å at $\Delta t1 = 200$ ns to 3.96 Å at $\Delta t1 = 30\,\mu s$. By $\Delta t1 = 200\,\mu s$, the difference densities in the $Y_Z$ area vanish completely, whereas those surrounding Mn4 and Ca increase, and the Mn4–Ca distance further extends to 4.10 Å (Fig. 3a, Extended Data Fig. 3, Extended Data Table 2 and Supplementary Video 3). The results suggest that Mn4(III) of the OEC donates one electron to $Y_Z^•$ at $\Delta t1 = 1\,\mu s$ to $\Delta t1 = 200\,\mu s$. At the completion of Mn4 oxidation by $\Delta t1 = 200\,\mu s$, a negative difference density emerges on O5, suggesting its instability, which is subsequently stabilized at $\Delta t1 = 5$ ms. In addition, at $\Delta t1 = 5$ ms, a positive difference density appears near Mn1 but outside of the OEC, suggesting the movement of Mn1 away from the OEC. These structural changes might stabilize the positive charge on the OEC.

In correlation with the outward movement of Ca from $\Delta t1 = 30\,\mu s$ to $\Delta t1 = 5$ ms, one of the carboxyl oxygens of D1-E189 located close to Ca shifts slightly away from Ca. Because the movement of Ca is larger than that of D1-E189, the Ca–D1-E189 distance decreases from 3.02 Å ($\Delta t1 = 1\,\mu s$) to 2.86 Å ($\Delta t1 = 5$ ms) (Fig. 3a, Extended Data Fig. 3, Extended Data Table 2 and Supplementary Video 3). In addition, W10, which is located in the proximity of D1-E189, becomes disordered in the same time range, and this correlates with the motion of Ca and D1-E189.

## Insertion of O6 in the S₂–S₃ transition

No difference density appears on the OEC at $\Delta t2 \leq 200$ ns, suggesting that no structural changes to the OEC occur in this time range (Fig. 3b and Supplementary Video 4). One notable positive difference density—designated as O6*—emerges approximately 2.2 Å away from Ca during $\Delta t2 = 1\,\mu s$ to $\Delta t2 = 200\,\mu s$, and disappears by $\Delta t2 = 5$ ms, with the concomitant increase of the O6 density from $\Delta t2 = 200\,\mu s$ to $\Delta t2 = 5$ ms (Fig. 3b, Extended Data Fig. 4a, Supplementary Table 3 and Supplementary Video 4). These observations suggest that O6* is the origin of O6, and that O6* binds to Ca at $\Delta t2 = 1\,\mu s$ to $\Delta t2 = 30\,\mu s$, translocates to O6 at $\Delta t2 = 200\,\mu s$ and completes its translocation by $\Delta t2 = 5$ ms.

At $\Delta t2 \leq 1\,\mu s$, there are no difference densities among the neighbouring water molecules of O6*, indicating that O6* does not originate from any stable water molecules nearby. Instead, it is likely to be derived from an aqueous water—specifically, W10—located 2.5 Å away from O6* in the $S_1$ state, and becomes disordered at $\Delta t1 = 30\,\mu s$–5 ms (Fig. 3 and Supplementary Video 3). The 2.2-Å distance between O6* and Ca indicates that O6* could be a hydroxide ion (OH⁻) rather than a water molecule, because water molecules W3 and W4 bind to Ca at distances ranging from 2.4 to 2.6 Å. Indeed, theoretical calculations indicate that an OH⁻ ion positioned close to O6* in the $S_2$ state (Extended Data Fig. 5a,b) exhibits low energy and high stability, whereas the placement of a water molecule is not feasible. The deprotonation of water most likely occurs at $\Delta t2 = 200$ ns–1 μs, during which time the simultaneous existence of $Y_Z^{•+}/Y_Z^•$ and OEC⁺ might collectively promote the deprotonation of the O6* precursor. Consequently, the resulting OH⁻ ion binds to Ca, neutralizing the positively charged OEC⁺. The achievement of a neutral OEC is crucial for the donation of an electron from the OEC to $Y_Z^•$, because it is energetically unfavourable for the OEC⁺ to continuously lose one more electron to $Y_Z$ and transform into OEC²⁺. The subsequent transfer of one electron from the OEC to $Y_Z^•$, occurring at $\Delta t2 = 30\,\mu s$ and $\Delta t2 = 200\,\mu s$, leads to a decrease in the difference densities at the $Y_Z$ region and a simultaneous increase in the difference densities on the OEC (Fig. 3b, Supplementary Table 3 and Supplementary Video 4). The presence of paired positive and negative difference densities on both sides of Mn1 and Mn4 indicates outward movements of Mn1 and

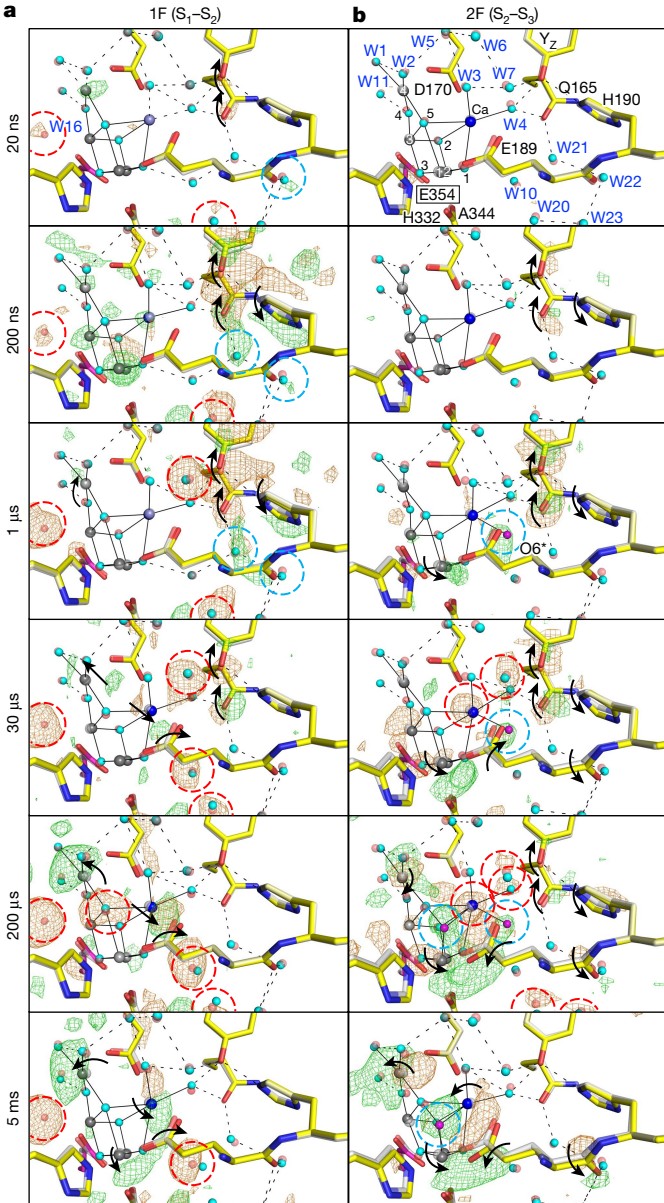

**Fig. 3 | Structural dynamics of the OEC during S₁–S₂–S₃ transitions.**
**a**,**b**, Structures of PSII in the OEC area are superposed with $F_{obs}(1F) - F_{obs}(Dark)$ (**a**) and $F_{obs}(2F) - F_{obs}(1F)$ (**b**) difference density maps contoured at $+4.0\sigma$ (green) and $-4.0\sigma$ (orange), with delay times from 20 ns to 5 ms. Residues of CP43 (a subunit of PSII) are shown in magenta and encircled by rectangles. The oxo-oxygen in the OEC and ligand waters are linked to the metal ions by black solid lines. The colour scheme used for other residues and atoms is the same as in Figs. 1 and 2.

Mn4 in the OEC (Fig. 3b and Supplementary Video 4), which results in an increase in the Mn1–Mn4 distance from 4.94 Å ($\Delta t2 = 30\,\mu s$) to 5.22 Å ($\Delta t2 = 200\,\mu s$), as well as an increase in the Mn1–Mn3 distance from 3.16 Å ($\Delta t2 = 30\,\mu s$) to 3.38 A ($\Delta t2 = 200\,\mu s$) (Extended Data Fig. 3 and Extended Data Table 2). In addition, the single negative difference density on Ca suggests the disorder of Ca (Fig. 3b and Supplementary Video 4). Of note, at $\Delta t2 = 200\,\mu s$, while O6* is still present, a positive difference density emerges in the location of O6, indicating that O6 is incorporated into the OEC. The structural changes to the OEC at $\Delta t2 = 30\,\mu s$ and $\Delta t2 = 200\,\mu s$ can be explained as follows: Mn1 undergoes oxidation from Mn1(III) to Mn1(IV), thereby attracting the negatively charged O6*, resulting in its translocation to the O6 position and the disorder of Ca. Simultaneously, Mn1 and Mn4 move outwards to create

room for O6 (the outward movement of Mn1 might also be triggered by its own oxidation). At $\Delta t2$ = 200 µs, the translocation of O6* is not yet complete, resulting in simultaneous observations of both O6* and O6.

We observed no apparent difference density on W3 at all time points (Fig. 3b, Supplementary Table 3 and Supplementary Video 4), which is inconsistent with a role of W3 as the entry point for the origin of O6 (refs. 7,35,36). W4 also seems an unlikely candidate for an entry point owing to spatial constraints, because W4 needs to pass through W3 to reach the O6 position (Extended Data Fig. 4b). The remaining potential pathway is a direct translocation of O6* to the O6 site (Extended Data Fig. 4b). Although the 3.08-Å distance between D1-E189 and Ca might not allow the passage of O6*, this distance represents the average distance observed both in PSII molecules that have successfully completed the O6* translocation and in those that have not, but not in PSII in which O6* is being translocated. Therefore, this distance might transiently extend when O6* is passing. Furthermore, an OH⁻ ion is smaller than a water molecule, so the direct translocation of O6* (OH⁻) is possible. By $\Delta t2$ = 5 ms, the translocation is completed, and O6 becomes the eighth ligand to Ca and the sixth ligand to Mn1 (Fig. 3b and Supplementary Video 4). The negative difference density near Ca indicates the inward movement of Ca towards the centre of the OEC. By contrast, Mn1 and Mn4 move further outwards from the OEC, which further opens the OEC (Fig. 3b, Extended Data Fig. 3, Extended Data Table 2 and Supplementary Video 4).

Determining the accurate positions of O5 and O6 using electron density alone at the current 2.25-Å resolution is challenging, owing to the influence of mixed populations of different $S_i$ states and the presence of neighbouring electron-rich metal ions. To refine the structures of OEC at $\Delta t2$ = 200 µs and $\Delta t2$ = 5 ms, we chose three O5–O6 distances of 1.9 Å, 2.4 Å and 2.2 Å, respectively, corresponding to oxyl/oxo, hydroxyl/oxo and deprotonated hydroxyl/oxo coupling species[37]. The optimal positions of O5 and O6 were determined with the smallest residual densities in the $mF_o$-$DF_c$ map, which showed an O5–O6 distance of 1.9 Å at $\Delta t2$ = 200 µs, whereas the residual densities at $\Delta t2$ = 5 ms were almost comparable for the O5–O6 distances of 1.9–2.4 Å (Extended Data Fig. 6). This suggests the existence of a mixed species at room temperature—different from what is observed at low temperature[6]. To maintain consistency, we set the O5–O6 distance at 1.9 Å for both $\Delta t2$ = 200 µs and $\Delta t2$ = 5 ms. We note that the OEC at $\Delta t2$ = 5 ms is more open as compared with the structure predicted by theoretical calculations and the OEC structure solved at cryo-temperature[6,37], as evidenced by the lengthened Mn1–Mn3 distance of 3.5 Å observed here. This might leave some room for a hydroxyl/oxo coupling mechanism, and there is a crystallographic debate regarding the existence of O6/Ox in the $S_3$ structure[38] (Supplementary Fig. 1 and Supplementary Discussion).

## Water inlet from the O1 channel

We observed previously that 1F leads to the disorder of two water molecules—one in the O1 channel and the other in the O4 channel[6,25]. The current study reveals the dynamic behaviour of the water molecules in these channels. At $\Delta t1$ = 30 µs to 5 ms, disorder of W10 and the concomitant movement of D1-D342 were observed, consistent with the previously solved structure of the $S_2$ state[25] (Extended Data Fig. 7a). At $\Delta t1 \leq$ 200 µs, dynamic difference densities appear in the O1 channel. These span from a five-water cluster located near the OEC to the PsbU-K104–D2-R348 (PsbU is a subunit of PSII) salt bridge near the lumen (Extended Data Fig. 7a). Difference densities appear at $\Delta t1$ = 20 ns on W20, W22 and the nearby D1-D342 main chain; these are likely to be induced by correlated movements of the neighbouring $Y_Z$ and D1-Q165 (Extended Data Fig. 7a). As $Y_Z$ and D1-Q165 move to a greater extent at $\Delta t1$ = 200 ns and $\Delta t1$ = 1 µs, the positive electron density on W22 increases and spreads to cover W21 and W22. Subsequently, $Y_Z$ and D1-Q165 move backwards at $\Delta t1$ = 30 µs to 5 ms, and the difference densities vanish (Extended Data Fig. 7a; see Fig. 3a for a closer view). By

contrast, the difference densities near the main chain of D1-D342 persist from $\Delta t1$ = 20 ns to $\Delta t1$ = 5 ms, indicating its shift throughout the $S_1$–$S_2$ transition. Furthermore, during $\Delta t1$ = 20 ns to $\Delta t1$ = 200 µs, movement of the D1-D342 main chain induces disorders or shifts of W20, W24, W52 and D1-E329, which are connected to D1-D342 by hydrogen bonds. At $\Delta t1$ = 30–200 µs, a negative difference density arises on W53′, which is located in the cavity surrounded by OEC-O1, D1-E189, D1-E329, D1-H332 and D1-D342 (Extended Data Fig. 7a), indicating that W53′ becomes further disordered. Here, W53 in the $S_1$ state is only observable under cryo-temperature conditions (PDB codes: 3WU2 (ref. 2) and 4UB6 (ref. 3)) but is not detectable at room temperature (PDB codes: 5WS5 (ref. 4) and 7CJI (ref. 25)). Therefore, we denote this invisible water as W53′ (Extended Data Fig. 1c and Extended Data Fig. 7a).

A negative difference density appears on the PsbU-K104 carboxy terminal at the O1-channel entrance during $\Delta t1$ = 200 ns–200 µs and subsequently disappears by $\Delta t1$ = 5 ms (Extended Data Fig. 7a). The PsbU-K104–D2-R348 salt bridge might function as a gate for the O1 channel, and the PsbU-K104 disorder implies the breakage or loosening of the salt bridge (Extended Data Fig. 7a), resulting in the opening of the gate and the entry of water into the giant cavity that houses Gol1, W55, W56, W59, W61 and W62 (Extended Data Fig. 7a).

At $\Delta t2$ = 20 ns to $\Delta t2$ = 200 ns, there is no difference density in the O1 channel (Extended Data Fig. 7b). At $\Delta t2$ = 1 µs, the most noticeable difference density occurs on O6* (Fig. 4b). When O6* is being prepared and translocated to O6 at $\Delta t2$ = 30 µs to $\Delta t2$ = 200 µs, negative difference densities appear on the main chains of D1-D342, D1-E329, W24, W52, W55 and Gol2, indicating that they are disordered during the O6* translocation. At $\Delta t2$ = 5 ms, the disordered components become ordered again after the completion of the O6* translocation. In addition, paired positive and negative difference densities appeared around CP43-V410 (CP43 is a subunit of PSII), suggesting a rotation of the CP43-V410 side chain by 120° rather than a mere shift as proposed previously[6] (Extended Data Fig. 7b). In conjunction with the CP43-V410 rotation, a partially occupied water designated as W74 emerges in the proximity of the pre-rotation conformation of CP43-V410 (Extended Data Figs. 5c and 7b). Furthermore, two small positive densities appear near Gol1 and PsbU-K104 at $\Delta t2$ = 5 ms, which suggests that two new water molecules become ordered at the end of the O1 channel after O6 translocation, similar to the findings of a previous study[6] (Extended Data Fig. 5c).

## Structural changes in the O4 channel

W16 is the second water in the O4 channel and is disordered after 1F (Fig. 4a). This disorder is maintained after 2F, until returning to a stable state after 3F (refs. 5–7). Disorder of W16 initiates at $\Delta t1$ = 20 ns, increases progressively, and reaches a maximum at $\Delta t1$ = 200 µs (Fig. 4a and Supplementary Table 3). The W16 disorder is expected to be influenced by the charge rearrangement that occurs at $\Delta t1$ = 20–200 ns, the oxidation of Mn4 at $\Delta t1$ = 1–30 µs and the stabilization of the remaining positive charge on the OEC at $\Delta t1$ = 200 µs–5 ms (Fig. 3a). One potential explanation is that the alternation of Mn4 charge influences W11 through O4, leading to the disruption of the hydrogen bond between W11 and W16 and the W16 disorder. The W16 disorder further affects the hydrogen-bonding network at the O4 channel, leading to shifts of W18, W31, W33 and W34 from $\Delta t1$ = 20 ns to $\Delta t1$ = 5 ms (Fig. 4a). In addition, when the difference densities in the O4 channel are strongest at $\Delta t1$ = 200 µs, the main chains of D1-R334-N335-A336 showed slight shifts towards the OEC. Movement of D1-D61 towards the OEC is also observed at $\Delta t1$ = 30 µs–5 ms (Fig. 4a).

No noticeable difference densities are observed in the O4 channel at $\Delta t2$ = 20 ns–1 µs. At $\Delta t2$ = 30 µs, negative difference densities emerge on W11, CP43-E354 and D1-D61 (Fig. 4b), indicating their instability during this period. The nearby CP43-M356 appears to move toward this region (Fig. 4b), possibly to fill the space. At $\Delta t2$ = 200 µs, W11 maintains its disorder, whereas CP43-E354, D1-D61 and CP43-M356 are restabilized

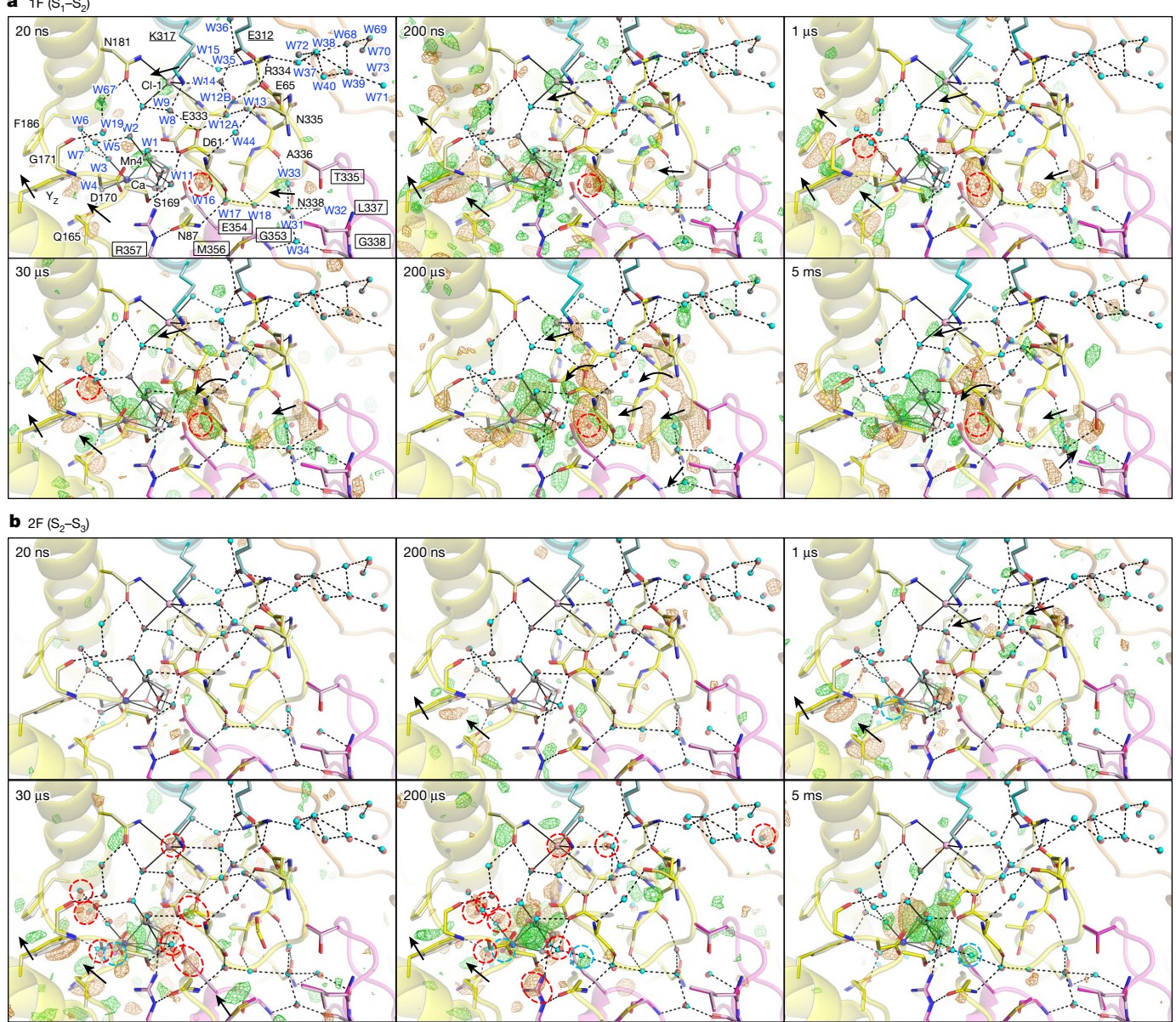

**Fig. 4 | Structural dynamics at the O4 and Cl-1 channels during $S_1$–$S_2$–$S_3$ transitions. a,b**, Structures of PSII at the O4 and Cl-1 channels are superposed with $F_{obs}(1F) - F_{obs}(Dark)$ (**a**) and $F_{obs}(2F) - F_{obs}(1F)$ (**b**) difference density maps contoured at $+3.5\sigma$ (green) and $-3.5\sigma$ (orange) with delay times from 20 ns to 5 ms. The intermediate structures of D1, D2, CP43 and PsbO proteins are depicted in yellow, cyan, magenta and orange, respectively, and the colours of other atoms are the same as those in Figs. 1 and 2.

(Fig. 4b). CP43-R357, which is located at a hydrogen-bonding distance to O4, becomes disordered at $\Delta t2 = 200$ μs. All of these residues and water molecules become ordered at $\Delta t2 = 5$ ms (Fig. 4b). The oxidation of the OEC that takes place at $\Delta t2 = 30$–200 μs could potentially contribute to the structural changes on W11, CP43-E354 and CP43-R357; all are directly connected to the OEC.

## Roles of Cl-1 in the S-state transitions

Difference densities near Cl-1 are observed at $\Delta t1 = 20$ ns–5 ms; however, they fluctuate over time, in contrast to the nearly continuously growing difference densities observed on W16 (Fig. 4a and Supplementary Table 3). The difference densities near Cl-1 arise at $\Delta t1 = 20$ ns, reach a maximum at $\Delta t1 = 200$ ns, decline at $\Delta t1 = 1$–30 μs, increase again at $\Delta t1 = 200$ μs and finally decrease at $\Delta t1 = 5$ ms. Considering the dynamics of the $Y_Z$ area and the OEC at $\Delta t1 = 20$ ns–5 ms, we hypothesize that

the electrostatic effect of $Y_Z$ and the OEC influences Cl-1, causing the fluctuation of the difference densities (Figs. 2a, 3a and 4a and Supplementary Table 3). The difference densities near Cl-1 are observed when $Y_Z$ is oxidized to $Y_Z^{\cdot+}$ at $\Delta t1 = 20$ ns (Figs. 2a and 4a). These difference densities reach their peaks when $Y_Z^{\cdot+}$ loses one proton, resulting in the formation of more oxidized $Y_Z^{\cdot}$ at $\Delta t1 = 200$ ns (Figs. 2a and 4a). Subsequently, difference densities near Cl-1 decrease during the reduction of $Y_Z^{\cdot}$ at $\Delta t1 = 1$ μs and $\Delta t1 = 30$ μs. The disruption of the hydrogen-bonding network between $Y_Z$ and Cl-1 could also contribute to the decreased signals (Figs. 2a and 4a). As the reduction of $Y_Z^{\cdot}$ is completed and the OEC is oxidized to OEC$^+$ at $\Delta t1 = 200$ μs, the difference densities near Cl-1 increase again (Figs. 3a and 4a). The subsequently decreased difference density at $\Delta t1 = 5$ ms could be attributable to a stabilization effect of the positive charge on the OEC. On the basis of these observations, Cl-1 might actively contribute to stabilizing the positively charged $Y_Z$ and OEC during the $S_1$–$S_2$ transition.

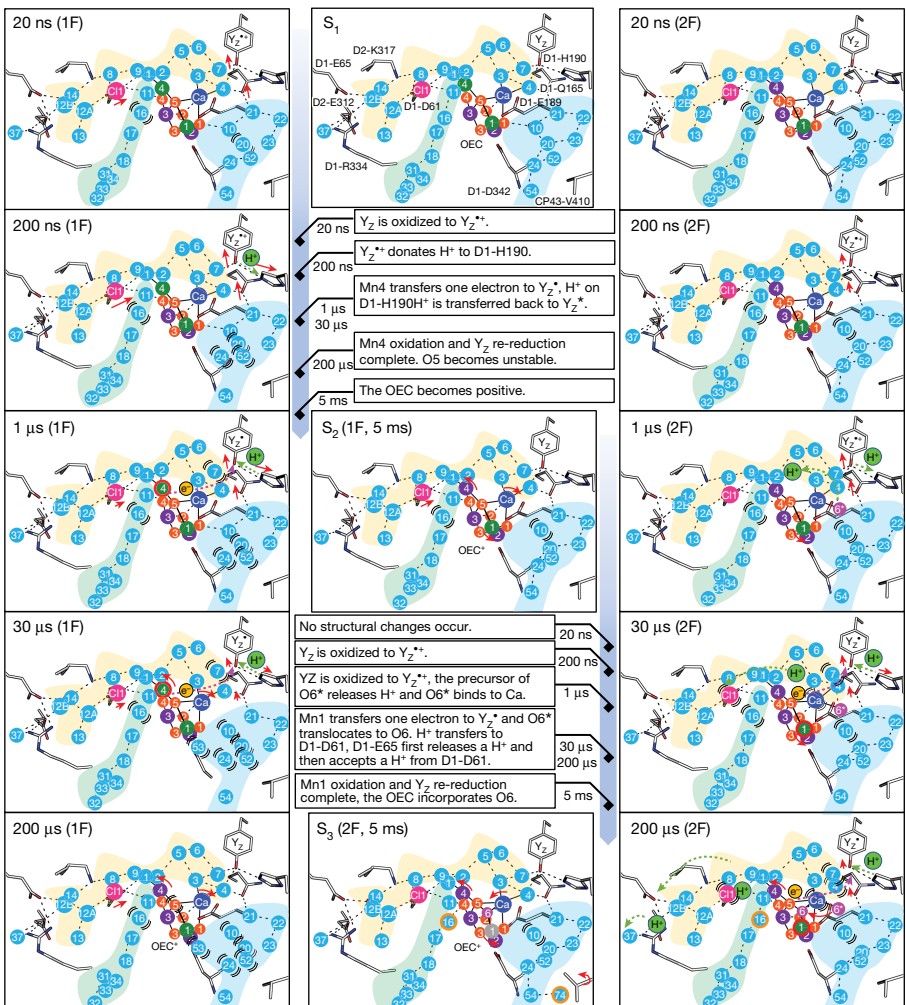

**Fig. 5 | Schematic of events occurring during $S_1$–$S_2$–$S_3$ transitions at the electron donor side.** The small orange spheres correspond to O1–O5 and are numbered 1 to 5 in the OEC. O6* and O6 are shown as magenta spheres. The larger green, purple and grey spheres represent Mn1–Mn4 with labels 1 to 4. Specifically, the green spheres correspond to Mn(III), the purple spheres correspond to Mn(IV) and the grey sphere represents either Mn(III) or Mn(IV). A red outer ring of the spheres signifies that the Mn ion is undergoing oxidation. The Cl-1, O4 and O1 channels are depicted in yellow, light green and cyan backgrounds, respectively. Water molecules are depicted as cyan spheres, with their corresponding numbers labelled. Disordered water molecules and other disordered atoms are depicted with arched lines, and an orange outer ring of the water molecules indicate that they become ordered. The red arrows indicate the movements of residues and atoms; the length of the arrows roughly represents the travelled distance for $Y_Z$ and Cl-1. The purple- and green-dotted arrows indicate the movements of electrons and protons, respectively. The proton transfer from $Y_Z$ to D1-H190 takes place between 2F (1 μs) and 2F (30 μs), which was depicted at 2F (1 μs) owing to the absence of time points between 1 μs and 30 μs.

Although paired positive and negative difference densities surrounding the two sides of Cl-1 were observed at $\Delta t1$ = 20 ns–5 ms (some negative densities are weaker, which cannot be visible at the contour level ±3.5σ), a single negative difference density was observed overlaying Cl-1 at $\Delta t2$ = 30 μs and $\Delta t2$ = 200 μs (Fig. 4b and Supplementary Table 3). This indicates the disorder of Cl-1 during this time period after 2F (Fig. 4b), which is apparently different from the movements of Cl-1 observed after 1F. These differences show that Cl-1 has different roles in the $S_1$–$S_2$ and $S_2$–$S_3$ transitions, and the instability of Cl-1 after 2F might reflect the proton transfer along the Cl-1 channel as described below.

## Proton transfer through the Cl-1 channel

After 2F, structural changes in the Cl-1 channel are relatively small compared with those at other sites, and take place mainly between $\Delta t2$ = 1 μs and $\Delta t2$ = 200 μs (Fig. 4b and Extended Data Fig. 8a). At $\Delta t2$ = 1 μs, W2 and D1-D61 become unstable, as indicated by the negative difference densities on them (Extended Data Fig. 8a), which might be attributable to the release of a proton from the precursor of O6* during this period.

Concomitantly, W44 becomes transiently stable, probably as a result of the movement of D1-E65, which is connected to W44 (Extended Data Fig. 8a). Another structural change occurring at $\Delta t2$ = 1 μs is the movement of the D1-H332 to D1-A336 backbones towards the OEC, which probably occurs owing to structural changes in the OEC. At $\Delta t2$ = 30 μs, a larger number of water molecules in the hydrogen-bonding network become unstable (W1, W3–W7 and W11), and Cl-1 also starts to become unstable (Extended Data Fig. 8a). These structural changes imply that the proton is transferred to D1-D61 (Extended Data Fig. 8b). The movement of the D1-H332 to D1-A336 backbones is transmitted to the D1-R334 and D1-N335 side chains, leading to a shift of D1-R334 towards the OEC. This results in an instability of W14, and could potentially influence the gate between D1-E65, D2-E312 and D1-R334 (refs. 8,17). At $\Delta t2$ = 200 μs, water molecules in the bulk region (W37, W70 and W73) also become unstable, in addition to the unstable water molecules near the OEC and Cl-1 (W1, W3–W5, W7–W9, W11 and W14) (Extended Data Fig. 8a), suggesting a possible proton transfer to the lumen. The instability of Cl-1 further increases at $\Delta t2$ = 200 μs, and D1-R334 also exhibits instability (Extended Data Fig. 8a), presumably reflecting the pass of

the proton[39,40]. By $\Delta t2$ = 5 ms, the structural changes in the gate area, as well as the high mobilities of water molecules observed at $\Delta t2$ = 200 μs, disappear entirely, and Cl-1 becomes ordered and returns to its original position (Extended Data Fig. 8). This indicates that the Cl-1 channel has been restored and reset to the subsequent S-state transition.

In conclusion, our time-resolved SFX experiments reveal the important roles of protein structural dynamics in electron transfer, water insertion, proton release and O–O bond formation in PSII. We summarize our results in a model presented in Fig. 5, and a detailed discussion is provided in the Supplementary Information.

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

## Methods

### Sample preparation

Samples of the PSII microcrystals were prepared as in the previous SFX studies conducted at room temperature[4,25], with a few minor adjustments. In brief, cells of the thermophilic cyanobacterium *Thermosynechococcus vulcanus* were grown in a previously described medium[41–44] in eight 5-l bottles, to a density of $OD_{730\,nm} = 2.5$–3.0, and collected as described previously[41–44]. The cells were resuspended in a buffer of 40 mM $KH_2PO_4$-KOH (pH 6.8) and 0.4 M mannitol, and treated with 1.21 g l$^{-1}$ lysozyme (FUJIFILM Wako Pure Chemical Corporation) at 37 °C for 90 min with constant shaking. The treated cells were pelleted by centrifugation at 13,700$g$ for 15 min, suspended in 25% (w/v) glycerol, 20 mM HEPES-NaOH (pH 7.0) and 10 mM $MgCl_2$ (buffer A), and stored at −80 °C until use.

The frozen cells were thawed, to which ten folds of a buffer containing 30 mM HEPES-NaOH (pH 7.0) and 10 mM $MgCl_2$ were added to disrupt the cells by freeze-thawing and osmotic shock. After centrifugation at 13,700$g$ for 15 min, pelleted thylakoids were suspended in 5% (w/v) glycerol, 20 mM HEPES-NaOH (pH 7.0) and 10 mM $MgCl_2$. Crude PSII particles were obtained from the thylakoids by a two-step solubilization with a detergent *N,N*-dimethyldodecylamine *N*-oxide (LDAO) (Sigma-Aldrich, 40236-250ML). In the first step, the thylakoids were treated with 0.16% (w/v) LDAO for 5 min on ice, and centrifuged at 43,200$g$ for 60 min. The pellet obtained was suspended in buffer A, and treated with 0.27% (w/v) LDAO for 5 min again. The mixture was centrifuged at 100,000$g$ for 1 h, and the supernatant was recovered. After the addition of 50 (w/v) polyethylene glycol (PEG) 1450 to a final concentration of 15%, crude PSII particles were recovered by centrifugation at 100,000$g$ for 30 min, and resuspended in buffer A[41–44].

The PSII crude particles were treated with 1.0% *n*-dodecyl-β-D-maltoside (β-DDM) (FUJIFILM Wako Pure Chemical Corporation, D316) for 5 min, and loaded onto a Q-Sepharose high-performance column (Cytiva) pre-equilibrated with 5% (w/v) glycerol, 30 mM MES-NaOH (pH 6.0), 3 mM $CaCl_2$ and 0.03% β-DDM (buffer B) in a cooled chamber at 6 °C. The column was washed with eight to ten folds of the column volume of buffer B containing 170 mM NaCl, and eluted with a liner gradient of 12.5 folds of the column volume of 170–300 mM NaCl in buffer B. Elution peaks first appeared for PSII monomer, followed by PSII dimer and PSI monomer, among which PSII dimers were collected. The PSII dimers collected were diluted threefold by buffer B without DDM, and PEG 1450 was added to a final concentration of 13%. The PSII dimers were centrifuged at 100,000$g$ for 30 min, and the pellet was suspended in buffer B without DDM and stored in liquid nitrogen until use[41–44].

To make microcrystals of the PSII dimer, the sample was diluted with 20 mM MES-NaOH (pH 6.0), 40 mM $MgSO_4$, 20 mM NaCl and 10 mM $CaCl_2$, followed by additions of *n*-heptyl-β-D-thioglucopyranoside (HTG) (FUJIFILM Wako Pure Chemical Corporation, H015) and PEG 1450 to final concentrations of 0.85% (w/v) and around 5.50–5.75% (w/v), respectively, at a final concentration of 2.25 mg chlorophyll per ml (refs. 4,6). Microcrystals were grown in a 2.0-ml glass vial (J.G. Finneran Associates, 9800-1232), and 150 µl PSII dimer sample was put into each vial. After standing for 20–30 min at 20 °C, the solution was mixed gently and left to stand for another 10–30 min to allow the microcrystals to grow. In cases in which microcrystals did not appear or appeared in small numbers, the mixing-and-standing step was repeated until enough microcrystals appeared.

After the microcrystals appeared, they were allowed to grow to a maximum size of 100 µm in length for several hours to overnight, following which 150 µl of a crystal storage buffer containing 7% (w/v) PEG 1450, 20 mM MES-NaOH (pH 6.0), 20 mM NaCl, 10 mM $CaCl_2$ and 0.85% (w/v) HTG was added to stop the growth of the microcrystals. After collection of the microcrystals, the supernatant was discarded, and the microcrystals were stored in the crystal storage buffer at 20 °C until the X-ray free electron laser (XFEL) experiments. It is important to store the microcrystals in the crystal storage buffer for more than 24 h to ensure high resolution, and they are stable in the crystal storage buffer for at least three days but not more than seven days[4,6].

Before conducting the diffraction experiment, a 10 mM potassium ferricyanide solution was added to the PSII microcrystal solution under dim green light, and one pre-flash was given at 20 °C with a laser at a wavelength of 532 nm and an energy of 52 mJ cm$^{-2}$. The microcrystals were subsequently transferred to 7% (w/v) PEG 1450, 20 mM MES-NaOH (pH 6.0), 20 mM NaCl, 10 mM $CaCl_2$, 0.85% (w/v) HTG, 2% dimethyl sulfoxide (DMSO) and 10 mM potassium ferricyanide, and incubated for 10 min at 20 °C. The solution was finally replaced by a cryoprotectant solution containing 10% (w/v) PEG 1450, 10% (w/v) PEG monomethyl ether 5000, 23% (w/v) glycerol, 20 mM NaCl, 10 mM $CaCl_2$, 0.85% (w/v) HTG, 2% DMSO and 10 mM potassium ferricyanide for six steps, with each step for 10 min at 20 °C (refs. 4,6).

After replacement of the solution with the cryoprotectant solution, PSII microcrystals were gently mixed with a vacuum grease of a nuclear power grade (Super Lube, 42150)[45]. The ratio of grease to microcrystals was 200 µl to 50 µl (obtained from 4–5 mg chlorophyll), and to avoid physical damage to the microcrystals, the mixing was conducted gently for 2 min. The mixture was exposed to air at 20 °C for around 30–60 min to dehydrate further, before being used for the diffraction experiments at room temperature in darkness[4]. The total time from cryoprotectant replacement to XFEL experiments was one to two hours.

### Diffraction experiment

The dark and 1F data, as well as the 1F and 2F time-delayed data, were collected in two independent experiments, resulting in a total of 14 experimental datasets (Extended Data Table 1). Unless otherwise stated, the experimental set-ups were identical for both beamtimes. Diffraction images were obtained using single-shot XFELs collected at the BL2 beamline in the SPring-8 Ångstrom Compact Free Electron Laser (SACLA)[46]. The parameters of the XFEL pulses were as follows: pulse duration 10 fs, energy 10 keV, beam size 3.0 µm (H) × 3.0 µm (W) and repetition rate 10 Hz (ref. 4). The PSII microcrystals were excited using pump lasers with the following parameters: pulse duration 6 ns (FWHM, Gaussian), energy 42 mJ cm$^{-2}$, focused spot size 240 µm (top-hat), wavelength 532 nm and frequency rate 10 Hz (ref. 4). To ensure efficient excitation, one laser beam was split into two beams that focused on the same point of the sample from two different directions separated by an angle of 160° (ref. 4).

The injector containing the mixture of PSII microcrystals and grease was carefully inserted into a sample chamber, in which the mixture was ejected from the injector using liquid pressure, ultimately forming a micrometre-sized liquid stream[47,48].

The sample flow rate is regulated by adjusting the fluid pressure in the injector. For the 'dark' sample, the flow rate is 1.99 µl min$^{-1}$, whereas for the 'light' samples, it is 7.80 µl min$^{-1}$. As described previously, by maintaining this flow rate, contamination from the prior lasers is effectively avoided[25]. The dark dataset was obtained by directly exposing the sample stream to XFELs, whereas the 1F and 2F datasets were acquired by illuminating the sample stream with the pump laser first, followed by exposure to the XFELs after a specified delay time $\Delta t$. The values of $\Delta t1$ and $\Delta t2$ were set to 20 ns, 200 ns, 1 µs, 30 µs, 200 µs and 5 ms, respectively (Extended Data Fig. 1d). In addition, in the 2F time-delay experiment, the time interval between the first and second flash was set to 5 ms (Extended Data Fig. 1d), which is enough to fully transform the $S_1$ state to the $S_2$ state after 1F. The focal centres of the lasers and XFELs were the same for data with a $\Delta t$ of 20 ns–200 µs, but for data with a $\Delta t$ of 5 ms, the focal centres of lasers were set 60 µm higher than those of the XFELs to prevent the light-excited microcrystals from escaping the XFEL irradiations after a $\Delta t$ of 5 ms. Diffraction spots were recorded using a Rayonix MX300-HS detector, which was positioned 240 mm from the sample.

## Data processing

During the beamtime, we used Cheetah[49] (https://github.com/keit-aroyam/cheetah) and CrystFEL (v.0.6.3)[50,51] to observe and analyse the diffraction images. The analyses provide hit rates, the number of indexed images and approximate resolutions for each dataset, which greatly aided us in devising an effective data-collection strategy. For the processing of diffraction images at the beamline, we at first used approximately 10,000 indexed diffraction images from lysozyme crystals to determine the beam centre and camera length accurately. These parameters were then supplied to CrytFEL for processing the PSII diffraction images. The PSII diffraction images were indexed with 'indexamajig', using the Dirax[50,51] indexing method with unit-cell parameters of $a = 124.7$ Å, $b = 229.89$ Å, $c = 285.5$ Å, $\alpha = \beta = \gamma = 90°$ adopted from PDB code 5WS5 (ref. 4). The resulting individual intensities were merged using 'process_hkl' and the reflection data were evaluated using 'compare_hkl' (refs. 50,51).

After data collection, 'cctbx.xfel' was used for the indexing and integration of diffraction images, as well as for merging reflections[52,53]. The accuracy of the beam centre and camera length obtained from CrystFEL were verified by using the program 'cspad.cbf_metrology' (refs. 52,53). The PSII diffraction images were indexed and integrated using 'dials.stills_process' (ref. 54), incorporating the determined detector information and targeted unit-cell parameters mentioned above. Individual reflections were merged by the program 'cxi.merge' (refs. 52,53) with the post-refinement rs2 algorithm, and a filter based on the value of I/sigma was not applied so as to include weak signals at high resolutions. The average unit cell, calculated from all of the datasets collected in the same experiment, was used to merge each individual dataset once again. All datasets were processed to a resolution of 2.15–2.30 Å on the basis of the criteria of $CC_{1/2}$ of around 50% (Extended Data Table 1).

## Structural refinement for the dark and 1F datasets

Molecular replacement for the dark data was performed using Phaser-MR from PHENIX[55] with the PSII structure solved at 2.35-Å resolution and at room temperature (PDB code: 5WS5) as the search model, in which water molecules and the OEC were removed[4]. Next, rigid body refinement was applied to the resultant model for one cycle. Subsequently, the B factor was set to 20 for all atoms in the model, and the atomic coordinates and temperature factors of atoms were refined by 'Phenix.refine' in the resolution range of 2.15–20.0 Å, in conjunction with manual modifications by Coot[56]. We iteratively carried out reciprocal space refinement using 'Phenix.refine' and real-space refinement using Coot until the structures of residues and cofactors were confined. Then, the OEC and water molecules were added to the model. Geometric restraints of the OEC are based on the $Mn_4CaO_5$ cluster solved at 2.15 Å using microcrystals at cryo-temperature (PDB code: 6JLJ)[6], with a loose distance restraint of $\sigma = 0.06$ Å on Mn–O and Ca–O distances, whereas no restraints were provided for the Mn–Mn and Mn–Ca distances. Any pre-existing water molecules exhibiting negative $mF_o$-$DF_c$ signals or lacking $2mF_o$-$DF_c$ signals were removed from the model. New water molecules were constructed at the positions of positive spherical $mF_o$-$DF_c$ signals over $4\sigma$, and these water molecules were examined after subsequent rounds of reciprocal and real-space refinements to confirm. Finally, a TLS refinement was applied.

For the refinement of the 1F model in the two-flash time-delay experiments, we assigned a single conformation to the OEC and ligands, considering that the geometry of the OEC does not differ much between $S_1$ and $S_2$ states. During the refinement process, the Mn–Mn and Mn–Ca distances were not restrained, whereas the distances of Mn–O and Ca–O were restrained to the values observed in the 1F model solved at 2.15 Å (PDB code: 6JLK)[6], and refined with a loose restraint ($\sigma = 0.06$ Å). W16 was removed from the model owing to the emergence of a negative $mF_o$-$DF_c$ signal when W16 was present, even at low occupancy.

Conversely, W10 was retained because its deletion resulted in a significant positive $mF_o$-$DF_c$ signal at the corresponding location.

## Difference-map calculations and structural refinement of intermediates

The phases obtained from the well-refined dark and 1F models were used to calculate isomorphous-difference Fourier maps between dark and 1F time-delayed data, and between 1F and 2F time-delayed data, respectively. Substantial difference densities were detected in the $Q_A$–$Q_B$, P680, $Y_Z$ and OEC channel regions at each time point, with their locations dynamically varying over time (Figs. 1–4 and Extended Data Fig. 7). To refine the dynamic intermediate structures conveniently and effectively, we devised double conformations for all residues, water molecules and ligands within a spherical range of 20 Å centred on the Ca of the OEC and the non-haem iron, with A and B conformations corresponding to structures of the ground state and intermediate state, respectively. In this case, unstable water molecules and residues in the intermediate state were also built into the structures. Whether to preserve or delete these water molecules is decided by examining the $mF_o$-$DF_c$ signal. For example, in the case of W16, which became very unstable after 1F, building two conformations resulted in a strong negative signal on W16. Therefore, we deleted the B conformation of W16. On the other hand, for other unstable water molecules, such as W7 and W10, building two conformations did not result in a particularly strong negative $mF_o$-$DF_c$ signal, so their B conformations were preserved. Populations of $S_i$ state in PSII crystals were estimated to be 0.4/0.6 for $S_1/S_2$ after 1F and 0.49/0.51 for $S_2/S_3$ after 2F, on the basis of flash-induced Fourier transform infrared (FTIR) measurements[4,6,57]. On the basis of these ratios, we constructed the 1F structure by adopting two conformations for those atoms or residues that showed structural changes between $S_1$ and $S_2$. The $S_2$-state structure was refined against the density map, whereas the $S_1$-state structure was taken from the dark structure solved in the present study. On the other hand, in the 2F data, the structure of PSII that does not advance to the $S_3$ state is a mixture of $S_1$ and $S_2$. Owing to the small structural changes between $S_1$ and $S_2$, we fixed the structure to the $S_2$ state for PSII that does not advance to the $S_3$ state after 2F, and refined the $S_3$-state structure against the density map. These assignments do not pose major problems for modelling the structures according to the densities obtained. We refined the $xyz$ coordinates of the B conformation, followed by refining the B factors of both the A and the B conformation, and applied TLS refinement at last.

O6* was modelled as a water molecule with an occupancy of 0.51, without imposing artificial constraints on its distance to Ca and the nearby water molecules. The structures of the OEC containing O6 at $\Delta t2 = 200$ μs and $\Delta t2 = 5$ ms were investigated using three different O5–O6 distances: 1.9 Å, 2.2 Å and 2.4 Å, as indicated by theoretical calculations[37,58]. The optimal distance was determined by assessing the magnitude of the adjacent $mF_o$-$DF_c$ signals (Extended Data Fig. 6).

We need to point out that, although the XFEL data collected in the present study have a high quality, and the resolutions obtained are high, uncertainties exist with regard to the subtle structural changes that occur during $S_1$–$S_2$–$S_3$ transitions, and it is important not to over-interpret the crystallographic data presented in this study.

## Estimation of errors in inter-atomic distances

To estimate the errors in the inter-atomic distances, we used the resampling method, creating ten substructures with reduced data multiplicity. Subsequently, we calculated the standard deviations of atom–atom distances within these ten substructures. We resampled our XFEL data by the jackknifing method[59]. We began with a dataset consisting of 100% images and created ten sub-datasets by merging 75% randomly selected images. Subsequently, we refined ten substructures against these sub-datasets. To initiate the refinement of the substructures, we used the well-refined structure derived from the 100% image dataset as our starting model, resetting the temperature factors of all atoms

to 20 $Å^2$ and applying simulated annealing. After this, we performed refinements on the rigid body, atom position coordinates, temperature factors and TLS. The standard deviations of atom–atom distances were calculated across the ten substructures, which were used as estimates of the errors associated with the corresponding atom–atom distances in the determined structures (Extended Data Fig. 3).

## Density functional theory calculations

An OEC model of the $S_3$ state for density functional theory (DFT) calculations was constructed from the XFEL model (monomer A) of PSII (PDB code: 6JLL)[6]. This model comprises 408 atoms, including the inorganic $Mn_4CaO_5$ cluster, 4 terminal aqua/hydroxo ligands at Ca and Mn4, 15 crystal waters along with one extra hydroxide anion referred to as O6*, one chloride anion, and the following amino acid residues: D1-D61, D1-N87, Yz, D1-Q165, D1-S169, D1-D170, D1-N181, D1-V185, D1-F182 (backbone only), D1-E189, D1-H190, D1-N296, D1-N298 (fragment), D2-K317 (fragment), D1-H332, D1-E333, D1-A336, D1-H337, D1-D342, D1-A344 (C terminus), CP43-E354, CP43-R357, CP43-L401, CP43-V410 and CP43-A411. The revision made to the previous computational model[6,37] involves augmenting it with the incorporation of five water molecules next to O6*, called a 'water wheel', along with four supporting amino acid residues (D1-N296, CP43-L401, CP43-V410 and CP43-A411). Geometric optimizations for the hydroxo form of O6* bound to the Ca site of $(Mn^{IV})_3Mn^{III}CaO_5$ were carried out at multiplicity 14 ($M_S = 13/2$) using the B3LYP hybrid functional[60] augmented with the D3 version of Grimme's empirical dispersion correction and the Becke–Johnson damping function[61,62], in combination with the Los Alamos (LANL2DZ) pseudopotential basis set for Ca and Mn and 6-31G(d) for all other atoms[63–66]. A crucial requirement for the production of meta-stable $Ca^{2+}$-bound hydroxo form of O6*, as displayed in Extended Data Fig. 5a,b, is the absence of a $Y_Z$ radical ($Tyr_Z–O^•...^+HN–His190$), as the $pK_a$ value of $Ca^{2+}$-bound water (around 12.7 in aqueous solution)[67,68] might be much higher than that of the histidine residue (6.0) (ref. 69), even within the protein environment.

## Reporting summary

Further information on research design is available in the Nature Portfolio Reporting Summary linked to this article.

## Data availability

The atomic coordinates and structure factors have been deposited in the PDB under the following accession codes: 8IR5 for 0F (dark, ground state for the $\Delta t1$ structures), 8IR6 for $\Delta t1 = 20$ ns, 8IR7 for $\Delta t1 = 200$ ns, 8IR8 for $\Delta t1 = 1$ μs, 8IR9 for $\Delta t1 = 30$ μs, 8IRA for $\Delta t1 = 200$ μs, 8IRB for $\Delta t1 = 5$ ms, 8IRC for 1F (ground state for the $\Delta t2$ structures), 8IRD for $\Delta t2 = 20$ ns, 8IRE for $\Delta t2 = 200$ ns, 8IRF for $\Delta t2 = 1$ μs, 8IRG for $\Delta t2 = 30$ μs, 8IRH for $\Delta t2 = 200$ μs and 8IRI for $\Delta t2 = 5$ ms. All other data with a PDB code used in this study are from the PDB data bank.

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

**Acknowledgements** We thank T. Nakane and K. Yamashita for their assistance in data processing and structural analysis. The XFEL experiments were performed at SACLA approved by the Japan Synchrotron Radiation Research Institute (JASRI) (proposals 2018A8037, 2018A8010, 2018B8029, 2018B8055, 2019A8019, 2019A8032, 2019B8028, 2019B8020, 2020A8003, 2020A8059, 2021A8003, 2021B8012 and 2022A8007), and we thank the staff at SACLA for their help. The best condition for the collection of diffraction data was determined at beamlines 41XU and 44XU in SPring-8 (proposals 2018B2530, 2019A2559, 2019B2559, 2020A2550, 2021A2550, 2021A2741, 2021B2741, 2021B6618, 2022A2728 and 2022B2728). This research was supported by MEXT KAKENHI JP17H06434 (J.-R.S.) and JP22H04916 (J.-R.S., F.A., K.Y. and M. Suga), JP23H02450, JP22H04754, JP20H03226, JP20H05446 and JST PREST grant JPMJPR18G8 (M. Suga), JP19H05784 (M.K.); a Platform Project for Supporting Drug Discovery and Life Science Research (Basis for Supporting Innovative Drug Discovery and Life Science Research (BINDS)) from AMED under grant number JP21am0101070; and MEXT KAKENHI JP19H05777 (S.I.).

**Author contributions** J.-R.S. conceived the project. Y.N., F.A., S.K., N.M. and S.Y. made the samples. H.L., Y.N., E.N., K.H., F.L., R.T., F.A., K.K., J.K., Y.S., S.K., H.Y., N.M., H.F., M. Sugahara, M. Suziki, T.M., T.K., T.N.T., S.Y., L.-J.Y., T.T., M. Suga and J.-R.S. participated in the data collection. K.T., Y.J., T.H., E.N., S.I. and M.Y. developed the data-collection set-up. S.O., D.Y. and M.K. developed the laser set-up. H.L., K.H. and M. Suga processed the diffraction data and analysed the structure. H.I. and K.Y. performed theoretical calculations. H.L., M. Suga and J.-R.S. wrote the manuscript, and all authors contributed to the discussion and improvement of the manuscript.

**Competing interests** The authors declare no competing interests.

**Additional information**
**Correspondence and requests for materials** should be addressed to Michihiro Suga or Jian-Ren Shen.

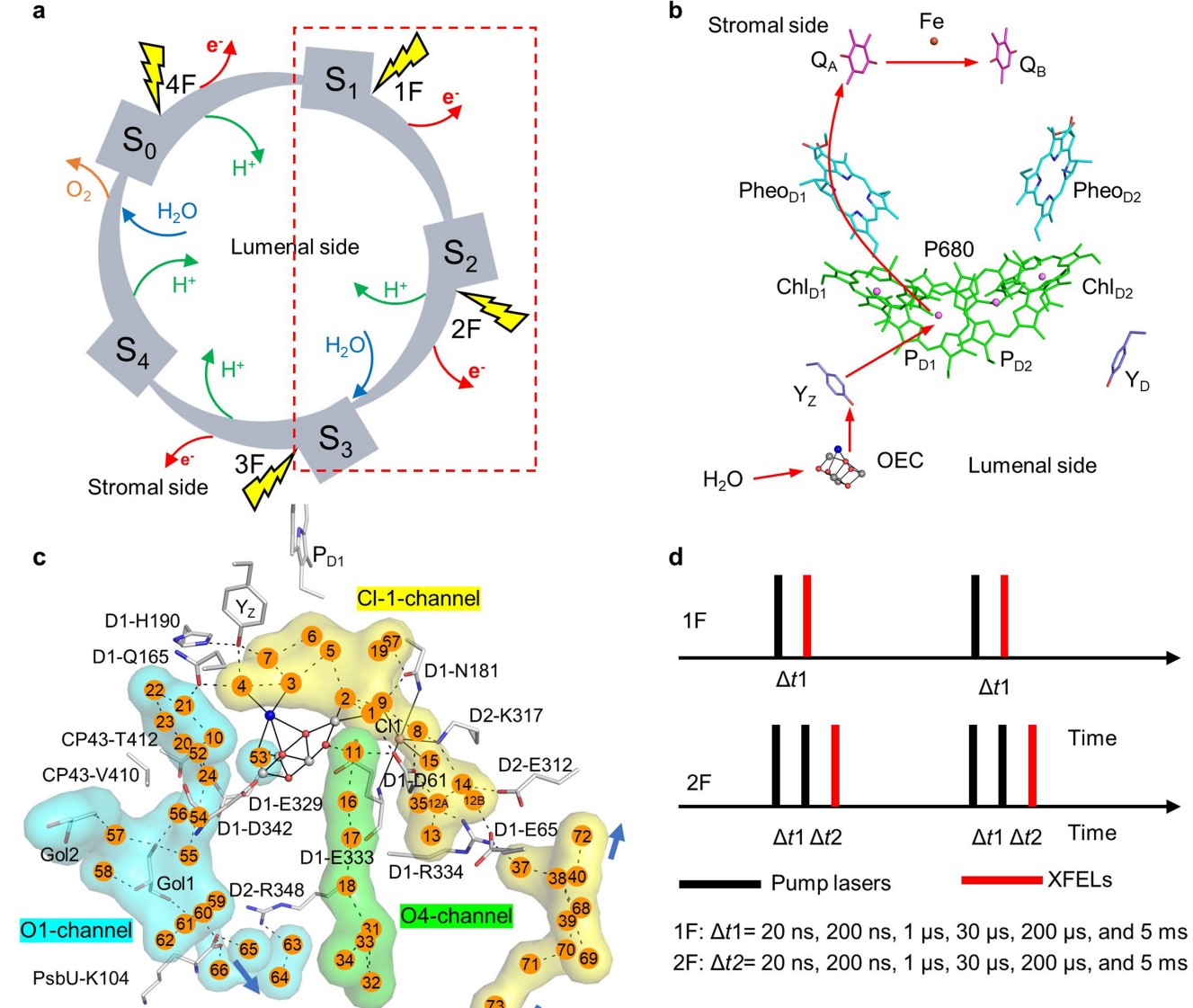

**Extended Data Fig. 1 | The Kok cycle and schematic of the present study. a**, The Kok cycle. A rectangular box circled by red, dotted lines highlights the major objective of this study. **b**, Electron transfer pathway in PSII. Red arrows indicate the path of the electron. **c**, Water channels connecting the OEC with the lumenal surface. Blue arrows indicate the exits of water channels. 53′ indicates a water molecule not visible in the current model. **d**, Schematic of the pump-probe TR-SFX using one (1F) or two flashes (2F) with delay times ranging from 20 ns to 5 ms. In the 2F experiment, the interval between the first and second flash is 5 ms.

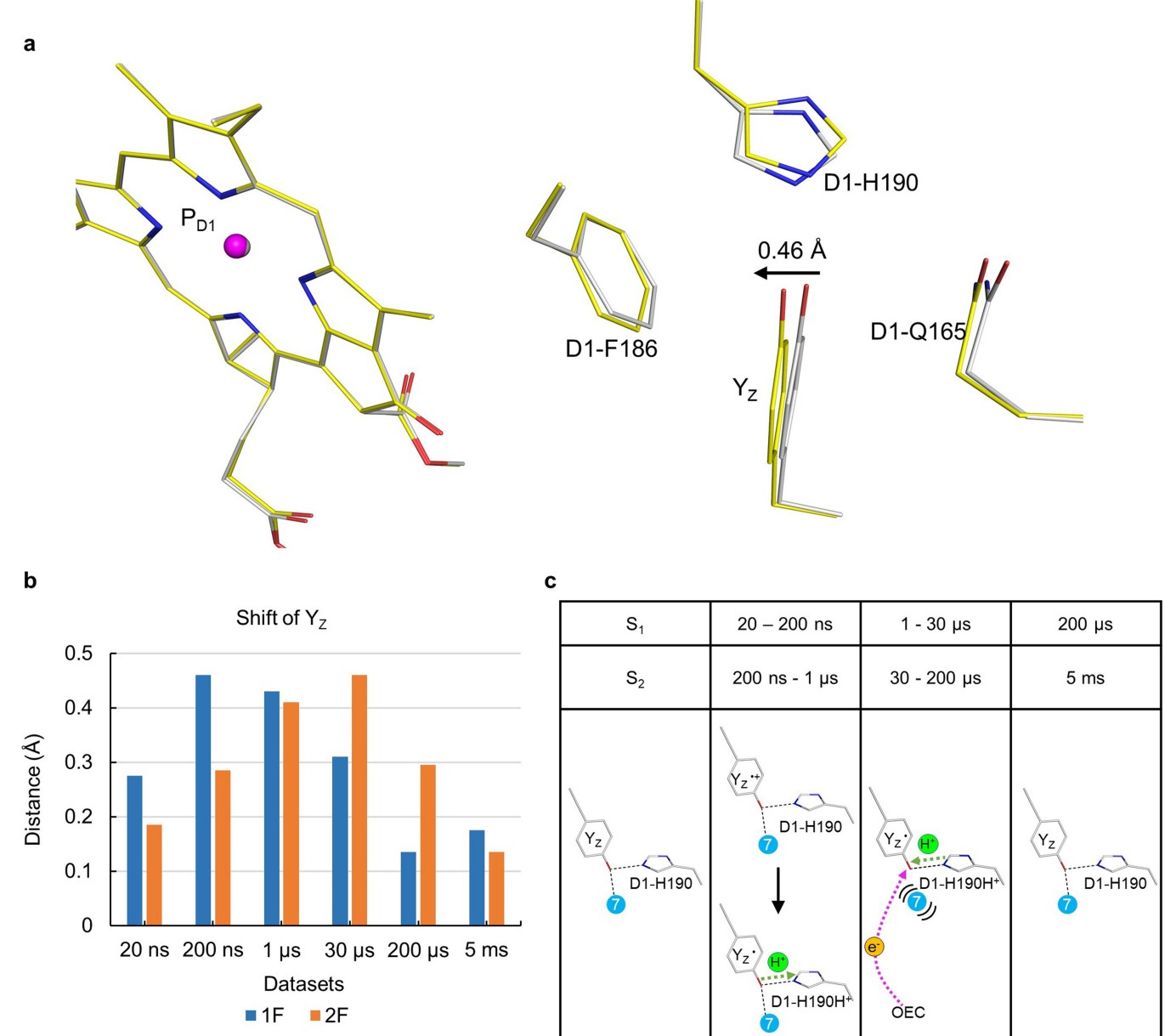

**Extended Data Fig. 2 | Redox status of $Y_Z$. a**, Overlap of the structures of dark (grey stick) and 1F (200 ns) (yellow stick). The black arrow indicates the direction of the $Y_Z$ shift, with 0.46 Å representing the shift distance of the phenolic oxygen. **b**, Distances of the shift of the phenolic oxygen at different time points after 1F and 2F. **c**, Redox status of $Y_Z$ at various time points following 1F and 2F. The black arrow at $\Delta t1$ = 20–200 ns and $\Delta t2$ = 200 ns–1 µs (and possibly between 1–30 µs) indicates the transformation of $Y_Z$. Dotted arrows in green and magenta represent the translocations of proton and electron, respectively. The cyan sphere labelled with '7' represents a water molecule W7, and curves surrounding it at $\Delta t1$ = 1–30 µs and $\Delta t2$ = 30–200 µs indicate its disordered structure.

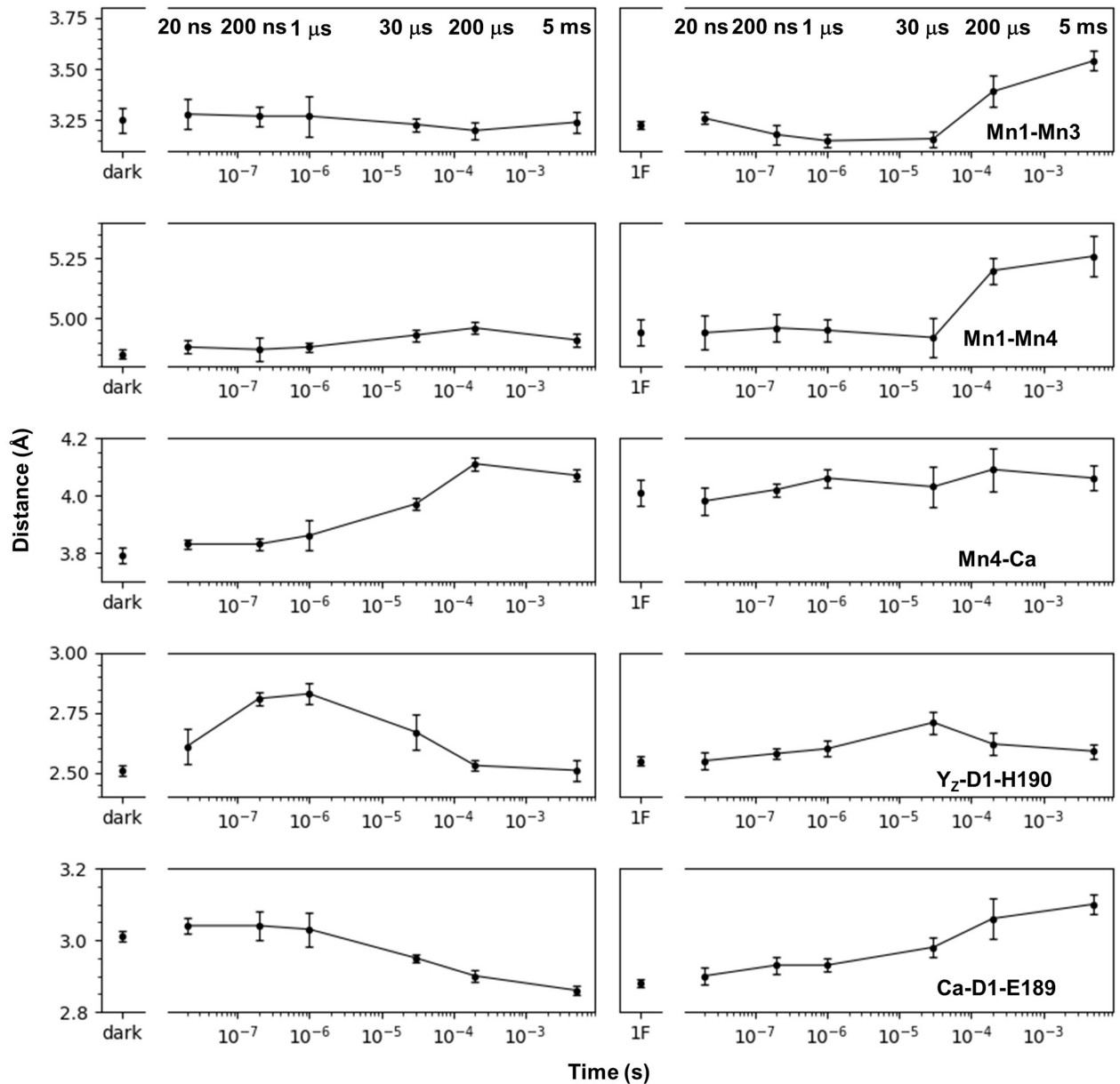

**Extended Data Fig. 3 | Variance in inter-atomic distances during $S_1$–$S_2$–$S_3$ transitions.** Inter-atomic distances are measured at all time points collected after 1F (left) and 2F (right). The mean values for the error bars are obtained with the structures derived from 100% indexed images. The error bars are estimated using a resampling approach, with the standard deviations determined using ten structures derived from ten sub-datasets, each comprising 75% randomly selected indexed images (see Methods for more details).

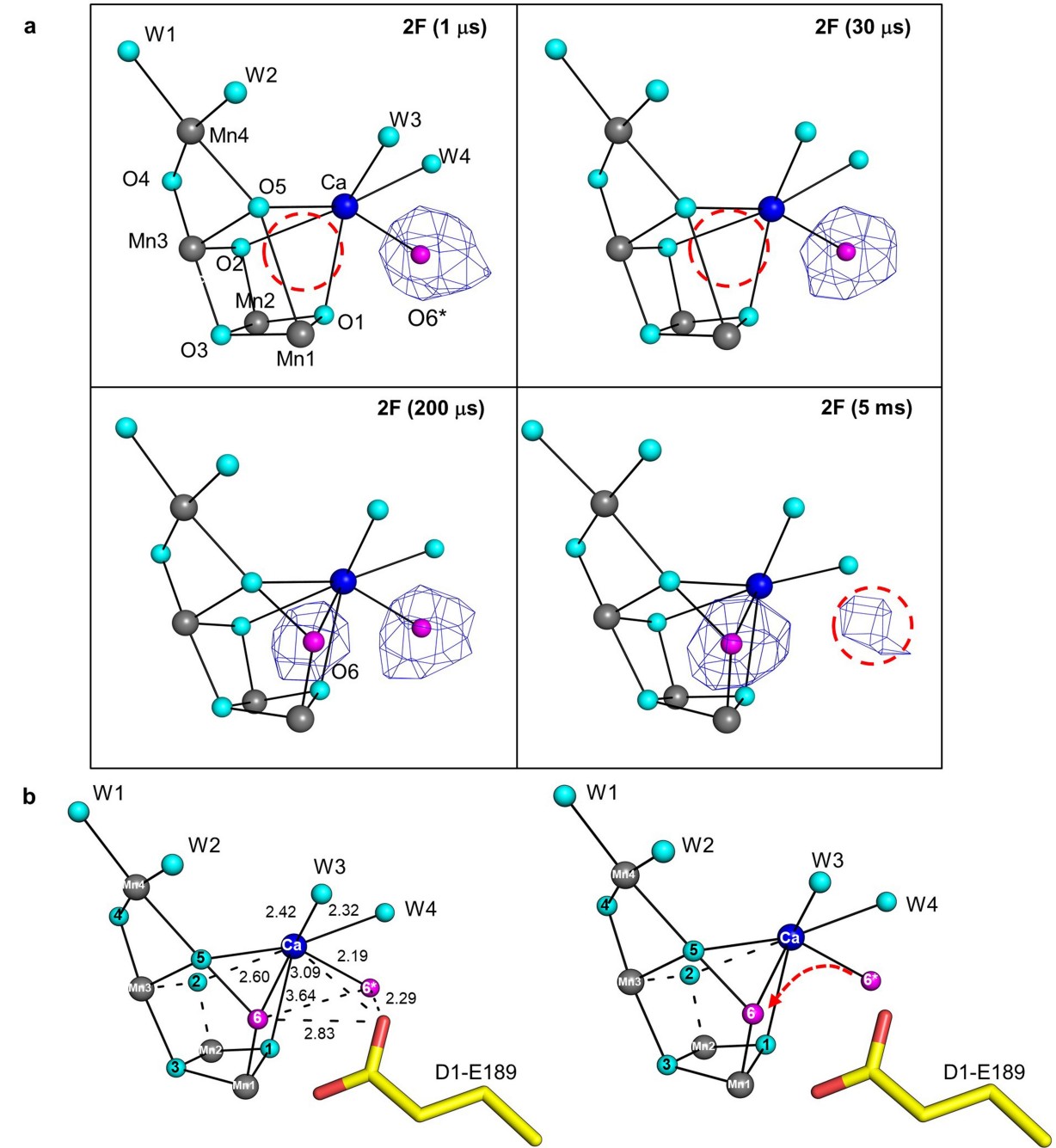

**Extended Data Fig. 4 | Translocation of O6* to O6. a**, Polder omit maps contoured at +3.0σ (blue mesh) of O6* and O6 are superposed with OEC models after 2F. The red dashed circles indicate positions where O6 or O6* is either absent or weak during that specific time point. **b**, Translocation pathway of O6*. Numbers represent the inter-atomic distances in angstrom (Å). The red line represents the potential translocation pathway for O6*.

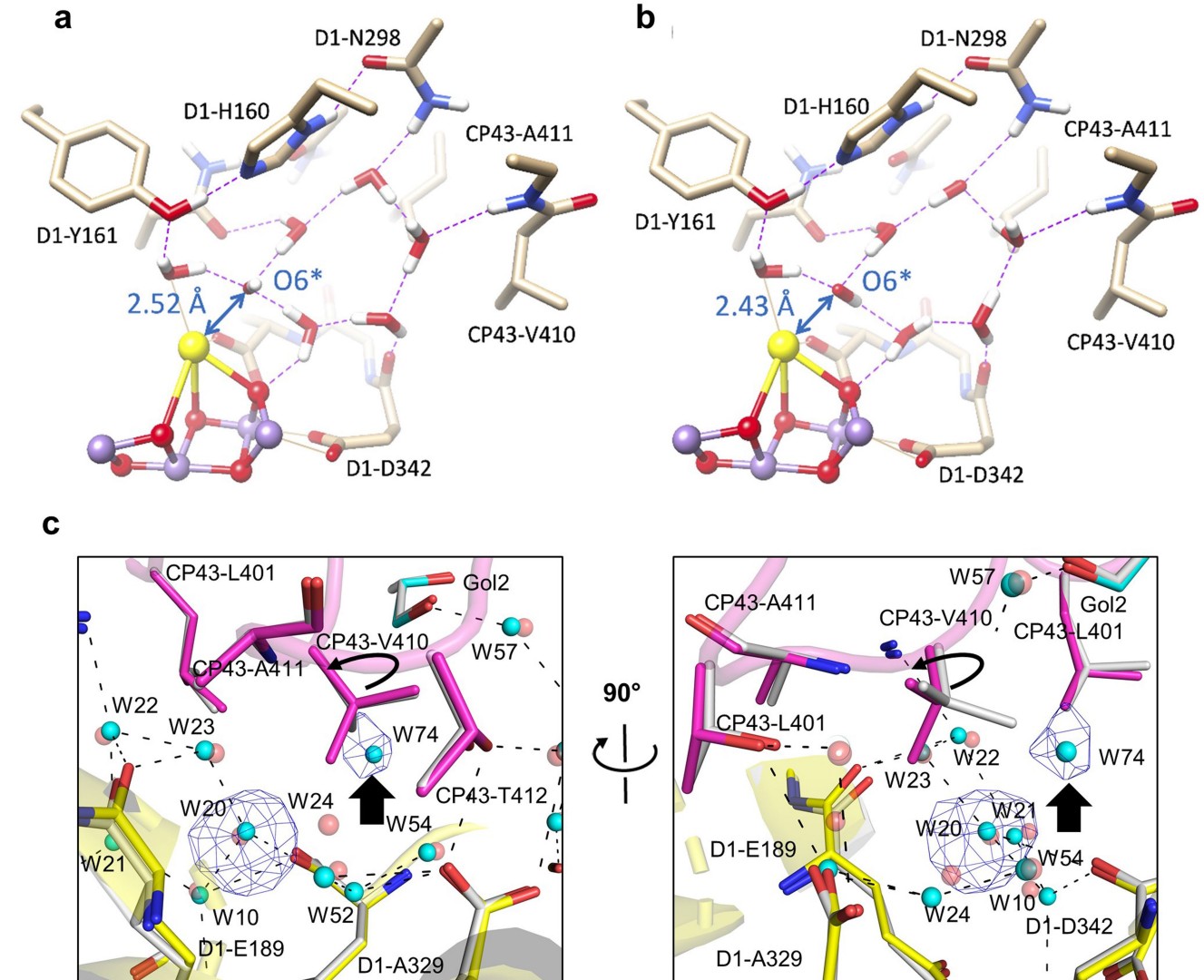

**a**

D1-N298

D1-H160

CP43-A411

D1-Y161

2.52 Å O6*

CP43-V410

D1-D342

**b**

D1-N298

D1-H160

D1-Y161

CP43-A411

2.43 Å O6*

CP43-V410

D1-D342

**c**

CP43-L401

Gol2

CP43-V410

W57

CP43-A411

W22

W23

W74

W20 W24

CP43-T412

W21

W54

W10 W52

D1-E189

D1-A329

**90°**

CP43-A411

W57

CP43-V410

Gol2

CP43-L401

CP43-L401

W22

W74

W23

W20 W21

W54

D1-E189

W24 W10

D1-D342

D1-A329

**Extended Data Fig. 5 | Two representative hydrogen-bonding arrangements modelled by DFT calculations, and the rotation of CP43-V410 and appearance of one new water molecule. a**,**b**, O6* is modelled as a hydroxide anion bound to Ca²⁺ in the S₂ state: one having a hydrogen bond between O6* and Glu189 (not depicted) (**a**), and the other one lacking this hydrogen bond (**b**). It is unstable if O6* is assumed to be a water molecule. Hydrogen atoms bonded to carbon atoms are omitted for clarity. **c**, Polder omit maps on W20 and W74 contoured at +3.5σ (blue mesh) superimposed with PSII structures of 1F state (grey sticks and red spheres) and 2F (5 ms) (magenta sticks for CP43 and yellow sticks for D1, as well as cyan spheres for water molecules).

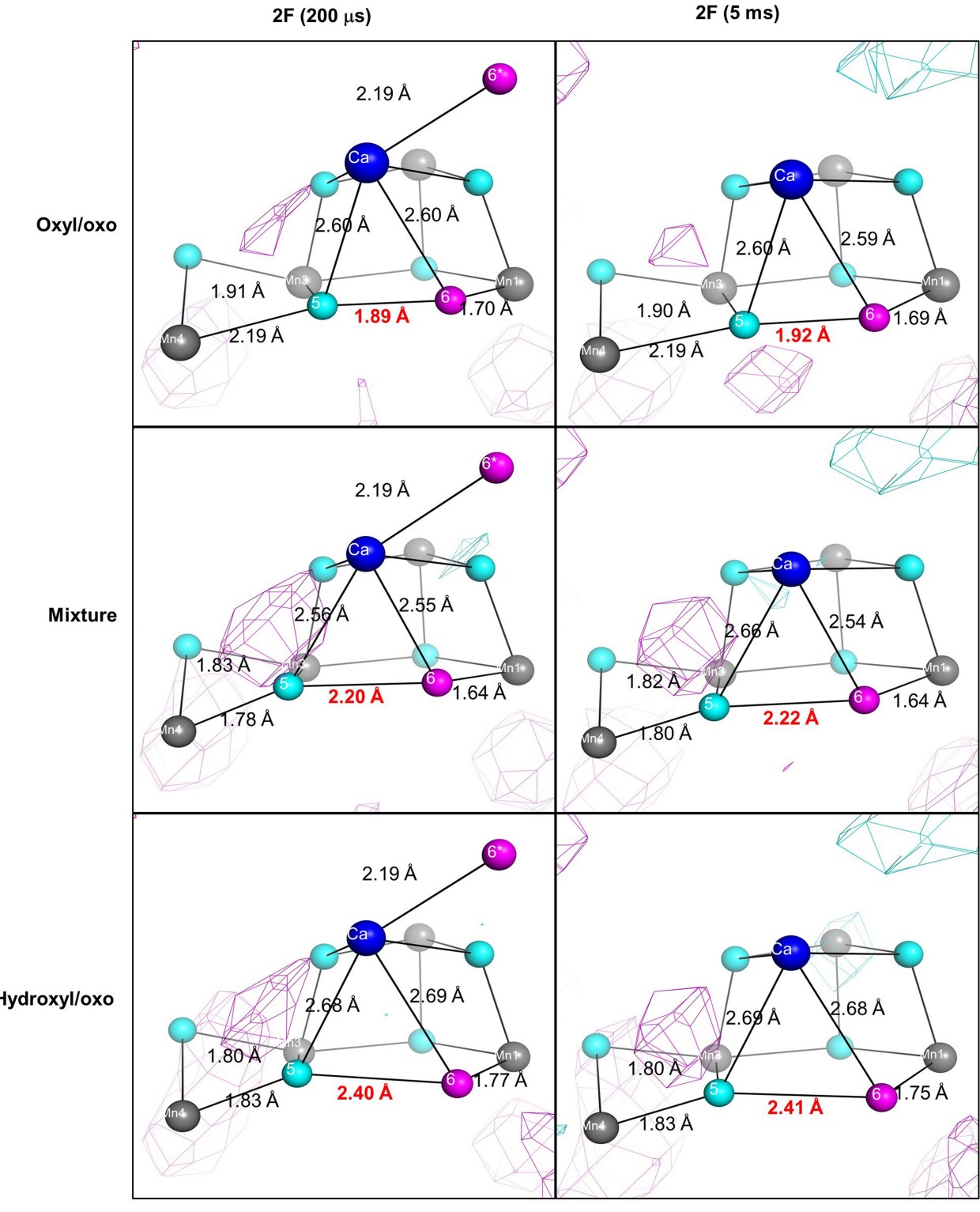

**2F (200 μs)** **2F (5 ms)**

**Oxyl/oxo**

**Mixture**

**Hydroxyl/oxo**

**Extended Data Fig. 6 | Examinations of the oxyl/oxo and hydroxyl/oxo species between O5 and O6.** The m$F_o$-D$F_c$ maps contoured at +2.5σ (cyan) and −2.5σ (magenta) superposed with OEC structures of 2F (200 μs) (left) and 2F (5 ms) (right). O5/O6 at distances of 1.9 Å (oxyl/oxo, upper side), 2.4 Å (hydroxyl/oxo, bottom) and 2.2 Å (mixture of the two coupling species, middle) are examined. On the basis of the residual densities, the distance of 1.9 Å fits best with the electron density at 200 μs, whereas it is difficult to distinguish between the distances of 1.9 Å–2.4 Å at 5 ms.

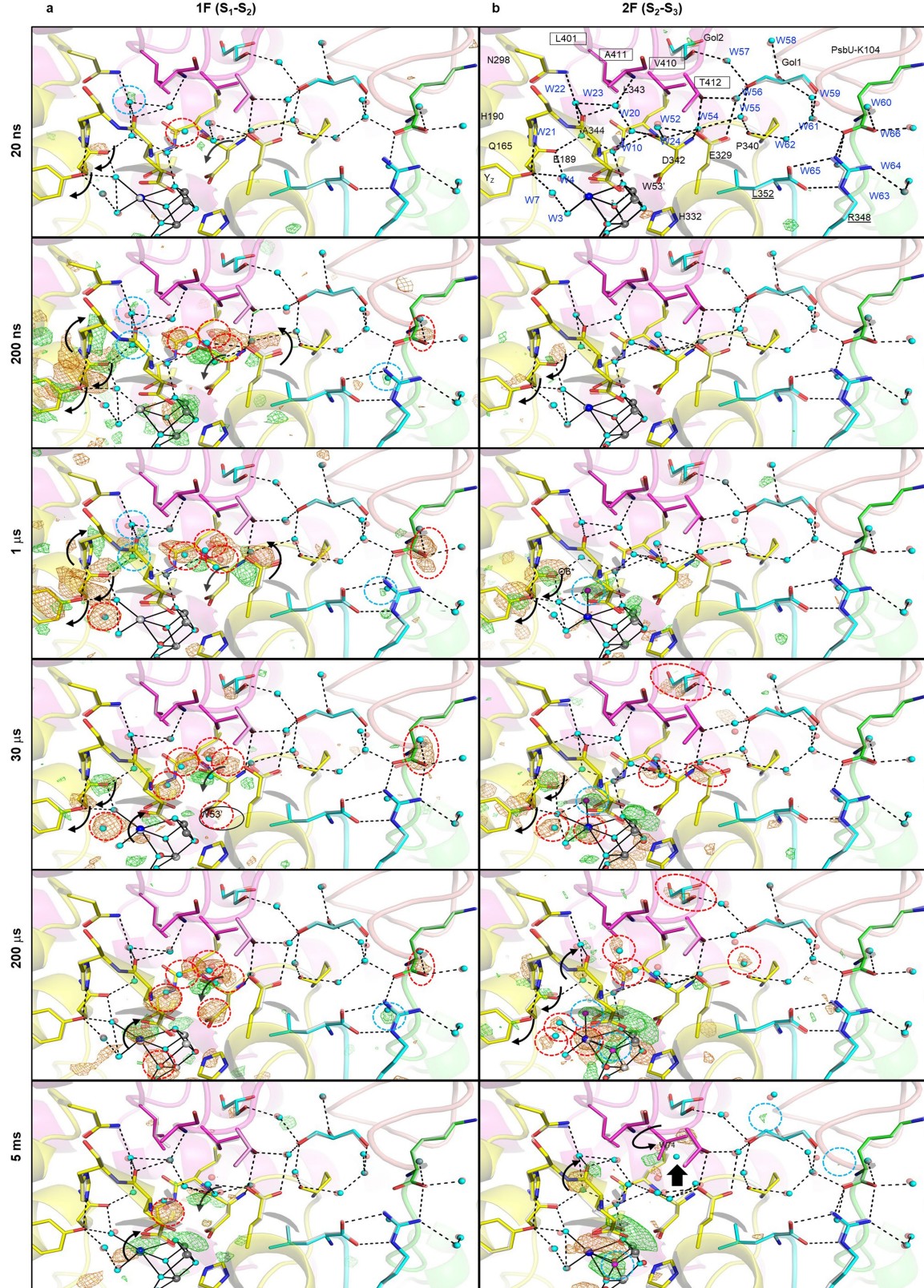

**Extended Data Fig. 7 | Structural dynamics at the O1 channel during S₁–S₂–S₃ transitions. a**,**b**, Structures of PSII at the O1 channel are superposed with $F_{obs}$(1F) − $F_{obs}$(dark) (**a**) and $F_{obs}$(2F) − $F_{obs}$(1F) (**b**) difference density maps contoured at +4.0$\sigma$ (green) and −4.0$\sigma$ (orange), with delay times from 20 ns to 5 ms. The intermediate structures of D1, D2, CP43, PsbU and PsbV proteins are depicted in yellow, cyan, magenta, green and pink, respectively. Water molecule W53 in the S₁ state is only observable under cryo-temperature conditions (PDB codes: 3WU2 and 4UB6) and is not detectable at room temperature (PDB codes: 5WS5 and 7CJI). Because the protein environment around W53 is the same regardless of temperature, W53 is considered present but fluctuating. Therefore, in this study, we denote this invisible water as W53′.

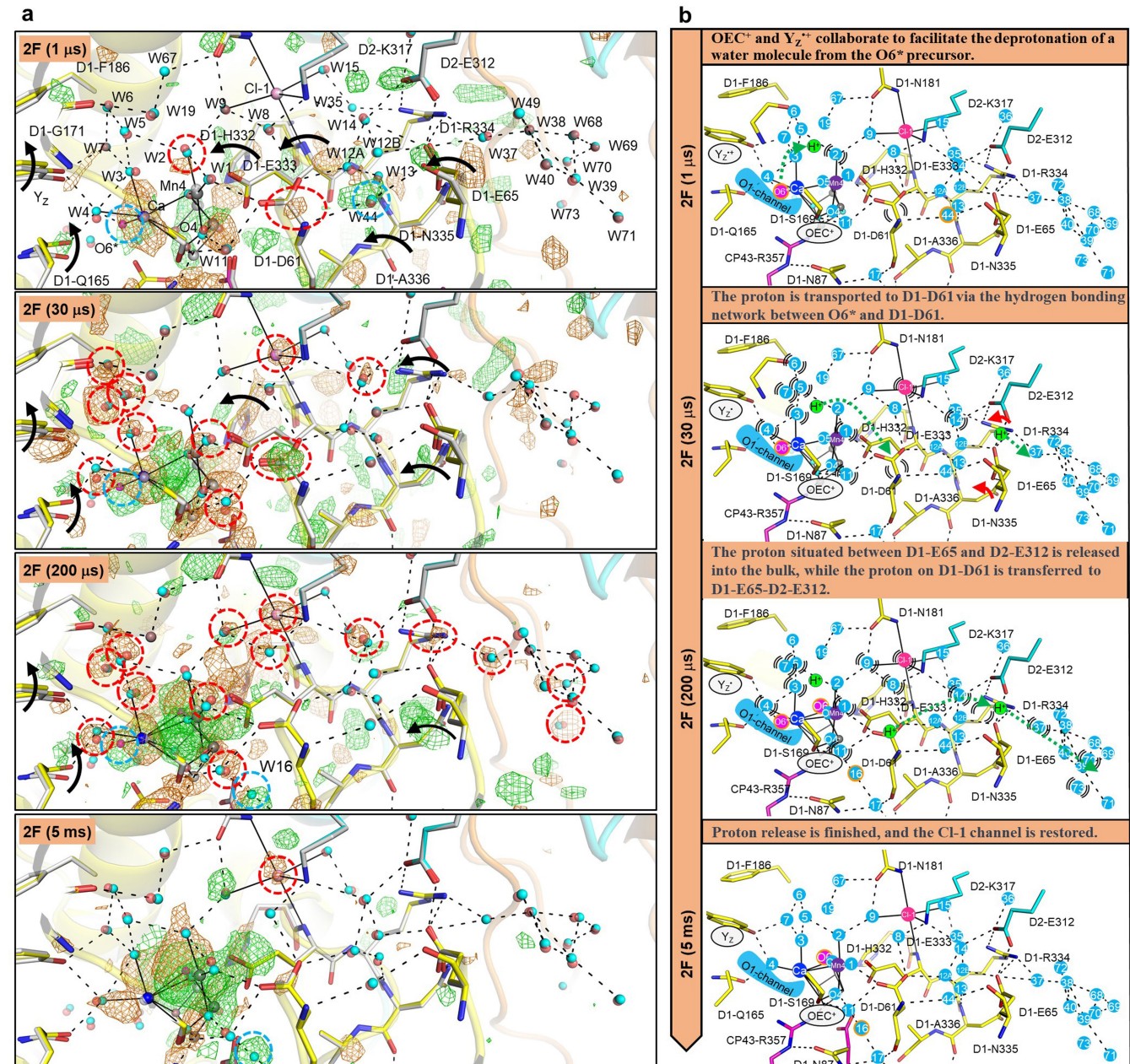

**Extended Data Fig. 8 | Structural changes at the Cl-1 channel and proton release pathways after 2F. a**, Structures of PSII at Cl-1 channels superposed with $F_{obs}(2F) - F_{obs}(1F)$ difference densities contoured at $+3.0\sigma$ (green) and $-3.0\sigma$ (orange), with delay times from 1 μs to 5 ms. The 1F model is depicted in grey, and intermediate structures of D1, D2, CP43 and PsbO proteins are depicted in yellow, cyan, magenta and orange, respectively. Water molecules at their ground states and intermediate states are depicted in red and cyan spheres, respectively. O6* and O6 are represented by magenta spheres. Black dotted lines represent hydrogen bonds. Black solid lines link the oxo-oxygen in the OEC and the ligand water to the metal ions, as well as ligand residues and water

molecules to Cl-1. Black arrows indicate structural changes. Disordered water and residues are encircled by red dotted lines, and ordered water molecules in the intermediate structures are encircled by cyan dotted lines. **b**, Possible proton release pathways after 2F. Water molecules are depicted as cyan spheres, with their corresponding numbers labelled. Disordered water molecules, Cl-1 and residues are depicted with arched lines, and an orange outer ring around water molecules indicates that they became ordered. The red arrows indicate the movements of residues. The green dotted arrows indicate the movements of protons.

# Extended Data Table 1 | Data processing and structure refinement statistics

| Data set | Dark | 1F (20 ns) | 1F (200 ns) | 1F (1 μs) | 1F (30 μs) | 1F (200 μs) | 1F (5 ms) | 1F | 2F (20 ns) | 2F (200 ns) | 2F (1 μs) | 2F (30 μs) | 2F (200 μs) | 2F (5 ms) |
|---|---|---|---|---|---|---|---|---|---|---|---|---|---|---|
| Wavelength (Å) | | | | 1.24 | | | | | | | 1.24 | | | |
| No. of collected images | 250,748 | 127,254 | 145,599 | 187,489 | 118,559 | 186,590 | 77,543 | 244,369 | 197,256 | 225,474 | 185,967 | 104,358 | 110,258 | 89,971 |
| No. of hit images | 105,314 | 62,354 | 65,520 | 61,871 | 54,537 | 65,307 | 32,568 | 69,520 | 66,006 | 70,063 | 72,171 | 41,691 | 30,754 | 44,134 |
| No. of indexed images | 61,207 | 43,511 | 48,580 | 42,595 | 41,261 | 46,061 | 24,670 | 55,155 | 50,078 | 56,728 | 59,038 | 33,962 | 23,851 | 34,176 |
| Space group | | | | $P2_12_12_1$ | | | | | | | $P2_12_12_1$ | | | |
| Cell dimension (Å) | | | | a = 125.75, b = 231.60, c = 288.28 | | | | | | | a = 125.77, b = 231.76, c = 288.58 | | | |
| Resolution (Å) | 47.45-2.15 (2.19-2.15)* | 47.45-2.20 (2.24-2.20) | 47.45-2.25 (2.29-2.25) | 47.45-2.25 (2.29-2.25) | 47.45-2.20 (2.24-2.20) | 47.45-2.20 (2.24-2.20) | 47.45-2.30 (2.34-2.30) | 47.45-2.25 (2.29-2.25) | 47.45-2.30 (2.34-2.30) | 47.45-2.25 (2.29-2.25) | 47.45-2.25 (2.29-2.25) | 47.45-2.30 (2.34-2.30) | 47.45-2.25 (2.29-2.25) | 47.45-2.25 (2.29-2.25) |
| Unique reflections | 453,468 (22,435) | 427,344 (21,214) | 399,706 (19,836) | 399,706 (19,836) | 427,344 (21,214) | 427,344 (21,214) | 374,348 (18,515) | 399,706 (19,836) | 374,348 (18,515) | 399,706 (19,836) | 399,706 (19,836) | 374,348 (18,515) | 399,706 (19,836) | 399,706 (19,836) |
| Multiplicity | 597.1 (231.1) | 404.4 (197.3) | 477.5 (280.0) | 381.7 (222.7) | 402.1 (195.0) | 401.2 (192.5) | 237.9 (153.7) | 450.8 (229.4) | 306.5 (189.4) | 514.4 (265.7) | 525.7 (270.9) | 301.3 (186.7) | 210.1 (107.8) | 304.3 (154.8) |
| Completeness (%) | 100.0 (100.0) | 100.0 (100.0) | 100.0 (100.0) | 100.0 (100.0) | 100.0 (100.0) | 100.0 (100.0) | 100.0 (100.0) | 100.0 (100.0) | 100.0 (100.0) | 100.0 (100.0) | 100.0 (100.0) | 100.0 (100.0) | 100.0 (100.0) | 100.0 (100.0) |
| Mean I/σ (I) | 75.2 (1.3) | 56.0 (1.1) | 59.9 (1.5) | 52.7 (1.3) | 58.6 (1.4) | 53.5 (1.2) | 48.5 (1.6) | 61.2 (1.5) | 49.7 (1.5) | 64.8 (1.6) | 63.6 (1.4) | 56.5 (1.7) | 50.6 (1.4) | 52.9 (1.5) |
| Wilson B-factor (Å²) | 62.3 | 48.7 | 48.8 | 49.2 | 63.6 | 48.4 | 65.6 | 62.4 | 64.9 | 62.9 | 63.9 | 64.3 | 63.7 | 63.0 |
| R-split (%)‡ | 5.2 (67.5) | 6.8 (75.9) | 6.4 (65.8) | 7.2 (69.6) | 7.3 (68.1) | 7.0 (74.9) | 9.1 (64.6) | 8.2 (67.1) | 9.4 (69.2) | 7.6 (66.6) | 7.1 (69.1) | 10.4 (67.7) | 10.8 (74.5) | 9.5 (72.5) |
| CC1/2 (%) | 99.7 (59.2) | 99.5 (47.9) | 99.6 (49.2) | 99.4 (53.0) | 99.4 (58.6) | 99.5 (49.2) | 99.1 (59.9) | 99.3 (52.6) | 99.1 (54.3) | 99.4 (56.0) | 99.5 (55.4) | 98.7 (53.9) | 98.6 (47.8) | 99.0 (51.6) |
| Structure refinement | | | | | | | | | | | | | | |
| Resolution range (Å) | 19.98-2.15 (2.23-2.15) | 19.98-2.20 (2.28-2.20) | 19.98-2.25 (2.33-2.25) | 19.98-2.25 (2.33-2.25) | 19.98-2.20 (2.28-2.20) | 19.98-2.20 (2.28-2.20) | 19.98-2.30 (2.38-2.30) | 19.99-2.25 (2.33-2.25) | 19.99-2.30 (2.38-2.30) | 19.99-2.25 (2.33-2.25) | 19.99-2.25 (2.33-2.25) | 19.99-2.30 (2.38-2.30) | 19.99-2.25 (2.33-2.25) | 19.99-2.25 (2.33-2.25) |
| R-work | 0.145 (0.334) | 0.143 (0.336) | 0.146 (0.320) | 0.146 (0.321) | 0.140 (0.312) | 0.142 (0.334) | 0.139 (0.291) | 0.145 (0.303) | 0.142 (0.305) | 0.139 (0.298) | 0.140 (0.313) | 0.142 (0.294) | 0.143 (0.303) | 0.142 (0.303) |
| R-free | 0.175 (0.361) | 0.179 (0.370) | 0.186 (0.345) | 0.180 (0.349) | 0.176 (0.349) | 0.178 (0.367) | 0.180 (0.326) | 0.180 (0.334) | 0.184 (0.340) | 0.177 (0.328) | 0.177 (0.342) | 0.181 (0.329) | 0.183 (0.339) | 0.182 (0.338) |
| Number of non-H atoms | 52978 | 62674 | 62674 | 62674 | 62674 | 62674 | 62674 | 52977 | 62600 | 62600 | 62602 | 62602 | 62605 | 62604 |
| Macromolecules | 41835 | 49244 | 49244 | 49244 | 49244 | 49244 | 49244 | 41835 | 49173 | 49173 | 49173 | 49173 | 49173 | 49173 |
| Ligands | 9317 | 11275 | 11275 | 11275 | 11275 | 11275 | 11275 | 9317 | 11275 | 11275 | 11275 | 11275 | 11277 | 11277 |
| Solvent | 1826 | 2155 | 2155 | 2155 | 2155 | 2155 | 2155 | 1825 | 2152 | 2152 | 2154 | 2154 | 2155 | 2154 |
| Protein residues | 5289 | 5289 | 5289 | 5289 | 5289 | 5289 | 5289 | 5289 | 5289 | 5289 | 5289 | 5289 | 5289 | 5289 |
| RMS (bonds) (Å) | 0.008 | 0.009 | 0.009 | 0.009 | 0.009 | 0.009 | 0.009 | 0.008 | 0.009 | 0.009 | 0.009 | 0.009 | 0.009 | 0.009 |
| RMS (angles) (degree) | 1.26 | 1.28 | 1.28 | 1.28 | 1.28 | 1.28 | 1.29 | 1.27 | 1.30 | 1.30 | 1.32 | 1.31 | 1.29 | 1.29 |
| Ramachandran favored (%) | 97.83 | 97.79 | 97.81 | 97.81 | 97.91 | 97.83 | 97.81 | 97.83 | 97.64 | 97.89 | 97.66 | 97.69 | 97.87 | 97.81 |
| Ramachandran allowed (%) | 1.98 | 2.04 | 2.02 | 2.02 | 1.92 | 2.00 | 2.00 | 2.00 | 2.17 | 1.94 | 2.19 | 2.15 | 1.96 | 2.02 |
| Ramachandran outliers (%) | 0.19 | 0.17 | 0.17 | 0.17 | 0.17 | 0.17 | 0.19 | 0.17 | 0.19 | 0.17 | 0.15 | 0.15 | 0.17 | 0.17 |
| Rotamer outliers (%) | 2.16 | 2.15 | 2.15 | 2.15 | 2.21 | 2.27 | 2.15 | 2.41 | 2.29 | 2.29 | 2.14 | 2.19 | 2.21 | 2.16 |
| Clashscore | 4.70 | 5.00 | 5.13 | 5.13 | 4.99 | 5.02 | 5.39 | 4.54 | 5.20 | 4.96 | 5.02 | 5.19 | 4.88 | 4.75 |
| Average B-factor (Å²) | 65.71 | 65.06 | 65.31 | 65.56 | 65.26 | 65.08 | 65.54 | 63.81 | 63.61 | 64.11 | 64.46 | 65.35 | 63.89 | 63.19 |
| Macromolecules | 63.94 | 62.96 | 63.24 | 63.46 | 63.13 | 62.94 | 63.47 | 62.63 | 61.88 | 62.35 | 62.62 | 63.59 | 62.10 | 61.43 |
| Ligands | 73.23 | 73.82 | 73.99 | 74.33 | 74.15 | 74.07 | 74.32 | 69.16 | 70.97 | 71.63 | 72.35 | 72.89 | 71.55 | 70.69 |
| Solvent | 67.77 | 67.08 | 67.09 | 67.56 | 67.45 | 66.95 | 67.00 | 63.66 | 64.54 | 65.13 | 65.26 | 66.10 | 64.75 | 63.91 |
| PDB code | 8IR5 | 8IR6 | 8IR7 | 8IR8 | 8IR9 | 8IRA | 8IRB | 8IRC | 8IRD | 8IRE | 8IRF | 8IRG | 8IRH | 8IRI |

*Values in parentheses indicate those for the highest-resolution shells. ‡$R_{split} = \sqrt{2}\,\Sigma|I_{even} - I_{odd}|/\Sigma\,(I_{even} + I_{odd})$.

**Extended Data Table 2 | Atomic distances at various time points after 1F or 2F**

| Atom-Atom distances | Monomer | Dark | 1F (20 ns) | 1F (200 ns) | 1F (1 μs) | 1F (30 μs) | 1F (200 μs) | 1F (5 ms) | 1F | 2F (20 ns) | 2F (200 ns) | 2F (1 μs) | 2F (30 μs) | 2F (200 μs) | 2F (5 ms) |
|---|---|---|---|---|---|---|---|---|---|---|---|---|---|---|---|
| Fe-BCT_O1 | A | 2.18 | 2.18 | 2.17 | 2.22 | 2.21 | 2.28 | 2.36 | 2.33 | 2.46 | 2.40 | 2.39 | 2.21 | 2.32 | 2.28 |
| | B | 2.13 | 2.10 | 2.17 | 2.22 | 2.28 | 2.41 | 2.49 | 2.28 | 2.37 | 2.34 | 2.20 | 2.19 | 2.39 | 2.27 |
| | Ave. | 2.16 | 2.14 | 2.17 | 2.22 | 2.25 | 2.35 | 2.43 | 2.31 | 2.42 | 2.37 | 2.30 | 2.20 | 2.36 | 2.28 |
| Fe-BCT_O2 | A | 2.13 | 2.24 | 2.40 | 2.28 | 2.42 | 2.35 | 2.29 | 2.34 | 2.17 | 2.32 | 2.42 | 2.44 | 2.22 | 2.28 |
| | B | 2.42 | 2.52 | 2.61 | 2.53 | 2.61 | 2.50 | 2.59 | 2.49 | 2.60 | 2.52 | 2.61 | 2.59 | 2.38 | 2.53 |
| | Ave. | 2.28 | 2.38 | 2.51 | 2.41 | 2.52 | 2.43 | 2.44 | 2.42 | 2.39 | 2.42 | 2.52 | 2.52 | 2.30 | 2.41 |
| D1-Y246-BCT_O1 | A | 3.15 | 3.17 | 3.04 | 3.01 | 2.98 | 3.02 | 2.69 | 2.90 | 2.91 | 3.02 | 3.15 | 3.23 | 3.14 | 2.98 |
| | B | 3.26 | 3.42 | 3.27 | 3.23 | 3.17 | 3.15 | 2.94 | 3.17 | 3.10 | 3.24 | 3.40 | 3.32 | 3.04 | 3.18 |
| | Ave. | 3.21 | 3.30 | 3.16 | 3.12 | 3.08 | 3.09 | 2.82 | 3.04 | 3.01 | 3.13 | 3.28 | 3.28 | 3.09 | 3.08 |
| Mn1-Mn3 | A | 3.19 | 3.21 | 3.24 | 3.20 | 3.20 | 3.17 | 3.22 | 3.22 | 3.27 | 3.17 | 3.16 | 3.18 | 3.36 | 3.53 |
| | B | 3.29 | 3.34 | 3.28 | 3.34 | 3.26 | 3.22 | 3.27 | 3.23 | 3.26 | 3.20 | 3.15 | 3.14 | 3.39 | 3.52 |
| | Ave. | 3.24 | 3.28 | 3.26 | 3.27 | 3.23 | 3.20 | 3.25 | 3.23 | 3.27 | 3.19 | 3.16 | 3.16 | 3.38 | 3.53 |
| Mn1-Mn4 | A | 4.84 | 4.84 | 4.90 | 4.89 | 4.92 | 4.93 | 4.92 | 4.97 | 5.02 | 5.02 | 5.00 | 5.03 | 5.22 | 5.33 |
| | B | 4.83 | 4.91 | 4.83 | 4.86 | 4.93 | 4.97 | 4.91 | 4.88 | 4.87 | 4.91 | 4.92 | 4.85 | 5.22 | 5.23 |
| | Ave. | 4.84 | 4.88 | 4.87 | 4.88 | 4.93 | 4.95 | 4.92 | 4.93 | 4.95 | 4.97 | 4.96 | 4.94 | 5.22 | 5.28 |
| Mn4-Ca | A | 3.83 | 3.79 | 3.83 | 3.82 | 3.99 | 4.10 | 4.07 | 4.05 | 4.01 | 4.06 | 4.11 | 4.12 | 4.12 | 4.09 |
| | B | 3.76 | 3.84 | 3.83 | 3.88 | 3.93 | 4.10 | 4.06 | 3.97 | 3.93 | 3.99 | 4.10 | 4.01 | 4.06 | 4.03 |
| | Ave. | 3.80 | 3.82 | 3.83 | 3.85 | 3.96 | 4.10 | 4.07 | 4.01 | 3.97 | 4.03 | 4.11 | 4.07 | 4.09 | 4.06 |
| Yz-D1-H190 | A | 2.50 | 2.54 | 2.82 | 2.85 | 2.61 | 2.51 | 2.47 | 2.52 | 2.53 | 2.57 | 2.63 | 2.70 | 2.58 | 2.58 |
| | B | 2.51 | 2.67 | 2.78 | 2.80 | 2.71 | 2.54 | 2.54 | 2.58 | 2.58 | 2.58 | 2.56 | 2.70 | 2.63 | 2.56 |
| | Ave. | 2.51 | 2.61 | 2.80 | 2.83 | 2.66 | 2.53 | 2.51 | 2.55 | 2.56 | 2.58 | 2.60 | 2.70 | 2.61 | 2.57 |
| Ca-D1-E189 | A | 2.98 | 3.03 | 3.05 | 3.04 | 2.94 | 2.88 | 2.85 | 2.89 | 2.90 | 2.91 | 2.92 | 2.99 | 3.09 | 3.10 |
| | B | 3.03 | 3.03 | 3.01 | 3.00 | 2.95 | 2.91 | 2.86 | 2.89 | 2.93 | 2.95 | 2.93 | 2.97 | 3.03 | 3.08 |
| | Ave. | 3.01 | 3.03 | 3.03 | 3.02 | 2.95 | 2.90 | 2.86 | 2.89 | 2.92 | 2.93 | 2.93 | 2.98 | 3.06 | 3.09 |
| D1-E65-D2-E312 | A | 2.53 | 2.64 | 2.61 | 2.59 | 2.61 | 2.56 | 2.62 | 2.56 | 2.68 | 2.59 | 2.66 | 2.58 | 2.60 | 2.66 |
| | B | 2.54 | 2.57 | 2.59 | 2.55 | 2.60 | 2.62 | 2.58 | 2.61 | 2.59 | 2.60 | 2.58 | 2.59 | 2.61 | 2.58 |
| | Ave. | 2.54 | 2.61 | 2.60 | 2.57 | 2.61 | 2.59 | 2.60 | 2.59 | 2.64 | 2.60 | 2.62 | 2.59 | 2.61 | 2.62 |
| D1-E65-D1-R334 | A | 3.00 | 3.05 | 3.10 | 2.96 | 3.09 | 3.10 | 3.11 | 2.99 | 3.07 | 3.00 | 2.99 | 2.93 | 3.16 | 3.06 |
| | B | 2.93 | 2.84 | 2.80 | 2.93 | 2.85 | 2.88 | 3.03 | 2.96 | 2.95 | 2.79 | 2.91 | 2.94 | 2.85 | 2.87 |
| | Ave. | 2.97 | 2.95 | 2.95 | 2.95 | 2.97 | 2.99 | 3.07 | 2.98 | 3.01 | 2.90 | 2.95 | 2.94 | 3.01 | 2.97 |
| D2-E312-D1-R334 | A | 3.06 | 3.06 | 2.96 | 2.91 | 2.96 | 2.98 | 3.05 | 2.99 | 2.97 | 2.91 | 2.94 | 3.07 | 2.90 | 3.02 |
| | B | 3.11 | 3.05 | 3.09 | 3.05 | 3.02 | 3.00 | 3.12 | 3.11 | 3.02 | 3.01 | 2.89 | 3.02 | 3.05 | 3.03 |
| | Ave. | 3.09 | 3.06 | 3.03 | 2.98 | 2.99 | 2.99 | 3.09 | 3.05 | 3.00 | 2.96 | 2.92 | 3.05 | 2.98 | 3.03 |

| | |
|---|---|

# Reporting Summary

## Statistics

For all statistical analyses, confirm that the following items are present in the figure legend, table legend, main text, or Methods section.

| n/a | Confirmed | |
|---|---|---|
| ☐ | ☒ | The exact sample size (*n*) for each experimental group/condition, given as a discrete number and unit of measurement |
| ☒ | ☐ | A statement on whether measurements were taken from distinct samples or whether the same sample was measured repeatedly |
| ☒ | ☐ | The statistical test(s) used AND whether they are one- or two-sided *Only common tests should be described solely by name; describe more complex techniques in the Methods section.* |
| ☒ | ☐ | A description of all covariates tested |
| ☒ | ☐ | A description of any assumptions or corrections, such as tests of normality and adjustment for multiple comparisons |
| ☐ | ☒ | A full description of the statistical parameters including central tendency (e.g. means) or other basic estimates (e.g. regression coefficient) AND variation (e.g. standard deviation) or associated estimates of uncertainty (e.g. confidence intervals) |
| ☒ | ☐ | For null hypothesis testing, the test statistic (e.g. *F*, *t*, *r*) with confidence intervals, effect sizes, degrees of freedom and *P* value noted *Give P values as exact values whenever suitable.* |
| ☒ | ☐ | For Bayesian analysis, information on the choice of priors and Markov chain Monte Carlo settings |
| ☒ | ☐ | For hierarchical and complex designs, identification of the appropriate level for tests and full reporting of outcomes |
| ☒ | ☐ | Estimates of effect sizes (e.g. Cohen's *d*, Pearson's *r*), indicating how they were calculated |

*Our web collection on statistics for biologists contains articles on many of the points above.*

## Software and code

Policy information about availability of computer code

| Data collection | Cheetah (https://github.com/keitaroyam/cheetah), CrystFEL (version: 0.6.3) |
|---|---|
| Data analysis | CrystFEL (version: 0.6.3), dials.still_process in DIALs (version: 1.16), Publicly available CCP4 modules (CCP4 7.1.015) including CCTBX, SFTOOLS, Truncate, CAD, SCALEIT, and FFT. Packages in phenix (version: 1.19.2-4158) including cxi.merge and Phaser (2.8.3). Coot (version: 0.9.8.8), Pymol (version: 2.5.4) |

For manuscripts utilizing custom algorithms or software that are central to the research but not yet described in published literature, software must be made available to editors and reviewers. We strongly encourage code deposition in a community repository (e.g. GitHub). See the Nature Portfolio guidelines for submitting code & software for further information.

## Data

Policy information about availability of data

All manuscripts must include a data availability statement. This statement should provide the following information, where applicable:
- Accession codes, unique identifiers, or web links for publicly available datasets
- A description of any restrictions on data availability
- For clinical datasets or third party data, please ensure that the statement adheres to our policy

The atomic coordinates and structure factors have been deposited in the Protein Data Bank under the following IDs: 8IR5 for 0F (dark, ground state for the Δt1 structures), 8IR6 for Δt1 = 20 ns, 8IR7 for Δt1 = 200 ns, 8IR8 for Δt1 = 1 μs, 8IR9 for Δt1 = 30 μs, 8IRA for Δt1 = 200 μs, 8IRB for Δt1 = 5 ms, 8IRC for 1F (ground state

for the Δt2 structures), 8IRD for Δt2 = 20 ns, 8IRE for Δt2 = 200 ns, 8IRF for Δt2 = 1 µs, 8IRG for Δt2 = 30 µs, 8IRH for Δt2 = 200 µs, and 8IRI for Δt2 = 5 ms.  All other data with a PDB code used in this study are adopted from the PDB databank.

## Research involving human participants, their data, or biological material

Policy information about studies with human participants or human data. See also policy information about sex, gender (identity/presentation), and sexual orientation and race, ethnicity and racism.

| | |
|---|---|
| Reporting on sex and gender | n/a |
| Reporting on race, ethnicity, or other socially relevant groupings | n/a |
| Population characteristics | n/a |
| Recruitment | n/a |
| Ethics oversight | n/a |

Note that full information on the approval of the study protocol must also be provided in the manuscript.

# Field-specific reporting

Please select the one below that is the best fit for your research. If you are not sure, read the appropriate sections before making your selection.

☒ Life sciences          ☐ Behavioural & social sciences          ☐ Ecological, evolutionary & environmental sciences

For a reference copy of the document with all sections, see nature.com/documents/nr-reporting-summary-flat.pdf

# Life sciences study design

All studies must disclose on these points even when the disclosure is negative.

| | |
|---|---|
| Sample size | The sample size is determined based on the number of diffraction images that produced reasonably high resolution and  multiplicity or number of unique diffractions in each  time-resolved serial crystallography data set. The microcrystals obtained were reproducible, and the diffraction data was processed with standard crystallographic software (Methods section), which resulted in standard data statistics as shown in Extended Data Table 1. |
| Data exclusions | No data was excluded; for details of the data analysis statistics, see Extended Data Table 1. |
| Replication | Thousands of microcrystals were used to obtain a full diffraction dataset, which showed that the results are well reproduced. The repetition rate of a specific dataset was shown in Extended Data Table 1, in which, it was shown that the repetition rate is over 100 even for the highest resolution shell for each of the dataset. |
| Randomization | The microcrystals used for the diffraction experiments were not chosen, and diffraction data from good microcrystals were used for the structural analysis. The criterion for choosing good diffraction data is based on the resolution as well as their deviations from standard cell parameters, as described in the Methods section. |
| Blinding | Not applicable. All experimental maps were included in the statistical analysis. |

# Reporting for specific materials, systems and methods

We require information from authors about some types of materials, experimental systems and methods used in many studies. Here, indicate whether each material, system or method listed is relevant to your study. If you are not sure if a list item applies to your research, read the appropriate section before selecting a response.

## Materials & experimental systems

| n/a | Involved in the study |
|-----|----------------------|
| ☒ ☐ | Antibodies |
| ☒ ☐ | Eukaryotic cell lines |
| ☒ ☐ | Palaeontology and archaeology |
| ☒ ☐ | Animals and other organisms |
| ☒ ☐ | Clinical data |
| ☒ ☐ | Dual use research of concern |
| ☒ ☐ | Plants |

## Methods

| n/a | Involved in the study |
|-----|----------------------|
| ☒ ☐ | ChIP-seq |
| ☒ ☐ | Flow cytometry |
| ☒ ☐ | MRI-based neuroimaging |

## Plants

| Seed stocks | n/a |
|-------------|-----|

| Novel plant genotypes | n/a |
|-----------------------|-----|

| Authentication | n/a |
|----------------|-----|

