## [Peer Review File · Nature]

Manuscript Title: Oxygen-evolving photosystem II structures during S1-S2-S3 transitions

Reviewer Comments & Author Rebuttals

Reviewer Reports on the Initial Version:

Referees' comments:

Referee #1 (Remarks to the Author):

The manuscript entitled "Structural dynamics of oxygen-evolving photosystem II during S1-S2-S3 transitions" by Professor J. R. Shen and colleagues reported 14 new XFEL structures of the dark-adapted photosystem II (PSII) microcrystals, or after a single flash (1F), or after two flashes (2F) of illumination at time intervals of 20 ns, 200 ns, 1 μ s, 30 μ s, 200 μ s, and 5 ms at resolution of 2.15 to 2.30 Å. They applied the classic isomorphous Fo-Fo difference Fourier analysis to these data sets, which is a highly suitable method for revealing subtle structural changes associated with light-dependent processes within the oxygen-evolving center (OEC). The experiments and analysis described in the paper were solid and the manuscript was well written, making this complex system understandable to a general reader. Every figure and tables in the main text and supporting information is necessary, important, and clear. This is an excellent example how crystallographic structures of reaction intermediates should be presented. Nevertheless, there are a number of important issues that need to be addressed before this manuscript can be accepted for publication. Although the data were solid, alternative interpretations could explain some of their data better than the ones provided.

First of all, I suggest replacing the title S1-S2-S3 transitions with the title 0F-1F-2F transitions, which is a more accurate description of their results because the authors provided no direct crystallographic evidence to show that the dominant form of the 0F PSII was indeed S1, nor whether it advances according to the Kok cycle. I believe that the simplest interpretation for crystallographic data is the most likely to be the correct one according to the Occam Razor principle. If possible, one must avoid imposing any spectroscopically inferred and theoretical derived metal ion cluster models for crystallographic interpretations. Proper interpretation of crystallographic data should not be reliant on spectroscopic data or theoretical OEC models. It should be emphasized that Pantazis and colleagues noted that the atomic models of the T. vulcanus PSII intermediates did not agree with their theoretical models (Drosou et al., 2023). This is an important finding to me because it clearly shows that the spectroscopically inferred and theoretical OEC models are incorrect because there is every reason to trust that the crystal structures are representative of active forms of PSII. When both crystallographic and spectroscopic data are INDEPENDENTLY interpreted properly, they must agree with each other.

The second point is in regard to the missing water molecule W16 in the O4-connected channel associated with increased peaks near Mn4 of the OEC during the 0F to 1F transition. In monomer A (Extended Table 4), this water molecule has an outstanding negative peak of -12.5σ in the 1F(30 μ s)-0F difference map, -17.0σ in the 1F(200 μ s)-0F map, and finally -14.3σ in the 1F(5 ms)-0F map. For monomer B, the corresponding peaks are -11.5σ , -15.8σ , and -12.3σ , respectively. For both monomers, the second time point corresponds to the most negative peak. My alternative interpretation is that water molecule W16 is a substrate water molecule that goes into the OEC during the 0F to 1F transition, occurring at 200 μ s as opposed to hypothesis of Shen and colleagues that W16 diffuses into bulk solvent through the O4-connected channel. After the 200 μ s time point, other water molecules in the channel gradually leak in to refill this vacated site or perhaps some fraction of water molecules from the OEC region may return to it. Evidence that supports the notion that water molecule W16 goes into the OEC is the following: the positive peak between Mn4, W1, and W2 of the OEC is $+12.3\sigma$. I propose that this is the destination of the

entered water molecule, consistent with the fact that the negative peak next to Mn4 was only -6.6 σ with the net gain of +5.7 σ at this site in monomer A at 200 μ s. The magnitude of these peaks decreases slightly at 5 ms to +9.9 σ and -3.6 σ with the remaining net gain of still +6.6 σ . This observation is consistent with the possibility that both W1 and W2 were only partially occupied in 0F and then the vacated ligand positions were refilled to higher occupancies in the 1F state. In fact, Mn4 is likely Mn(III) at 0F, therefore may have only 5 ligands on average. I propose that these 5 O ligands may be distributed among the six ligand-binding sites of which two protein ligands are invariant (D1-D170 and D1-E333) and remaining W1, W2, O5, and O4 positions share three O ligands. In this scheme, after the 0F to 1F transition, one water substrate fills the vacancy so that Mn4 has 6 ligands.

In the 2F-1F difference Fourier maps at 200 μ s and 5 ms, O5 of the OEC in monomer A was flanked by negative peaks of -6.7 σ and -8.2 σ on the Mn4 side and by positive peaks of 7.8 σ and 9.0 σ on the Mn1 side, currently assigned at O6 by Shen and colleagues. I would interpret this as features corresponding a displacement of O5 in a direction from Mn4 to Mn1, as suggested from results of single-conformer model refinement (Wang et al., 2021a). When O5 is displaced, Mn1 becomes 6-coordinate but Mn4 is 5-coordinate. Redox inactive Ca1 ion of the OEC has only negative peaks of -8.6 σ and -4.7 σ because its corresponding positive peaks were the currently being assigned to the hypothetical O6 position. By combining both sets of the observed differences, I conclude that the hypothetical O6 ligand does not exist.

During the 1F to 2F transition because there was no major loss of any water molecule nearby the OEC, nor was there loss of W16 in the O4-connected channel. This is consistent with no water addition to the OEC taking place during the 1F to 2F transition. The change of electron number in the so-called O6* position can be interpreted differently, having nothing to do with fact that O6 may not exist. The reason that this is only indirect but not direct evidence is that no missing water molecule in channels does not necessarily mean that there must have no movement of water molecules in the channel connected to the OEC. When a water molecule moves toward the OEC the next water in the channel may simply move up to take its place.

My above interpretation is clearly simpler than that of Shen and colleagues, but it does not exclude theirs. A difficulty I have with their interpretation is that it is not clear why W16 moves out only at 200 μ s to bulk solvent through the O4-connected channel after 1F but then moves back in partially at 1 ms afterwards and did not continue to do so afterwards. This reasoning was invoked in order to explain the pattern of increase, maximum, and then decrease in peak size as described above. My interpretation does not have this problem. I think their interpretation also has a weakness as to answering why the redox inactive Ca²⁺ ion of the OEC has only negative peaks associated with it because its corresponding positive peak overlapped with the hypothetical O6 peak. Correlation between O6 and O6* also appears very weak (their Extended Table 4), a point which is further discussed below.

Their extended Table 4 shows peak sizes in 6 pairs of 1F-0F and 2F-1F difference Fourier maps each for displacements of redox inactive Cl1A anion (monomer A) whose positive and negative peaks should have the same size. The mean positive value for the 12 pairs of Cl1A anion is 4.7 \pm 1.0 σ , and the mean negative value is -3.6 \pm 1.0 σ , which provides an error bar of 1.1 σ for the redox inactive Cl⁻ anion. Likewise, the mean positive value for 12 pairs of the D1-D342 sidechain of monomer A is 4.1 \pm 2.2 σ , and the mean negative value is -3.3 \pm 2.2 σ , which provides an error bar of 0.8 σ . The mean positive value for 12 pairs of the D1-E329 sidechain of monomer A is 3.9 \pm 1.0 σ , and the mean negative value is -3.5 \pm 1.2 σ , which provides an error bar of 0.4 σ . When such comparison is made between two monomers, an error bar is also about the same magnitude, i.e., it is indicative the two monomers in these structures being indeed highly synchronized. This property differs from many previous dark-adapted PSII structures analyzed (Wang et al., 2021b). Therefore, the net gain of electrons in units of +5.7 σ to +6.6 σ next to Mn4 of the OEC in two monomers are significant given the estimated error. If one considers +17.0 σ to be 8 electrons of one water O atom at W16(A), the +5.7 σ to +6.6 σ correspond to 2.7 and 3.1 electrons or about 33% to 39% of one water molecule, respectively. An addition of 8 electrons of one O ligand distributed among W1, W2, O4, and O5 is about 2 electrons each and is about 3.3 electrons among W1, W2, and O5 each (or 33% each).

Thirdly, I also have different interpretations for the O6* peaks as follows. For example, Kamiya and colleagues have shown that the two OECs of two monomers of PSII dimer have different structures (Tanaka et al., 2017). Monomer B of their 5b5e structure has an outstanding positive peak of $+6.4\sigma$ (which was the 13th highest unexplained peak in its Fo-Fc residual difference) near O1 and Ca1 of the OEC, and the D1-A344 sidechain (also distant to D1-E189) in a position similar to the O6* position of the OEC of monomer A of this study (as a proposed precursor for O6 in this study by Shen and colleagues). This peak is not present in monomer A. This residual peak was also present only in monomer B of their 5b66 structure although it was $\sim 4.0\sigma$ in monomer A (smaller but still significant, however was again not present). There are many other differences between two monomers of these structures according to Kamiya and colleagues. Likewise, Shen and colleagues earlier also showed that monomer B of their 4ub6 and 4ub8 structures had the same peak but was interpreted as a third water molecule bound to Ca1 of the OEC (Suga et al., 2015). This peak was not present in monomer A. That is, Ca1 of the OEC in monomer A had two terminal water molecule ligands whereas Ca1 of the OEC in monomer B had three water molecule ligands.

In more recent studies of Shen and colleagues (Suga et al., 2019), this peak was still present, again only in monomer B of their 6jll structure but not in monomer A even though this peak remained unexplained in the atomic models. Again, the replicated 6jlm and 6jll structures underwent very different light-dependent photochemical transitions but did not undergo the same structural transitions between the two monomers and between the two replicates. Moreover, Wang and colleagues showed that two monomers of dark-adapted PSII dimers had different metal ion compositions in the OEC binding pocket (Wang et al., 2021b). Not only did the two monomers of PSII differ, but also individual dimers of dark-adapted PSII structures had different metal ion compositions inside the OEC pocket.

Shen and colleagues reported that there appeared positive difference features first on Mn4 and then all four Mn ions and one Ca ion of the OEC, respectively, during 20 ns to 200 ns transition. Their interpretation was that this unusual feature had something to do with the change of electrostatic potential associated with the YZ dot/YZ pair, which remains very unclear and puzzling to this reviewer. Typically, positive difference features could be due to gains of electrons (such as addition of new O ligands in small fraction and distributed among multiple sites) or reduced atomic motions. In the latter case, halos of negative features of much smaller amplitudes would surround the positive peaks, which may remain invisible.

Fourthly, I like the amination presented by Shen colleagues because they help visualize structural changes very clearly. It is clear that the loss of W16 in the O4-connected channel occurred during the 0F to 1F transition, but not during the 1F to 2F transition. The authors used these observations to rule out a so-called carousel mechanism for substrate water molecule entry to the OEC in their paper. The authors propose an alternative water-substrate entry mechanism from the O1-connected channel via O6*. As discussed above, this O6* water molecule had a serious problem of its own. More importantly, Wang and colleagues showed that single-conformer of the OEC without inclusion of any sixth O ligand added to the OEC could fully explain the difference Fourier maps between the 2F and 1F data sets of both cyanobacterial species (Wang et al., 2021a). Their analysis implies that it is not crystallographically justifiable to arbitrarily add a second conformer containing the hypothetical O6-added OEC model, especially given the fact that a single-conformer model refinement without an O6-containing model could fully explain all the observed difference Fourier features.

Lastly, I acknowledge that Shen and colleagues have done excellent crystallographic work in this study and in the past, often in replicates, and continue to improve the degree of isomorphism between the data sets. With such small overall fractional amplitude differences of the observed structure factors in the current data sets (of only 7 to 11% at about 2.30 Å resolution, Extended Table 3), very subtle structural differences could indeed be revealed. Some of these differences may correspond to about one electron or less. It is important not to overinterpret the excellent experimental results either by the authors or readers. A clarification statement is needed to warn the reader not to overinterpret their crystallographic data. I also recommend to include full details

regarding the duration of time lapse between PSII sample preparation and data collection because the oxygen evolution activity of their crystals was not reported for their study and because the activity loss has been reported to be about 20% every three weeks (Suiura and Inoue, 1999). It is always important to include one or two sentences to explain how these new XFEL data sets differed from their own previous works already published.

References

- Drosou, M., Comas-Vila, G., Neese, F., Salvador, P., and Pantazis, D.A. (2023). Does Serial Femtosecond Crystallography Depict State-Specific Catalytic Intermediates of the Oxygen-Evolving Complex? *J Am Chem Soc* 145, 10604-10621.
- Suga, M., Akita, F., Hirata, K., Ueno, G., Murakami, H., Nakajima, Y., Shimizu, T., Yamashita, K., Yamamoto, M., Ago, H., et al. (2015). Native structure of photosystem II at 1.95 Å resolution viewed by femtosecond X-ray pulses. *Nature* 517, 99-103.
- Suga, M., Akita, F., Yamashita, K., Nakajima, Y., Ueno, G., Li, H., Yamane, T., Hirata, K., Umena, Y., Yonekura, S., et al. (2019). An oxyl/oxo mechanism for oxygen-oxygen coupling in PSII revealed by an x-ray free-electron laser. *Science* 366, 334-338.
- Sugiura, M., and Inoue, Y. (1999). Highly purified thermo-stable oxygen-evolving photosystem II core complex from the thermophilic cyanobacterium *Synechococcus elongatus* having His-tagged CP43. *Plant Cell Physiol* 40, 1219-1231.
- Tanaka, A., Fukushima, Y., and Kamiya, N. (2017). Two Different Structures of the Oxygen-Evolving Complex in the Same Polypeptide Frameworks of Photosystem II. *J Am Chem Soc* 139, 1718-1721.
- Wang, J., Armstrong, W.H., and Batista, V.S. (2021a). Do crystallographic XFEL data support binding of a water molecule to the oxygen-evolving complex of photosystem II exposed to two flashes of light? *Proc Natl Acad Sci U S A* 118.
- Wang, J., Gisriel, C.J., Reiss, K., Huang, H.L., Armstrong, W.H., Brudvig, G.W., and Batista, V.S. (2021b). Heterogeneous Composition of Oxygen-Evolving Complexes in Crystal Structures of Dark-Adapted Photosystem II. *Biochemistry* 60, 3374-3384.

Referee #2 (Remarks to the Author):

The authors have employed serial femtosecond crystallography (SFX) to characterize structural changes that accompany the S1 to S2 and S2 to S3 transitions in the O₂-forming reaction of Photosystem II. Importantly, the authors report structural models at various time points during each transition, specifically, at 20 ns, 200 ns, 1 μs, 20 μs, 200 μs, and 5 ms after the saturating flash that initiates each desired transition. Understanding the nature and sequence of the structural events during the individual S state transitions is necessary for understanding the mechanism of O₂ formation. This is especially true for the S2 to S3 transition because pulsed EPR data show that a water molecule (and possible substrate) deprotonates and relocates to a Mn ion during this transition. Determining the identity of this water molecule and how it reaches the Mn₄CaO₅ cluster from the lumen is necessary for differentiating between and refining theoretical models for O-O bond formation. This study is another impressive contribution from the Shen and Suga labs. It is the first study to report the nature and sequence of events during the S1 to S2 transition and provides much higher time resolution than a previous study of intermediate structures during the S2 to S3 transition (refs. 26 and 15). The data are exciting, well-described, and will be of interest to anyone interested in O₂ production in nature or artificial systems. However, before the study is published. The authors should address two points:

1. Given the authors' estimations of Si state populations in parallel crystalline samples with FTIR show that their 1-Flash samples contain 40% S1 and only 60% S2, and that their 2-Flash samples contain 49% S2 and only 51% S3 (bottom of page 28), the authors should describe the methods

they employed to estimate the contributions of the substantial fractions of PSII that did not advance to the desired S state because of misses (i.e., 40% of PSII in the 1F samples and 49% of PSII in the 2F samples).

2. The authors should address the criticisms raised in Wang et al. PNAS (2021) 118 e202392118 (<https://www.pnas.org/doi/full/10.1073/pnas.202392118>) regarding assigning electron density near the Mn ions of the Mn₄CaO₅ cluster, particularly to O6.

I applaud the authors for including Extended Data Table 1, showing the correspondence of their numbering of water molecules in the O1, O4, and Cl1 channels with the numbering in several PDB entries. I strongly urge the authors to add an additional column showing the correspondence of their numbering with that adopted by the Berkeley group in their published figures (e.g., those in refs 15, 25, and 26). Non-crystallographers would greatly appreciate it.

Finally, I noticed a few typographical errors:

1. Page 18, Line 401, CP43-V410
2. In Extended Data Table 2, the asterisk is missing from the CC1/2 data.
3. Supplementary Materials, line 124, "In the S2 state, THE following changes occur..."
4. Supplementary Materials, line 225, D1-D61 (not E61).

Referee #3 (Remarks to the Author):

Review of the manuscript entitled: "Structural Dynamics of oxygen-evolving Photosystem II during S1-S2-S3 Transition by Li et al.

A. Summary of the key results

This manuscript is very exciting as it reports on the first sub-microsecond dynamic studies on the mechanism of water splitting on Photosystem II by serial fs crystallography at an XFEL for the S1 to S2 and S2 to S3 S-state transition. While previous studies were limited to the us and millisecond time frame and mainly focused on the S2 to S3 transition, this work is of extremely high importance as it reveals with high time and spatial resolution the sequence of events that happen both at the QA/QB quinone acceptor site of PSII as well as at the donor site P680, the oxygen-evolving cluster (OEC) and the water and proton channels that bring the substrate waters to the OEC and mediate the proton release to the bulk. The authors have collected high-resolution SFX data sets at RT at the XFEL at SACLA for both the S1 to S2 transition and the S2 to S3 transition at time delays of 20ns, 200ns, 1us, 30us, 200us, and 5ms. 100um crystals were delivered to the XFEL beam in a viscous jet at RT and they were excited by a split laser beam from two sides "on the fly" prior to the 10fs X-ray exposure. This paper is based on 14 excellent time-resolved SFX data sets (7 for the S1-S2 transition including the dark data set and 7 for the S2 to S3 transition including the 1-flash S2 5ms data set that serves as the "baseline" for the calculation of the difference density maps for the S2 to S3 transition. The referee was very impressed and satisfied with the high quality and multiplicity of each of the data sets, where each data set consist of 30 000 to 66 0000 hits leading to 24 000 to 59 000 indexed images per data set. For all data sets the overall multiplicity is >200 and even in the highest resolution shell exceeds >100, at resolutions between 2.15 and 2.3 Å resolution. This far higher multiplicity of the data sets and the higher time resolution in the ns time range compared to previous studies on the S2 to S3 transition has allowed them to detect for the first time not only large conformational changes (like the appearance of O6) but for the first time reveal all the more subtle conformational changes that allowed them to detect the movement and flexibilities of the water networks, the movement of the Mn and the Ca in the metal cluster as well as conformational changes of the amino acids.

B. Originality and significance: If not novel, please include a reference

This paper and the results reported are highly novel. As summarized above the paper reports on

the first time-resolved studies with sub-us time resolution for the S1 to S2 and S2 to S3 transition of the oxygen-evolving complex in Photosystem II. Taking the extremely high relevance of the unraveling of the mechanism of water splitting into account, which provides all the oxygen for respiration and thereby all high life on Earth, this work is extremely important not only for the experts in Photosynthesis but also for the general audience. It has also a high significance for the development of clean energy in the future as the knowledge of the mechanism of water splitting in Nature can form the basis for the development of artificial systems that combine the efficiency of Nature with the stability of man-made systems.

C and D. Data & methodology: validity of approach, quality of data, quality of presentation

The TR-SFX studies presented in this work are state-of-the-art in the field. Reaching time resolutions nanoseconds brings time-resolved pump-probe SFX of Photosystem II into a new time regime and the large difference density changes observed at the 20 and 200ns time points show that multiple critical steps in the mechanism of water splitting are revealed in this study for the first time. The data quality is excellent. As outlined in the summary above the quality of the data sets collected with extremely high multiplicities is very critical for the quality of the difference density maps. In contrast to standard crystallography, all reflections are partials and Monte Carlo Integration over a large data set of these partial reflections is very important for the determination of accurate structure factors. The reviewer is thereby fully satisfied with the data, the methodology, and the data validation approach.

This study thereby by far exceeds previous studies on the S2 to S3 transition (Ibrahim 2020), where the fastest time point was 50us and which had significantly lower data redundancy thereby could detect only a few conformational changes compared to what has now been observed by Suga, Shen and their teams.

The large number of changes in the conformation that are detected in this study extend to multiple regions of Photosystem II including the acceptor site where the two quinones QA and QB are located, the metal cluster of the OEC and its bridging oxygen atoms, the water molecules in the vicinity of the OEC as well as the protein surrounding including side chain and backbone atoms. Furthermore, large changes are observed in the organization of the 3 water and proton channels connecting to the OEC.

The detection of all these changes that are observed at the 6 different time points is a real breakthrough but it also leads to serious difficulties in the presentation of the very comprehensive data. The main figures 2, 3, 5,6, and 7 consist of 12 panels featuring the difference density maps for each of the 6-time points for the S1 to S2 and S2 to S3 transition for the different regions of interest in PSII. Fig 2 shows the dynamics for the QA/QB acceptor site, Fig 3 shows the structural dynamics for the Yz area, Fig 5 features the dynamics of the OEC and its surroundings, Fig 6 shows the dynamics of the O1 channel and Fig 7 shows the dynamic of the O4 and Cl channels. This makes it very difficult for the reader to comprehend these figures and all the dynamic changes that are observed. The supplementary videos are really nice but also difficult to comprehend as so many changes happen at the same time. The problem of representation is most severe for Figures 2, 6, and 7 which show larger sections of the structure and make it very difficult to even read the labels of the amino acids and also prevents the reader from detecting the difference between the structural model of the ground state (dark S1 state for the S1-S2 transition) and (S2 state for the S2 to S3 transition). Here it would be helpful if a larger panel on top could be included that shows the dark structure with amino acids and water molecules labeled and then labelling of amino acids is avoided in the individual panels allowing for structural changes to be more clearly marked in the individual panels (they are currently only featured by small black arrows, which could be featured more prominently).

E. Conclusions: robustness, validity, reliability

The conclusions drawn in the paper are very well validated by the data and thereby reliable, however, the comprehensive conclusion of the order of events in the S1-S2-S3 transition is missing in the main text, which just finished with one very general conclusion sentence and no figure is provided in the main text that summarizes the exciting results.

However, when the reviewer was reading the supplementary discussion and looked at the

extended data file, the picture of events was very nicely summarized in the supplementary discussion and furthermore, extended data figures 7 and 8 provided a very nice comprehensive overview of the major events in the water channels in Figure 7 and at the electron donor site (including the OEC) in Figure 8. The reviewer therefore suggests that the supplementary discussion should be included as the conclusion in the main text and that at least Figure 8 should become the final figure of the main text. The reviewer is aware of the fact that Nature has a figure and word limit for the main body of the paper but due to the very complex Nature and the importance of the conclusion for the comprehension of the results by the general audience the reviewer hopes that the word and figure limit could be extended for this very important paper.

F. Suggested improvements: experiments, data for possible revision

While the reviewer was very satisfied with the level of detail provided for the X-ray data collection and data evaluation the reviewer was very disappointed by the lack of information on protein expression, purification, crystallization, crystal dehydration as well as embedding of the crystals into the viscous media. The reviewer therefore suggests MAJOR REVISIONS of this section of the paper in the materials and method section. The authors cite that the protein isolation and crystallization have been done as described previously with minor modifications but looking at all the previous publications none of them contains a detailed comprehensive description starting from cell growth to protein isolation, crystallization, and preparation of the crystals for sample delivery.

Growth conditions: Detailed information on the growth conditions including media light conditions, spectra of the cell's growth curves, and OD at the time of harvest should be included. This is essential as PSII undergoes constant reconstruction due to photodamage and the expression pattern changes of the three genes exist for the PsbA (D1) depending on the growth conditions this information is absolutely essential. It should also be indicated if the cells were WT cells of *TS vulcanus* or if a mutant lacking some of the PsbA genes has been grown for these studies.

Cell disruption and PSII isolation: Details including info on cell disruption, membrane isolation, protein detergent solubilization, and all purification steps must be included in the paper including info on all buffers and detergents used and times and temperature conditions for all protein isolation steps. Detergents of high purity are critical for the protein quality and also the quality of the crystals so please also include the info on the supplier of the detergents and their purity.

Crystallization of PSII: Details of the PSII crystallization must be included including all crystallization steps with protein concentration, buffers, detergents, incubation times (seeding conditions if used) additives etc. The crystals undergo a dehydration procedure before sample delivery which is critical for reaching high-resolution diffraction but currently, no information on this process is provided in this paper or previous publications. The reviewer therefore requests to include all details (including incubation steps and times) of this dehydration procedure and indicate how long the crystals are stable in the dehydration buffer. It would be also nice if a figure of the crystals used for the experiments could be included in the extended data file.

Embedding of crystals in viscous media:

PSII crystals are very sensitive and it is therefore critical that mechanical damage and stress are avoided in the embedding procedure of the crystals in the viscous media for sample delivery. The reviewer requests that details of the embedding procedure be provided.

G. References: appropriate credit to previous work?

All important relevant previous publications are cited in the paper.

H. Clarity and context: lucidity of abstract/summary, appropriateness of abstract, introduction, and conclusions.

The abstract/ summary represents a very good summary of the major findings and the introduction is also very appropriate summarizing the current status of the knowledge and the open questions on the function of Photosystem II and the mechanism of water splitting. As outlined above the paper is missing a major conclusion section in the main text as outline above and the reviewer suggests including therefore part of the supplementary discussion as the conclusion in the main text.

Minor comments:

Text on the QA-Fe-QB complex (last paragraph on Page 7): Please mention in the text at which time point the individual changes in the electron density map are visible.

Page 8, second paragraph: Please give an explanation for the shortcut BCT

Page 8 end of second paragraph: change the wording of "...BCT or reduction of partial QB by QA..." to: ...BCT or partial reduction of QB by QA...

Page 8 last sentence: The authors state that the structural changes on the QA binding site and the non-heme iron that occurred during this electron transfer are identical to those following the first flash. However, as stated in the next sentence the difference densities are weaker. This is an understatement as the difference densities are very strong for S1-S2 and very weak in this region for S2 to S3, so the reviewer can not understand how they can be identical when there are such large differences in the intensities. Please revise this paragraph.

Page 9 end of last paragraph: Please explain how the iron would promote electron transfer by becoming disordered.

Page 10 first sentence: The authors note that a positive difference density appears on the Mg atom of PD1. This is a very interesting finding as PD1 very likely represents the chlorophyll where the unpaired electron in P680+ λ is located that then accepts an electron from the Mn cluster via TryZ to return to the neutral P680 state. Does this mean that they have detected the re-reduction of P680+ λ to P680? If so that would be extremely exciting and should be highlighted more prominently in the main text.

Page 11 last paragraph: Here difference densities of Mn 4 are discussed which is later extended through the metal cluster while the metal-metal distances stay the same. The authors state that these differences may be caused by a charge rearrangement of the OEC triggered by the electrostatic effect of TryZ λ +/TyrZ λ . This is also a very important finding, could the authors include QMM calculation that model these charge rearrangements?

Page 17 first paragraph: It is not clear to the reviewer how a negative difference density could arise for W53' when W53' is not visible in the electron density in the first place as stated earlier.

Page 18: The authors mention Go11, however, the reviewer could not find it in Figure 7A

Methods page 26: The authors have used the SFX data evaluation program CrystFEL during the beamtime to observe and analyze the diffraction images and then used cctbx.xfel later for the indexing and integration of the diffraction images as well as for merging reflections after the beamtime. This is very interesting and it would be important to indicate why they switched from CrystFEL to cctbx. xfel in the later stages of the data analysis. It would be great if they could include a summary of the features in cctbx. xfel that they used. For example, was the program feature that allows for reconstruction of the peak profiles of the partial reflections in cctbx. xfel superior to the corresponding program element in CrystFEL? It would be nice if a comparison of the data quality obtained with the two programs could be included.

Methods page 28: last paragraph, the authors indicate that they determined the populations of the Si states in the crystals based on FTIR measurements for which they cite 3 references but no data. It is therefore not clear if they did the FTIR experiments under the same conditions and laser excitation as in the jet. It would be important to include the results of the FTIR experiments in the extended data and provide a description how they were done in the methods section.

Methods Page 30: the authors describe DFT calculations here but the results of the DFT calculations are not shown or discussed in detail in the main text.

Figures: The main request for revisions of the Figures of the main text has been discussed above. In addition, the reviewer wants to suggest a change in the color scheme for the water molecules in Figure 1. They are currently labeled with yellow numbers on an orange ball and this "tone in tone" labeling makes it very difficult to read the numbers. Please use a different color scheme here.

Author Rebuttals to Initial Comments:

Responses made in response to the comments by Referee #1

Referee #1 (Remarks to the Author):

The manuscript entitled "Structural dynamics of oxygen-evolving photosystem II during S1-S2-S3 transitions" by Professor J. R. Shen and colleagues reported 14 new XFEL structures of the dark-adapted photosystem II (PSII) microcrystals, or after a single flash (1F), or after two flashes (2F) of illumination at time intervals of 20 ns, 200 ns, 1 μ s, 30 μ s, 200 μ s, and 5 ms at resolution of 2.15 to 2.30 Å. They applied the classic isomorphous Fo-Fo difference Fourier analysis to these data sets, which is a highly suitable method for revealing subtle structural changes associated with light-dependent processes within the oxygen-evolving center (OEC). The experiments and analysis described in the paper were solid and the manuscript was well written, making this complex system understandable to a general reader. Every figure and tables in the main text and supporting information is necessary, important, and clear. This is an excellent example how crystallographic structures of reaction intermediates should be presented. Nevertheless, there are a number of important issues that need to be addressed before this manuscript can be accepted for publication. Although the data were solid, alternative interpretations could explain some of their data better than the ones provided.

(Response)

We thank the reviewer for his/her highly positive and encouraging comments and valuable suggestions to improve our manuscript. The reviewer's concerns about the data interpretations are important for us, and we have carefully examined these comments in revising our manuscript. In the following, we provide our point-by-point responses to the reviewer's comments.

First of all, I suggest replacing the title S1-S2-S3 transitions with the title 0F-1F-2F transitions, which is a more accurate description of their results because the authors provided no direct crystallographic evidence to show that the dominant form of the 0F PSII was indeed S1, nor whether it advances according to the Kok cycle. I believe that the simplest interpretation for crystallographic data is the most likely to be the correct one according to the Occam Razor principle. If possible, one must avoid imposing any spectroscopically inferred and theoretical derived metal ion cluster models for crystallographic interpretations. Proper interpretation of crystallographic data should not be reliant on spectroscopic data or theoretical OEC models. It should be emphasized that Pantazis and colleagues noted that the atomic models of the T. vulcanus PSII intermediates did not agree with their theoretical models (Drosou et al., 2023). This is an important finding to me because it clearly shows that the spectroscopically inferred and theoretical OEC models are incorrect because there is every reason to trust that the crystal structures are representative of active forms of PSII. When both crystallographic and spectroscopic data are INDEPENDENTLY interpreted properly, they must agree with each other.

(Response)

We appreciate the insightful suggestions, and have modified the title to "Structural dynamics of oxygen-evolving photosystem II during 0F-1F-2F transitions", according to the reviewer's suggestions. We agree with the reviewer's comment that "crystallographic data and spectroscopic data are independently interpreted properly", and have interpreted our data based

on our crystallographic studies as much as possible. We have noticed the studies by Pantazis and colleagues showing the inconsistency between the atomic models of the T. vulcanus PSII intermediates with their theoretical models (Drosou et al., 2023). In our opinion, both crystallographic data and theoretical studies have drawbacks in that, in the crystallographic studies there are errors in the inter-atomic distances, which depends on the resolutions. At current resolutions, these errors cannot be ignored when performing theoretical calculations based on the crystallographic structures. On the other hand, theoretical studies have to choose a limited set of atoms around the Mn₄CaO₅ cluster, and there may be uncertainties regarding the effects of ligand field on the geometric as well as electronic structures. These discrepancies may become smaller when higher resolutions of crystallographic structures are available and theoretical calculations may take the ligand effects into consideration more completely.

The second point is in regard to the missing water molecule W16 in the O4-connected channel associated with increased peaks near Mn4 of the OEC during the 0F to 1F transition. In monomer A (Extended Table 4), this water molecule has an outstanding negative peak of -12.5 σ in the 1F(30 μ s)-0F difference map, -17.0 σ in the 1F(200 μ s)-0F map, and finally -14.3 σ in the 1F(5 ms)-0F map. For monomer B, the corresponding peaks are -11.5 σ , -15.8 σ , and -12.3 σ , respectively. For both monomers, the second time point corresponds to the most negative peak. My alternative interpretation is that water molecule W16 is a substrate water molecule that goes into the OEC during the 0F to 1F transition, occurring at 200 μ s as opposed to hypothesis of Shen and colleagues that W16 diffuses into bulk solvent through the O4-connected channel. After the 200 μ s time point, other water molecules in the channel gradually leak in to refill this vacated site or perhaps some fraction of water molecules from the OEC region may return to it. Evidence that supports the notion that water molecule W16 goes into the OEC is the following: the positive peak between Mn4, W1, and W2 of the OEC is +12.3 σ . I propose that this is the destination of the entered water molecule, consistent with the fact that the negative peak next to Mn4 was only -6.6 σ with the net gain of +5.7 σ at this site in monomer A at 200 μ s. The magnitude of these peaks decreases slightly at 5 ms to +9.9 σ and -3.6 σ with the remaining net gain of still +6.6 σ . This observation is consistent with the possibility that both W1 and W2 were only partially occupied in 0F and then the vacated ligand positions were refilled to higher occupancies in the 1F state. In fact, Mn4 is likely Mn(III) at 0F, therefore may have only 5 ligands on average. I propose that these 5 O ligands may be distributed among the six ligand-binding sites of which two protein ligands are invariant (D1-D170 and D1-E333) and remaining W1, W2, O5, and O4 positions share three O ligands. In this scheme, after the 0F to 1F transition, one water substrate fills the vacancy so that Mn4 has 6 ligands.

(Response)

Although we cannot completely rule out the possibility of W16 being one of the substrate water molecules and going into the OEC during the 0F to 1F transition, we consider that this possibility is rather unlikely. The difference density in the proximity of Mn4 remains relatively small before $\Delta t_1 = 30 \mu$ s, but it becomes notable at 200 μ s and 5 ms (Fig. 1 below), following the Mn4's oxidation. As the positive density of Mn4 is correlated with a negative density, we consider that these densities reflect the movement of Mn4 toward the O4 side, although the pair of positive and negative densities surrounding Mn4 seems to be asymmetric. In contrast, the behavior of the negative difference density on W16 is dissimilar to that of Mn4 (Fig. 1 below), and it becomes significantly greater than those near Mn4 (Fig. 1 below) already at $\Delta t_1 = 1 \mu$ s and 30 μ s. These

results rule out the possibility of a direct correlation between the negative difference peak on W16 and the positive peak near Mn4.

We can also rule out the hypothesis based on the B-factors of W1, and W2. If W16 occupies the positions of W1 and W2 at $\Delta t=200 \mu\text{s}$, that would lead to a reduction in the B-factor of W1 and W2. However, no such reductions were observed at any time after the 1F (Table 1 below).

Therefore, we would like to keep the interpretation unchanged in the revised manuscript, and we consider that the disorder of W16 is likely influenced by the changes occurring around the OEC, rather than a substrate (see below).

Fig. 1 | $F_{obs}(1F) - F_{obs}(\text{dark})$ difference density maps contoured at $+4.0 \sigma$ (green) and -4.0σ (orange) with delay time from 20 ns to 5 ms.

Table 1 | Temperature factor (\AA^2) of W1 and W2 in the 1F experiment.

	dark	1F (20 ns)	1F (200 ns)	1F (1 μs)	1F (30 μs)	1F (200 μs)	1F (5 ms)
W1	45.55	45.78	47.55	48.52	44.60	45.76	46.55
W2	42.01	41.59	43.41	42.33	43.42	43.25	45.50

In the 2F-1F difference Fourier maps at 200 μs and 5 ms, O5 of the OEC in monomer A was flanked by negative peaks of -6.7σ and -8.2σ on the Mn4 side and by positive peaks of 7.8σ and 9.0σ on the Mn1 side, currently assigned at O6 by Shen and colleagues. I would interpret this as

features corresponding a displacement of O5 in a direction from Mn4 to Mn1, as suggested from results of single-conformer model refinement (Wang et al., 2021a). When O5 is displaced, Mn1 becomes 6-coordinate but Mn4 is 5-coordinate. Redox inactive Ca1 ion of the OEC has only negative peaks of -8.6σ and -4.7σ because its corresponding positive peaks were the currently being assigned to the hypothetical O6 position. By combining both sets of the observed differences, I conclude that the hypothetical O6 ligand does not exist.

(Response)

Wang et al. (2021a) employed XFEL data acquired initially by M. Ibrahim et al. (2020) to calculate the 2F-1F and 2F-Dark difference Fourier maps. The analysis revealed a positive difference density peak near O6 (Ox in M. Ibrahim et al.'s work), located at distances approximately 1.1 Å from O6 and 1.6 Å from O5. Wang et al. (2021a) postulated a discrepancy between the positions of O6 and this positive difference density. Considering the presence of negative difference density adjacent to O5 in the vicinity of Mn4, they proposed that the displacement of O5 might have contributed to the observed positive density near O6, rather than the insertion of a new water molecule. Subsequent refinements of the 2F data using a 1F single-conformation model, which lacks O6 and O5 moves toward Mn1 by 0.5 Å, yielded noise-level mF_o-DF_c signals on OEC. This observation illustrated that the 2F data (Ibrahim et al., 2020) can be adequately explained without invoking the insertion of O6.

However, we hold a different perspective regarding the insertion of O6 into the OEC after 2F. In the current study, the $F_o^{2F(5ms)}-F_o^{1F}$ difference *Fourier* map contoured at $\pm 7.0\sigma$, reveals that O6 is not located at the center of the positive density (with a peak of $+9\sigma$), but rather at the periphery of this approximate spherical positive density (Fig. 2a below). The distances from this positive peak to O5 and O6 are 1.9 Å and 0.4 Å, respectively (Fig. 2b below), suggesting that the positive difference density is more likely derived from the presence of O6. We can explain why O6 does not coincide perfectly with the center of the positive difference density as below. As the reviewer pointed out, there is a negative difference density on the opposing side of the positive difference density with respect to Ca (with a peak of -4.7σ) (Fig. 2a below). This pair of positive and negative difference densities can be interpreted as the inward movement of Ca toward the OEC and the positive peak near O6 may arise from a combination of O6 insertion and Ca movement.

To validate the location of O5 and O6, we employed Polder omit maps, wherein either O6 (as depicted in Fig. 2c below, left panel) or both O5 and O6 in the 2F structure (as shown in Fig. 2c, right panel) were omitted. The resultant maps clearly revealed that O6 was located at the peak's center. Moreover, the elliptical shape of the peak when both O5 and O6 were omitted strengthens the idea that two atoms O5 and O6 occupy their positions respectively, instead of a sole O5 atom in the 2F (5 ms) data.

Furthermore, we reproduced electron densities well when we attempted the methodology employed by Wang et al. (2021a) in which the OEC consists of a single O5 model. Residual positive F_o-F_c peak appeared near the O6 site, with a peak height of 3.45σ (Fig. 2d below). Because our analysis in which OEC contains both O5 and O6 atoms did not show such residual densities, we consider that both our interpretation and crystallographic analysis are valid. If we assume that OEC consists of the single O5 model, distances from the O5 atom to Mn1, Mn3, and Mn4 would become 2.65 Å, 2.04 Å, and 2.48 Å based on the present structure (it is 2.63, 2.25, 2.54 Å in Wang et al. 2021a), respectively. These distances are too long and would suggest that the O5 has no longer covalent bonds at least with both Mn1 and Mn4 atoms. This would disagree

with the Mn valances of this structure predicted by a number of other approaches including spectroscopic measurements. The structure is also inconsistent with general chemistry, even if it fits with the density map.

In summary, we consider that the insertion of O6 into OEC following 2F is the most appropriate scenario based on the analysis of $F_o^{2F(5ms)}-F_o^{1F}$ difference Fourier maps and the Polder omit maps.

Fig. 2 | Investigation of O6 insertion into OEC at delay time of 5 ms after the second flash. (a) and (b) shows the model of 2F (5 ms) superposed with the $F_o^{2F(5ms)}-F_o^{1F}$ difference Fourier maps. Map code and contour level are indicated in the figures. (c) Polder omit maps, with O6 omitted (left panel) and both O6 and the B conformation of O5 omitted (right panel). (d) Refinement of the OEC consists of a single O5 against 2F data. 2Fo-Fc map contoured at 1.0 σ is shown in blue mesh and Fo-Fc map contoured at $\pm 3.0 \sigma$ is shown in green and red mesh. The residual densities

near O5 and Mn1 is colored in green and has a peak height of $+3.45 \sigma$. Mn4-O5 distance was refined to be 2.48 Å, and Mn1-O5 distance was at 2.65 Å.

During the 1F to 2F transition because there was no major loss of any water molecule nearby the OEC, nor was there loss of W16 in the O4-connected channel. This is consistent with no water addition to the OEC taking place during the 1F to 2F transition. The change of electron number in the so-called O6* position can be interpreted differently, having nothing to do with fact that O6 may not exist. The reason that this is only indirect but not direct evidence is that no missing water molecule in channels does not necessarily mean that there must have no movement of water molecules in the channel connected to the OEC. When a water molecule moves toward the OEC the next water in the channel may simply move up to take its place.

My above interpretation is clearly simpler than that of Shen and colleagues, but it does not exclude theirs.

(Response)

As the reviewer pointed out, during the 1F to 2F transition, there was no major loss of water molecule nearby OEC. Since water molecules of the O1-channel are well connected in a hydrogen-bond network, the position of a water molecule that is taken up by OEC can be replaced by a neighboring water molecule immediately, so the loss of a water molecule is not necessarily observed. The prerequisite of being a substrate water is, we think, the high fluidity of water molecules so that they are quickly incorporated into the catalyst. In this respect, the following observations support our interpretation. First, W10 in the channel disorders during $\Delta t1 = 30 \mu\text{s} - 5 \text{ms}$ to increase its fluidity (Fig. 1 of this response). Second, O6* next to W10, which is probably a deprotonated W10, emerges at $\Delta t2 = 1 \mu\text{s}$. Third, the disappearance of O6* and emergence of O6 at $\Delta t2 = 1 \mu\text{s} - 5 \text{ms}$ are temporally synchronized. Forth, when the translocation of O6* to O6 completes, a new water molecule W74 near CP43-V410 appears (Main text Fig. 4 and Extended Data Fig. 7). All these results suggest that O6 is inserted into the position near O5, and O6* serves as a precursor for O6.

A difficulty I have with their interpretation is that it is not clear why W16 moves out only at 200 μs to bulk solvent through the O4-connected channel after 1F but then moves back in partially at 1 ms afterwards and did not continue to do so afterwards. This reasoning was invoked in order to explain the pattern of increase, maximum, and then decrease in peak size as described above. My interpretation does not have this problem.

(Response)

As we proposed in the manuscript, W16 becomes disordered but may remain in the vicinity of its original position and does not necessarily move out to bulk solvent, and its fluidity is likely affected by the charge of OEC that fluctuates during the S-state transition. At $\Delta t1 = 200 \mu\text{s}$, following the oxidation of Mn4, the OEC remains unstable, as evidenced by the negative difference density on O5, which is likely due to the presence of the positive charge on OEC. By $\Delta t1 = 5 \text{ms}$, OEC becomes relatively stable, as indicated by the disappearance of the negative difference density on O5 and the concurrent decrease in the difference density near Mn4. Such observation is reasonable if we consider that catalysts often become transiently less stable in their intermediate states in order to catalyze chemical reactions.

I think their interpretation also has a weakness as to answering why the redox inactive Ca^{2+} ion of the OEC has only negative peaks associated with it because its corresponding positive peak overlapped with the hypothetical O6 peak.

(Response)

As we mentioned in the response right before Figure 2, the positive difference density near Ca indeed overlaps with the O6 signal, which leads to a discrepancy between the positions of O6 and the combined positive difference density.

Correlation between O6 and O6* also appears very weak (their Extended Table 4), a point which is further discussed below.

Their extended Table 4 shows peak sizes in 6 pairs of 1F-0F and 2F-1F difference Fourier maps each for displacements of redox inactive Cl1A anion (monomer A) whose positive and negative peaks should have the same size. The mean positive value for the 12 pairs of Cl1A anion is $4.7 \pm 1.0\sigma$, and the mean negative value is $-3.6 \pm 1.0\sigma$, which provides an error bar of 1.1σ for the redox inactive Cl⁻ anion. Likewise, the mean positive value for 12 pairs of the D1-D342 sidechain of monomer A is $4.1 \pm 2.2\sigma$, and the mean negative value is $-3.3 \pm 2.2\sigma$, which provides an error bar of 0.8σ . The mean positive value for 12 pairs of the D1-E329 sidechain of monomer A is $3.9 \pm 1.0\sigma$, and the mean negative value is $-3.5 \pm 1.2\sigma$, which provides an error bar of 0.4σ . When such comparison is made between two monomers, an error bar is also about the same magnitude, i.e., it is indicative the two monomers in these structures being indeed highly synchronized. This property differs from many previous dark-adapted PSII structures analyzed (Wang et al., 2021b). Therefore, the net gain of electrons in units of $+5.7\sigma$ to $+6.6\sigma$ next to Mn4 of the OEC in two monomers are significant given the estimated error. If one considers $+17.0\sigma$ to be 8 electrons of one water O atom at W16(A), the $+5.7\sigma$ to $+6.6\sigma$ correspond to 2.7 and 3.1 electrons or about 33% to 39% of one water molecule, respectively. An addition of 8 electrons of one O ligand distributed among W1, W2, O4, and O5 is about 2 electrons each and is about 3.3 electrons among W1, W2, and O5 each (or 33% each).

(Response)

First of all, the mean values that the reviewer calculated are somewhat different from what we calculated from our Extended Data Table 2. As shown in Table 2 below, the mean values for the 12 pairs of positive and negative peaks around Cl-1A are $+4.2 \pm 1.4\sigma$ and $-3.8 \pm 1.3\sigma$, and the mean values for the 12 pairs of positive and negative peaks around D1-E329 are $+3.3 \pm 2.3\sigma$ and $-3.0 \pm 2.4\sigma$. The differences between the positive and negative values around these residues are smaller than the reviewer calculated. Only the mean values for the 12 pairs of positive and negative peaks around D1-D342 are the same as the reviewer pointed out, namely, $+4.1 \pm 2.3\sigma$ and $-3.3 \pm 2.3\sigma$. We do not know why the values are different between our calculation and the reviewer's calculation regarding the peaks around Cl-1A and D1-E329.

Table 2 | Mean values and standard deviations of some pairs of positive and negative difference densities. The mean values that match with the Reviewer's one are highlighted in yellow.

CI-1-channel	1F (20 ns)	1F (200 ns)	1F (1 μ s)	1F (30 μ s)	1F (200 μ s)	1F (5 ms)	2F (20 ns)	2F (200 ns)	2F (1 μ s)	2F (30 μ s)	2F (200 μ s)	2F (5 ms)	Mean value	Standard deviation
CI-1 (-) (A)	-3.5	-3.6	-2.4	-4.8	-4.7	-3	-3.3	-2.9	-2.4	-5.1	-6.8	-3.4	-3.8	1.3
CI-1 (+) (A)	5	7.2	4.3	4.1	5.9	4.3	2.2	2.8	2.9	3	4.8	3.9	4.2	1.4
CI-1 (-) (B)	-2.7	-4.6	-3.1	-4	-4.5	-2.6	-2.6	0	0	-3.5	-5.5	-3.8	-3.1	1.7
CI-1 (+) (B)	3.4	4.7	3.8	4	5.3	4.1	3.5	3	3.6	3.1	3	3	3.7	0.7
D1-D342 (-) (A)	-4	-6	-5.1	-5.2	-6.6	-4	0	-3.1	0	0	-2.2	-2.8	-3.3	2.3
D1-D342 (+) (A)	4.3	6.6	5.5	6.2	7.1	5.6	2.9	3.7	0	0	3	3.9	4.1	2.3
D1-D342 (-) (B)	-5.1	-4.1	-6.5	-5	-6.9	-5.4	-2.1	0	0	0	0	0	-2.9	2.8
D1-D342 (+) (B)	6.7	7.3	7.5	7.6	7.5	7.1	2.6	2.4	2.6	2.9	0	0	4.5	3.0
D1-E329 (-) (A)	-3.6	-4.9	-6.4	-4	-2.4	-3.3	-3.5	-2	-3	-3.7	-2.4	3.8	-3.0	2.4
D1-E329 (+) (A)	2.9	4.9	6	4.5	2.5	2.7	2.5	3.9	4.7	3.8	4.5	-3	3.3	2.3
D1-E329 (-) (B)	-2.4	-2.8	-6	-2.5	0	0	0	-2	-3.2	0	-2.3	3.7	-1.5	2.4
D1-E329 (+) (B)	0	2.5	4.8	3.6	2.6	0	0	0	2	2.4	0	-3	1.2	2.1
Mn4 (-) (A)	-3.9	-4.2	-2.9	0	-6	-3.6	0	0	0	-4.3	-6.7	-8.2	-3.3	2.8
Mn4 (+) (A)	5.7	4.9	4.4	4.4	12.3	9.9	3	0	0	4.6	5.9	8.2	5.3	3.6
Mn4 (-) (B)	-3.1	-3.7	0	0	-6.3	-5.7	-2.9	-3	0	-4.5	-8.2	-7.6	-3.8	2.9
Mn4 (+) (B)	5	4.9	0	6	10.7	9.9	4.2	3.3	3	4.8	5.2	4.4	5.1	2.9

Second, we need to point out that the correlation of peak intensities with the number of electrons should be different between a pair of positive and negative peaks, and those of a single negative (or positive) peak. As shown in Fig. 3 below, a pair of positive and negative peaks reflects the movement of an atom from the negative peak to the positive peak after pump (light). Since the electron density has a Gaussian distribution with the center of the atom most strong, the difference density between those after pump (light) and before pump (light) will show a smaller amplitude, as the center (strong) density is cancelled out by the subtraction, and the remaining positive and negative densities are at the periphery of the atom after and before movement, which are naturally smaller than the center of the atom. On the other hand, in the case of disappearance of an atom, the whole density is disappeared, from which the center (the most strong) density will become the most strong negative peak. So this negative density is naturally larger than those of the pair of positive and negative densities, and the number of electrons reflected by the negative density will be different from the number of electrons reflected by the pair of positive and negative densities. The reviewer may take the positive and negative densities around CI-1A, D1-D342 and D1-E329 for just showing the error bars in our data, but we feel that there is a need to point out this difference.

Fig. 3. Two cases of atom movement after pump in the pump-probe experiment. In the first case (upper side), an atom is originally in the A1 position, but moves to A2 position after the pump (light). Subtraction of A1 from A2 (A2-A1) will result in a pair of positive (green) and negative (red) densities. However, due to the Gaussian distribution of the electron density, the most strong densities in the center of the atom will be cancelled out, resulting in a smaller amplitude of the difference

densities. In the second case (lower side), an atom is disappeared from its original position after pump, which results in a negative difference density of the whole atom, which is naturally stronger than the pair of positive and negative densities seen in the first case.

Third, we are not sure where “ $+5.7\sigma$ to $+6.6\sigma$ next to Mn4” pointed out by the reviewer came from. As shown in Table 2 above as well as in our Extended Data Table 2 in the manuscript, we do not see the $+5.7\sigma$ to $+6.6\sigma$ peaks next to Mn4. The difference densities around Mn4 are formed by a pair of positive and negative ones after either 1F or 2F (Fig. 4 below), reflecting a movement of Mn4 after pump rather than an addition of electrons to W1, W2, O4 and O5. Although the negative density around Mn4 is rather small after 1F (Extended Data Table 2), the positive density is not paired with the negative density of W16 after 1F (in the case of 1F, it is paired with a small negative density near Mn4, and in the case of 2F, it is paired with a large negative density between Mn4 and O5, see Fig. 4 below). As a matter of fact, the difference densities around Mn4 are not overlapped with W1, W2, O4 and O5 (see Fig. 4 below, only O4 after 2F is somewhat overlapped with the positive difference density, but it is at the edge of the density and cannot be considered an addition of electrons to O4). These results suggest that no electrons are added to the position of W1, W2, O4 and O5, especially after 2F, which supports our interpretation that a new water molecule O6 has inserted into OEC during the 1F-to-2F transition.

Fig. 4. Fourier difference densities around Mn4 during 0F-1F (left side) and 1F-2F transitions (right side). The color of the densities are indicated in the figure.

Thirdly, I also have different interpretations for the O6* peaks as follows. For example, Kamiya and colleagues have shown that the two OECs of two monomers of PSII dimer have different structures (Tanaka et al., 2017). Monomer B of their 5b5e structure has an outstanding positive peak of $+6.4\sigma$ (which was the 13th highest unexplained peak in its F_o-F_c residual difference) near O1 and Ca1 of the OEC, and the D1-A344 sidechain (also distant to D1-E189) in a position similar to the O6* position of the OEC of monomer A of this study (as a proposed precursor for O6 in this study by Shen and colleagues). This peak is not present in monomer A. This residual peak was also present only in monomer B of their 5b66 structure although it was $\sim 4.0\sigma$ in monomer A (smaller but still significant, however was again not present). There are many other

differences between two monomers of these structures according to Kamiya and colleagues. Likewise, Shen and colleagues earlier also showed that monomer B of their 4ub6 and 4ub8 structures had the same peak but was interpreted as a third water molecule bound to Ca1 of the OEC (Suga et al., 2015). This peak was not present in monomer A. That is, Ca1 of the OEC in monomer A had two terminal water molecule ligands whereas Ca1 of the OEC in monomer B had three water molecule ligands.

In more recent studies of Shen and colleagues (Suga et al., 2019), this peak was still present, again only in monomer B of their 6jll structure but not in monomer A even though this peak remained unexplained in the atomic models. Again, the replicated 6jlm and 6jll structures underwent very different light-dependent photochemical transitions but did not undergo the same structural transitions between the two monomers and between the two replicates. Moreover, Wang and colleagues showed that two monomers of dark-adapted PSII dimers had different metal ion compositions in the OEC binding pocket (Wang et al., 2021b). Not only did the two monomers of PSII differ, but also individual dimers of dark-adapted PSII structures had different metal ion compositions inside the OEC pocket.

(Response)

Thanks for the insightful comments. We fully agree that there are some differences between the two monomers of PSII in the dark structures. Those differences become apparent even in OEC, especially when the resolution analyzed is high (Tanaka et al., 2017). They probably arise from sample preparation, manipulation of crystals, and crystal packing. Although we did it similarly, we sometimes noticed the differences in the environment of Ca, as the reviewer pointed out. We are not sure what is the source for the difference of one water molecule near Ca between the two monomers, but suspect that the additional water density observed in monomer B of our structure may be related with the movement of the Ca ion in monomer B. However, we want to emphasize that in our current time-resolved structural changes near the Ca, we found that the signals, including O6*, did not exist at a very early time after 2F, nor in the dark or 1F, and appeared from 1 μ s but disappeared by 5 ms during the S2-to-S3 transition. Comcomitant with this is the appearance of O6 density. These changes are consistent between two PSII monomers and two independent experiments. From these results, we can say that O6* does not exist in 0F, 1F states, and it transiently appears (or increases) right before the O6 insertion in the 2F state.

Shen and colleagues reported that there appeared positive difference features first on Mn4 and then all four Mn ions and one Ca ion of the OEC, respectively, during 20 ns to 200 ns transition. Their interpretation was that this unusual feature had something to do with the change of electrostatic potential associated with the YZ dot/YZ pair, which remains very unclear and puzzling to this reviewer. Typically, positive difference features could be due to gains of electrons (such as addition of new O ligands in small fraction and distributed among multiple sites) or reduced atomic motions. In the latter case, halos of negative features of much smaller amplitudes would surround the positive peaks, which may remain invisible.

(Response)

We appreciate this comment. We agree that positive difference features could be due to gains of electrons or reduced atomic motions. The positive features first on Mn4 and then all four Mn ions and one Ca ion early after 1F is indeed accompanied by small negative features (Extended

Data Table 2). We thus consider that this positive feature reflects the reduced atomic motions, rather than gains of electrons, as gains of electrons during 20 ns to 200 ns after 1F are not possible. We have revised this part of the manuscript to reflect this situation.

Fourthly, I like the animations presented by Shen colleagues because they help visualize structural changes very clearly. It is clear that the loss of W16 in the O4-connected channel occurred during the 0F to 1F transition, but not during the 1F to 2F transition. The authors used these observations to rule out a so-called carousel mechanism for substrate water molecule entry to the OEC in their paper. The authors propose an alternative water-substrate entry mechanism from the O1-connected channel via O6*. As discussed above, this O6* water molecule had a serious problem of its own. More importantly, Wang and colleagues showed that single-conformer of the OEC without inclusion of any sixth O ligand added to the OEC could fully explain the difference Fourier maps between the 2F and 1F data sets of both cyanobacterial species (Wang et al., 2021a). Their analysis implies that it is not crystallographically justifiable to arbitrarily add a second conformer containing the hypothetical O6-added OEC model, especially given the fact that a single-conformer model refinement without an O6-containing model could fully explain all the observed difference Fourier features.

(Response)

We are glad to hear the positive comments on the animation. It is indeed helpful to trace changes occurring at various time points. Regarding the O6* water molecule, we have already mentioned above. We would like to point out that the single-conformer model without invoking the insertion of O6 has its own problem in that, the distances of O5 to the near Mn-ions are too long, and the difference maps between 2F-1F or 2F-0F cannot be fully explained by the single-conformer model. As we already answered above, the O5 and O6-omitted map shows an elliptical shape which is best explained by putting two independent atoms instead of a single atom. The Mn1-Mn4 distance is elongated after 2F, which suggests the opening of OEC after 2F to accommodate the insertion of an additional atom (O6). In addition, D1-D189 also changed its position to accommodate O6.

Lastly, I acknowledge that Shen and colleagues have done excellent crystallographic work in this study and in the past, often in replicates, and continue to improve the degree of isomorphism between the data sets. With such small overall fractional amplitude differences of the observed structure factors in the current data sets (of only 7 to 11% at about 2.30 Å resolution, Extended Table 3), very subtle structural differences could indeed be revealed. Some of these differences may correspond to about one electron or less. It is important not to overinterpret the excellent experimental results either by the authors or readers. A clarification statement is needed to warn the reader not to overinterpret their crystallographic data. I also recommend to include full details regarding the duration of time lapse between PSII sample preparation and data collection because the oxygen evolution activity of their crystals was not reported for their study and because the activity loss has been reported to be about 20% every three weeks (Suiura and Inoue, 1999). It is always important to include one or two sentences to explain how these new XFEL data sets differed from their own previous works already published.

(Response)

We are grateful for your positive comments. Based on the reviewer's suggestion, we have added a statement for cautions on not to overinterpret the crystallographic data (last sentence of page 35), and explained the duration between sample preparation and data collection, in the revised manuscript (Last sentence of page 30).

References

- Drosou, M., Comas-Vila, G., Neese, F., Salvador, P., and Pantazis, D.A. (2023). Does Serial Femtosecond Crystallography Depict State-Specific Catalytic Intermediates of the Oxygen-Evolving Complex? *J Am Chem Soc* 145, 10604-10621.
- Suga, M., Akita, F., Hirata, K., Ueno, G., Murakami, H., Nakajima, Y., Shimizu, T., Yamashita, K., Yamamoto, M., Ago, H., et al. (2015). Native structure of photosystem II at 1.95 Å resolution viewed by femtosecond X-ray pulses. *Nature* 517, 99-103.
- Suga, M., Akita, F., Yamashita, K., Nakajima, Y., Ueno, G., Li, H., Yamane, T., Hirata, K., Umena, Y., Yonekura, S., et al. (2019). An oxyl/oxo mechanism for oxygen-oxygen coupling in PSII revealed by an x-ray free-electron laser. *Science* 366, 334-338.
- Sugiura, M., and Inoue, Y. (1999). Highly purified thermo-stable oxygen-evolving photosystem II core complex from the thermophilic cyanobacterium *Synechococcus elongatus* having His-tagged CP43. *Plant Cell Physiol* 40, 1219-1231.
- Tanaka, A., Fukushima, Y., and Kamiya, N. (2017). Two Different Structures of the Oxygen-Evolving Complex in the Same Polypeptide Frameworks of Photosystem II. *J Am Chem Soc* 139, 1718-1721.
- Wang, J., Armstrong, W.H., and Batista, V.S. (2021a). Do crystallographic XFEL data support binding of a water molecule to the oxygen-evolving complex of photosystem II exposed to two flashes of light? *Proc Natl Acad Sci U S A* 118.
- Wang, J., Gisriel, C.J., Reiss, K., Huang, H.L., Armstrong, W.H., Brudvig, G.W., and Batista, V.S. (2021b). Heterogeneous Composition of Oxygen-Evolving Complexes in Crystal Structures of Dark-Adapted Photosystem II. *Biochemistry* 60, 3374-3384.

Responses made in response to the comments by Referee #2

Referee #2 (Remarks to the Author):

The authors have employed serial femtosecond crystallography (SFX) to characterize structural changes that accompany the S1 to S2 and S2 to S3 transitions in the O₂-forming reaction of Photosystem II. Importantly, the authors report structural models at various time points during each transition, specifically, at 20 ns, 200 ns, 1 μ s, 20 μ s, 200 μ s, and 5 ms after the saturating flash that initiates each desired transition. Understanding the nature and sequence of the structural events during the individual S state transitions is necessary for understanding the mechanism of O₂ formation. This is especially true for the S2 to S3 transition because pulsed EPR data show that a water molecule (and possible substrate) deprotonates and relocates to a Mn ion during this transition. Determining the identity of this water molecule and how it reaches the Mn₄CaO₅ cluster from the lumen is necessary for differentiating between and refining theoretical models for O-O bond formation. This study is another impressive contribution from the Shen and Suga labs. It is the first study to report the nature and sequence of events during the S1 to S2 transition and provides much higher time resolution than a previous study of intermediate structures during the S2 to S3 transition (refs. 26 and 15). The data are exciting, well-described, and will be of interest to anyone interested in O₂ production in nature or artificial systems. However, before the study is published. The authors should address two points:

(Response)

We thank the reviewer for his/her highly positive and encouraging comments and valuable suggestions to improve our manuscript. Your suggestions and comments are insightful and valuable in improving the manuscript. We have considered the concerns raised by the reviewer carefully, and provide point-by-point responses to your comments in the following text.

1. Given the authors' estimations of Si state populations in parallel crystalline samples with FTIR show that their 1-Flash samples contain 40% S1 and only 60% S2, and that their 2-Flash samples contain 49% S2 and only 51% S3 (bottom of page 28), the authors should describe the methods they employed to estimate the contributions of the substantial fractions of PSII that did not advance to the desired S state because of misses (i.e., 40% of PSII in the 1F samples and 49% of PSII in the 2F samples).

(Response)

We estimated that the S1/S2 and S2/S3 ratios after the first and second flashes are 0.4/0.6 and 0.49/0.51, respectively, based on the FTIR studies conducted previously (Kato et al., 2018, Suga et al. 2017, 2019). Based on these ratios, we modeled the structures after 1F as 40% S1/60% S2, and after 2F as 49% S1+S2/51% S3. The S1 state structure in the 1F data is taken from the dark structure solved in the present study. On the other hand, in the 2F data, the structure of PSII that does not advance to the S3 state is a mixture of S1 and S2. However, due to the small structural changes between S1 to S2, we fixed the structure to the S2 state for PSII that does not advance to the S3 state after 2F. These assignments do not impose major problems in modelling the structures based on the densities obtained, thus, we consider that these assignments are reasonable. We have added these statements into the Methods section of the revised manuscript (lines 9-18, page 35, modified manuscript).

2. The authors should address the criticisms raised in Wang et al. PNAS (2021) 118 e202392118 (<https://www.pnas.org/doi/full/10.1073/pnas.2023982118>) regarding assigning electron density near the Mn ions of the Mn₄CaO₅ cluster, particularly to O₆.

(Response)

Wang et al. (PNAS, 2021) employed XFEL data acquired initially by M. Ibrahim et al. (2020) to calculate the 2F-1F and 2F-Dark difference Fourier maps. The analysis revealed a positive difference density peak near O₆ (O_x in M. Ibrahim et al.'s work), located at distances of approximately 1.1 Å from O₆ and 1.6 Å from O₅. Wang et al. postulated a discrepancy between the positions of O₆ and this positive difference density. Considering the presence of negative difference density adjacent to O₅ in the vicinity of Mn₄, they proposed that the displacement of O₅ might have contributed to the observed positive density near O₆, rather than the insertion of a new water molecule. Subsequent refinements of the 2F data using a 1F single-conformation model, which lacks O₆ and O₅ moves toward the presumed O₆ position by 0.5 Å, yielded noise-level mFo-DFc signals on OEC. This observation illustrated that the 2F data (M. Ibrahim et al, 2020) can be explained without invoking the insertion of O₆.

However, we hold a different perspective regarding the absence of O₆ insertion into the OEC after 2F. In the current study, the Fo_{2F(5ms)}-Fo_{1F} difference Fourier map, contoured at $\pm 7.0 \sigma$, reveals that O₆ is not located at the center of the adjacent positive density (with a peak at $+9 \sigma$), but rather at the periphery of this approximately spherical positive density (Fig. 1a below). The distances from this positive peak to O₅ and O₆ are 1.9 Å and 0.4 Å, respectively (Fig. 1b below), suggesting that the positive difference density is more likely derived from O₆. We can explain why O₆ does not coincide perfectly with the center of the positive difference density as below. There is a relatively small negative difference density on the opposite side of the positive difference density with respect to Ca (with a peak at -4.7σ) (Fig. 1a). This pair of positive and negative difference densities can be interpreted as the inward movement of Ca toward the OEC and the positive peak near O₆ may arise from a combination of O₆ insertion and Ca movement, which results in a deviation of the position of O₆ from the center of the positive density.

To validate the location of O₅ and O₆, we employed Polder omit maps, wherein either O₆ (as depicted in Fig. 1c, left panel) or both O₅ and O₆ in the 2F structure (as shown in Fig. 1c, right panel) were omitted. The resultant maps clearly revealed that O₆ was located at the peak's center. Moreover, the elliptical shape of the peak when both O₅ and O₆ were omitted strengthens the idea that two atoms O₅ and O₆ occupy their positions rather than a single O₅ atom in the 2F (5 ms) data.

Furthermore, when we reproduce the single O₅ model by the methodology of Wang et al. (2021), residual positive Fo-Fc peak appeared near the O₆ site, which has a peak height of 3.45σ (Fig. 1d). Because our model containing both O₅ and O₆ atoms did not show such residual densities, we consider that both our interpretation and crystallographic analysis are valid. If we assume that the OEC consists of a single O₅ atom, distances from the O₅ atom to Mn₁, Mn₃, and Mn₄ would become 2.65 Å, 2.04 Å, and 2.48 Å, respectively. These distances, especially the distances of O₅-Mn₁ and O₅-Mn₄, are too long for a covalent bond, suggesting that both Mn₁ and Mn₄ would be isolated. This is inconsistent with the Mn valences suggested by both spectroscopic measurements as well as theoretical calculations, and is also improbable in chemistry.

In summary, we concluded the insertion of O6 into OEC following 2F based on the analysis of Fo2F(5ms)-Fo1F difference Fourier maps and the Polder omit maps.

Fig. 1 | Investigation of O6 insertion into OEC at delay time of 5 ms after the second flash. (a) and (b) shows the model of 2F (5 ms) superposed with the $F_o^{2F(5ms)}-F_o^{1F}$ difference Fourier maps. Map code and contour level are indicated in the figures. (c) Polder omit maps, with O6 omitted (left panel) and both O6 and the B conformation of O5 omitted (right panel). (d) Refinement of the OEC consists of single O5 against 2F data. 2Fo-Fc map contoured at 1.0 σ is shown in blue mesh and Fo-Fc map contoured at $\pm 3.0 \sigma$ is shown in green and red mesh. The residual densities near O5 and Mn1 is colored in green and has a peak height of $+3.45 \sigma$. Mn4-O5 distance was refined to be 2.48 Å.

I applaud the authors for including Extended Data Table 1, showing the correspondence of their

numbering of water molecules in the O1, O4, and C11 channels with the numbering in several PDB entries. I strongly urge the authors to add an additional column showing the correspondence of their numbering with that adopted by the Berkeley group in their published figures (e.g., those in refs 15, 25, and 26). Non-crystallographers would greatly appreciate it.

(Response)

Thanks for the reviewer's suggestion. We added a column describing the correspondence of numbering of ours with those of Hussein et al. to Supplementary Table 1 in the revised manuscript.

Finally, I noticed a few typographical errors:

1. Page 18, Line 401, CP43-V410
2. In Extended Data Table 2, the asterisk is missing from the CC1/2 data.
3. Supplementary Materials, line 124, "In the S2 state, THE following changes occur..."
4. Supplementary Materials, line 225, D1-D61 (not E61).

(Response)

We apologize for the typos, and corrected all of them in the revised manuscript based on the reviewer's comments. We are grateful for the reviewer's careful reading of our manuscript. One thing we want to point out is that we are not using CC* but using CC1/2.

Responses made in response to the comments by Referee #3

Referee #3 (Remarks to the Author):

Review of the manuscript entitled: “Structural Dynamics of oxygen-evolving Photosystem II during S1-S2-S3 Transition by Li et al.

A. Summary of the key results

This manuscript is very exciting as it reports on the first sub-microsecond dynamic studies on the mechanism of water splitting on Photosystem II by serial fs crystallography at an XFEL for the S1 to S2 and S2 to S3 S-state transition. While previous studies were limited to the us and millisecond time frame and mainly focused on the S2 to S3 transition, this work is of extremely high importance as it reveals with high time and spatial resolution the sequence of events that happen both at the QA/QB quinone acceptor site of PSII as well as at the donor site P680, the oxygen-evolving cluster (OEC) and the water and proton channels that bring the substrate waters to the OEC and mediate the proton release to the bulk. The authors have collected high-resolution SFX data sets at RT at the XFEL at SACLA for both the S1 to S2 transition and the S2 to S3 transition at time delays of 20ns, 200ns, 1us, 30us, 200us, and 5ms. 100um crystals were delivered to the XFEL beam in a viscous jet at RT and they were excited by a split laser beam from two sides “on the fly” prior to the 10fs X-ray exposure. This paper is based on 14 excellent time-resolved SFX data sets (7 for the S1-S2 transition including the dark data set and 7 for the S2 to S3 transition including the 1-flash S2 5ms data set that serves as the “baseline” for the calculation of the difference density maps for the S2 to S3 transition. The referee was very impressed and satisfied with the high quality and multiplicity of each of the data sets, where each data set consist of 30 000 to 66 0000 hits leading to 24 000 to 59 000 indexed images per data set. For all data sets the overall multiplicity is >200 and even in the highest resolution shell exceeds >100, at resolutions between 2.15 and 2.3 Å resolution. This far higher multiplicity of the data sets and the higher time resolution in the ns time range compared to previous studies on the S2 to S3 transition has allowed them to detect for the first time not only large conformational changes (like the appearance of O6) but for the first time reveal all the more subtle conformational changes that allowed them to detect the movement and flexibilities of the water networks, the movement of the Mn and the Ca in the metal cluster as well as conformational changes of the amino acids.

(Response)

We thank the reviewer for his/her highly positive evaluation and comments on our manuscript. We are very much encouraged by the reviewer’s comments on the data quality of our manuscript. We have considered the concerns raised by the reviewer carefully, and provide point-by-point responses to these comments in the following text.

B. Originality and significance: If not novel, please include a reference

This paper and the results reported are highly novel. As summarized above the paper reports on the first time-resolved studies with sub-us time resolution for the S1 to S2 and S2 to S3 transition of the oxygen-evolving complex in Photosystem II. Taking the extremely high relevance of the unraveling of the mechanism of water splitting into account, which provides all the oxygen for respiration and thereby all high life on Earth, this work is extremely important not only for the

experts in Photosynthesis but also for the general audience. It has also a high significance for the development of clean energy in the future as the knowledge of the mechanism of water splitting in Nature can form the basis for the development of artificial systems that combine the efficiency of Nature with the stability of man-made systems.

(Response)

We thank the reviewer for his/her highly positive and encouraging comments

C and D. Data & methodology: validity of approach, quality of data, quality of presentation
The TR-SFX studies presented in this work are state-of-the-art in the field. Reaching time resolutions nanoseconds brings time-resolved pump-probe SFX of Photosystem II into a new time regime and the large difference density changes observed at the 20 and 200ns time points show that multiple critical steps in the mechanism of water splitting are revealed in this study for the first time. The data quality is excellent. As outlined in the summary above the quality of the data sets collected with extremely high multiplicities is very critical for the quality of the difference density maps. In contrast to standard crystallography, all reflections are partials and Monte Carlo Integration over a large data set of these partial reflections is very important for the determination of accurate structure factors. The reviewer is thereby fully satisfied with the data, the methodology, and the data validation approach.

This study thereby by far exceeds previous studies on the S2 to S3 transition (Ibrahim 2020), where the fastest time point was 50us and which had significantly lower data redundancy thereby could detect only a few conformational changes compared to what has now been observed by Suga, Shen and their teams.

The large number of changes in the conformation that are detected in this study extend to multiple regions of Photosystem II including the acceptor site where the two quinones QA and QB are located, the metal cluster of the OEC and its bridging oxygen atoms, the water molecules in the vicinity of the OEC as well as the protein surrounding including side chain and backbone atoms. Furthermore, large changes are observed in the organization of the 3 water and proton channels connecting to the OEC.

The detection of all these changes that are observed at the 6 different time points is a real breakthrough but it also leads to serious difficulties in the presentation of the very comprehensive data. The main figures 2, 3, 5,6, and 7 consist of 12 panels featuring the difference density maps for each of the 6-time points for the S1 to S2 and S2 to S3 transition for the different regions of interest in PSII. Fig 2 shows the dynamics for the QA/QB acceptor site, Fig 3 shows the structural dynamics for the Yz area, Fig 5 features the dynamics of the OEC and its surroundings, Fig 6 shows the dynamics of the O1 channel and Fig 7 shows the dynamic of the O4 and Cl channels. This makes it very difficult for the reader to comprehend these figures and all the dynamic changes that are observed. The supplementary videos are really nice but also difficult to comprehend as so many changes happen at the same time. The problem of representation is most severe for Figures 2, 6, and 7 which show larger sections of the structure and make it very difficult to even read the labels of the amino acids and also prevents the reader from detecting the difference between the structural model of the ground state (dark S1 state for the S1-S2 transition) and (S2 state for the S2 to S3 transition). Here it would be helpful if a larger panel on top could be included that shows the dark structure with amino acids and water

molecules labeled and then labelling of amino acids is avoided in the individual panels allowing for structural changes to be more clearly marked in the individual panels (they are currently only featured by small black arrows, which could be featured more prominently).

(Response)

We are grateful for the reviewer's suggestion. As the reviewer pointed out, representing twelve intermediate states in a single figure is indeed challenging, especially in Figures 2, 6, and 7. To improve the reader comprehension and comply with Nature's figure size requirements, we have carefully optimized the layout of multiple panels, labels, arrows, and circles with the dashed lines within the figures. In response to the review's suggestion, we changed the labels to a larger font in the top panel of each figure, instead of including a large top panel illustrating the dark structure with labeled amino acids and water molecules due to the limited space. We hope these modifications enhanced the clarity and satisfied the reviewer's suggestions.

E. Conclusions: robustness, validity, reliability

The conclusions drawn in the paper are very well validated by the data and thereby reliable, however, the comprehensive conclusion of the order of events in the S1-S2-S3 transition is missing in the main text, which just finished with one very general conclusion sentence and no figure is provided in the main text that summarizes the exciting results.

However, when the reviewer was reading the supplementary discussion and looked at the extended data file, the picture of events was very nicely summarized in the supplementary discussion and furthermore, extended data figures 7 and 8 provided a very nice comprehensive overview of the major events in the water channels in Figure 7 and at the electron donor site (including the OEC) in Figure 8. The reviewer therefore suggests that the supplementary discussion should be included as the conclusion in the main text and that at least Figure 8 should become the final figure of the main text. The reviewer is aware of the fact that Nature has a figure and word limit for the main body of the paper but due to the very complex Nature and the importance of the conclusion for the comprehension of the results by the general audience the reviewer hopes that the word and figure limit could be extended for this very important paper.

(Response)

We appreciate the reviewer's positive and encouraging comments and suggestions. We have moved Extended Figure 8 in the original manuscript into the main body of the modified manuscript. To comply with the length limitation of Nature, we instead moved Figs. 1 and 4 in the original manuscript to Extended Data, and also kept the Discussion in the Supplementary Information. Instead, we added a sentence in the end of the main text (line 2, page 20, modified manuscript) to refer readers to our Supplementary Discussion (...detailed discussion on the model is described in the Supplementary Information). We hope our modifications are satisfactory to the reviewer.

F. Suggested improvements: experiments, data for possible revision

While the reviewer was very satisfied with the level of detail provided for the X-ray data collection and data evaluation the reviewer was very disappointed by the lack of information on protein expression, purification, crystallization, crystal dehydration as well as embedding of the

crystals into the viscous media. The reviewer therefore suggests MAJOR REVISIONS of this section of the paper in the materials and method section. The authors cite that the protein isolation and crystallization have been done as described previously with minor modifications but looking at all the previous publications none of them contains a detailed comprehensive description starting from cell growth to protein isolation, crystallization, and preparation of the crystals for sample delivery.

Growth conditions: Detailed information on the growth conditions including media light conditions, spectra of the cell's growth curves, and OD at the time of harvest should be included. This is essential as PSII undergoes constant reconstruction due to photodamage and the expression pattern changes of the three genes exist for the PsbA (D1) depending on the growth conditions this information is absolutely essential. It should also be indicated if the cells were WT cells of *TS vulcanus* or if a mutant lacking some of the PsbA genes has been grown for these studies.

Cell disruption and PSII isolation: Details including info on cell disruption, membrane isolation, protein detergent solubilization, and all purification steps must be included in the paper including info on all buffers and detergents used and times and temperature conditions for all protein isolation steps. Detergents of high purity are critical for the protein quality and also the quality of the crystals so please also include the info on the supplier of the detergents and their purity.

Crystallization of PSII: Details of the PSII crystallization must be included including all crystallization steps with protein concentration, buffers, detergents, incubation times (seeding conditions if used) additives etc. The crystals undergo a dehydration procedure before sample delivery which is critical for reaching high-resolution diffraction but currently, no information on this process is provided in this paper or previous publications. The reviewer therefore requests to include all details (including incubation steps and times) of this dehydration procedure and indicate how long the crystals are stable in the dehydration buffer. It would be also nice if a figure of the crystals used for the experiments could be included in the extended data file.

Embedding of crystals in viscous media:

PSII crystals are very sensitive and it is therefore critical that mechanical damage and stress are avoided in the embedding procedure of the crystals in the viscous media for sample delivery. The reviewer requests that details of the embedding procedure be provided.

(Response)

We apologize for the lack of details of the conditions for cell growth, protein isolation, crystallization, and preparation of the crystals for sample delivery. We have added these detailed information into the Method section and cited the previous publications in appropriate positions. This part of the Methods section has been thoroughly re-written, and the detailed information has been added. Please refer to page 28-30 of the revised manuscript.

G. References: appropriate credit to previous work?

All important relevant previous publications are cited in the paper.

H. Clarity and context: lucidity of abstract/summary, appropriateness of abstract, introduction, and conclusions.

The abstract/ summary represents a very good summary of the major findings and the introduction is also very appropriate summarizing the current status of the knowledge and the

open questions on the function of Photosystem II and the mechanism of water splitting. As outlined above the paper is missing a major conclusion section in the main text as outline above and the reviewer suggests including therefore part of the supplementary discussion as the conclusion in the main text.

(Response)

We appreciate the suggestion. We have added a short paragraph to present the major conclusions of the paper (page 19-20, last paragraph⁹). Due to the strict length limitation of Nature, we have to keep the discussion section in the Supplementary Information. However, we have added a sentence in the last line of the main text to refer the readers to the Supplementary Discussion. We hope that these modifications are satisfactory to the reviewer.

Minor comments:

Text on the QA-Fe-QB complex (last paragraph on Page 7): Please mention in the text at which time point the individual changes in the electron density map are visible.

(Response)

Thanks for the suggestion. We have modified the text in the manuscript as below: “These changes correspond to formation of Q_A^- at $\Delta t_1 = 20$ ns and 200 ns, oxidation of Q_A^- to Q_A at $\Delta t_1 = 1$ μ s to 200 μ s, and completion of Q_A^- oxidation at $\Delta t_1 = 5$ ms, respectively. Formation of Q_A^- at $\Delta t_1 = 20$ ns and 200 ns causes rotation of its head group counterclockwise...” (lines 3-6 of last paragraph, page 6, revised manuscript).

Page 8, second paragraph: Please give an explanation for the shortcut BCT

(Response)

We have spelled BCT as bicarbonate in the first line of the second paragraph, page 7 of the revised manuscript.

Page 8 end of second paragraph: change the wording of “...BCT or reduction of partial QB by QA...” to: ...BCT or partial reduction of QB by QA...

(Response)

We modified the text in the revised manuscript as the reviewer suggested; thank you.

Page 8 last sentence: The authors state that the structural changes on the QA binding site and the non-heme iron that occurred during this electron transfer are identical to those following the first flash. However, as stated in the next sentence the difference densities are weaker. This is an understatement as the difference densities are very strong for S1-S2 and very weak in this region for S2 to S3, so the reviewer can not understand how they can be identical when there are such large differences in the intensities. Please revise this paragraph.

(Response)

Thanks for the critical comments. We have modified the text to avoid the term identical as follows:

“During $\Delta t_2 = 20 \text{ ns}$ to $30 \mu\text{s}$, Q_A head rotated counterclockwise along with a movement of the non-heme iron toward Q_A , which is similar to structural changes observed after 1F. However, difference densities associated with these structural changes were much weaker after 2F (Fig. 1 and Supplementary Movie 2). The non-heme-iron photo-reduced by 1F maintains as Fe^{2+} at 5 ms after 1F, as its re-oxidation by ferricyanide takes 20 s (Extended Data Fig. 1d)....” (last paragraph of page 7, revised manuscript).

Page 9 end of last paragraph: Please explain how the iron would promote electron transfer by becoming disordered.

(Response)

Thanks for the insightful comment. Since this statement seems speculative, we have modified this sentence as follows:

“The non-heme-iron becomes disordered at $\Delta t_2 = 200 \mu\text{s}$ but ordered by $\Delta t_2 = 5 \text{ ms}$, which is presumably related to electron transfer from Q_A^- to Q_B during this time” (lines 1-3, page 8, revised manuscript).

Page 10 first sentence: The authors note that a positive difference density appears on the Mg atom of PD1. This is a very interesting finding as PD1 very likely represents the chlorophyll where the unpaired electron in $\text{P680}+\lambda$ is located that then accepts an electron from the Mn cluster via TryZ to return to the neutral P680 state. Does this mean that they have detected the re-reduction of $\text{P680}+\lambda$ to P680? If so that would be extremely exciting and should be highlighted more prominently in the main text.

(Response)

Thanks for the insightful comments. At $\Delta t_1 = 200 \text{ ns}$, we detected the positive density of $4.9/4.0 \sigma$ (monomer A/B) on the Mg atom in PD1. This could indeed reflect the re-reduction of $\text{P680}+\lambda$ to P680 by the electron from the Mn_4CaO_5 cluster. We added a sentence to reflect this in the revised manuscript:

“...a positive difference density appears on the Mg atom of PD1 (Fig. 2a and Extended Data Table 2), which may reflect re-reduction of P680^+ by Yz (last two lines of the second paragraph, page 8, revised manuscript).

Page 11 last paragraph: Here difference densities of Mn 4 are discussed which is later extended through the metal cluster while the metal-metal distances stay the same. The authors state that these differences may be caused by a charge rearrangement of the OEC triggered by the electrostatic effect of $\text{TryZ}\lambda^+/\text{TyrZ}\lambda$. This is also a very important finding, could the authors include QMM calculation that model these charge rearrangements?

(Response)

Thanks for the positive comment. While we believe this finding is important, and the QM/MM calculations may clarify this question, we have not conducted QM/MM calculations as this will take time and may exceed the scope of the present study. We hope to address this question in our future studies.

Page 17 first paragraph: It is not clear to the reviewer how a negative difference density could arise for W53' when W53' is not visible in the electron density in the first place as stated earlier.

(Response)

Thanks for the critical comment. W53 is located within the cavity surrounded by OEC-O1, D1-E189, D1-E329, D1-H332, and D1-D342 (Fig. 1 below). It is only visible in dark structures collected at cryogenic temperatures and high resolution, but invisible at room temperature, including the current dark structure (Fig. 1 below). Since the protein environment around W53 is the same regardless of the temperature, W53 is considered present but fluctuating at room temperature, making it invisible at room temperature. At $\Delta t_1 = 30 \mu\text{s}$ to $200 \mu\text{s}$, a negative difference density appears at the position of W53 at room temperature. A negative difference in electron density can be attributed to a disorder or electron loss. This signal disappears by 5 ms, and at this time point only W10 in the O1 channel remains unstable. We speculate that the negative difference density observed at the W53 position in the present study may be due to a small fraction of W53 becoming further unstable transiently.

To improve the clarity of the text, we modified the original text to the following:

“W53 in S₁-state is only observable under cryo-temperature conditions (pdb codes: 3wu2² and 4ub6³), but is not detectable at room temperature (pdb codes: 5ws5⁴ and 7cji2⁵). Since the protein environment around W53 is the same regardless of temperature, W53 is considered present but fluctuating. Therefore, in this study, we denote this invisible water as W53' ” (lines 5-10 of the second paragraph, revised manuscript).

Fig. 1 | W53' at cryo-temperature (4ub6 and 5b5e) and room temperature (dark and 1F (200 μs)-dark).

Page 18: The authors mention Go11, however, the reviewer could not find it in Figure 7A

(Response)

We apologize for the unclarity. We have resized the images and adjusted the resolution and size to improve the labels. Go11 is now well-recognized near the top-right corner of Fig. 4 of the revised manuscript.

Methods page 26: The authors have used the SFX data evaluation program CrystFEL during the beamtime to observe and analyze the diffraction images and then used cctbx.xfel later for the indexing and integration of the diffraction images as well as for merging reflections after the beamtime. This is very interesting and it would be important to indicate why they switched from CrystFEL to cctbx. xfel in the later stages of the data analysis. It would be great if they could include a summary of the features in cctbx. xfel that they used. For example, was the program feature that allows for reconstruction of the peak profiles of the partial reflections in cctbx. xfel superior to the corresponding program element in CrystFEL? It would be nice if a comparison of the data quality obtained with the two programs could be included.

(Response)

During our data collection, we chose to use *CrystFEL* over *cctbx.xfel* for the reason that *CrystFEL* is integrated with the *Cheetah* software installed at SACLA, which enables fast indexing and merging of data in real time, aiding us in effectively managing our data collection strategy. Subsequent to data collection, we performed analyses with both *CrystFEL* and *cctbx.xfel* software packages and compared the results, which showed that indexing with *cctbx.xfel* gave us a larger number of images indexed than with *CrystFEL*, resulting in an enhancement of approximately 0.1-0.2 Å in resolution. Additionally, *cctbx.xfel* allowed us to refine models with lower B factors. Consequently, our decision was to process the data using *cctbx.xfel*. However, it is important to clarify that the findings mentioned above were based on the analysis of PSII data using older versions of *CrystFEL* (0.6.3) and *cctbx.xfel* (*dials.stills_process* and *cxi.merge*), as our data analysis has started around 3 years ago when the new versions of these softwares were not available. Recent updates in *CrystFEL* (0.10.2), notably the introduction of the '*xgandalf*' indexing method, may significantly increase the number of indexed images. Additionally, the merging program in *cctbx.xfel* has transitioned to *cctbx.xfel.merge*. As a result, we are currently uncertain which software may produce superior results using the new versions of these two software packages. Performing a thorough comparison of these two software packages is beyond the scope of our current manuscript. However, we may conduct a more in-depth analysis of these software packages in our future research.

Methods page 28: last paragraph, the authors indicate that they determined the populations of the Si states in the crystals based on FTIR measurements for which they cite 3 references but no data. It is therefore not clear if they did the FTIR experiments under the same conditions and

laser excitation as in the jet. It would be important to include the results of the FTIR experiments in the extended data and provide a description how they were done in the methods section.

(Response)

We are grateful for the comments and admit the inadequate description and references regarding estimation of the S-state populations. We did not perform the FTIR experiments in this study since the crystal conditions are the same as in our previous study (Suga et al., Nature 2017). We have modified the manuscript to describe the estimation and referred to the paper for the estimation of S2/S3 populations (Kato et al., 2018) in our revised manuscript.

Methods Page 30: the authors describe DFT calculations here but the results of the DFT calculations are not shown or discussed in detail in the main text.

(Response)

We apologize for the missing descriptions of DFT calculations in the main text. We have included the results of the DFT calculation in Extended Data Figure 5 and described them in the main text (last two lines of page 11, revised manuscript).

Figures: The main request for revisions of the Figures of the main text has been discussed above. In addition, the reviewer wants to suggest a change in the color scheme for the water molecules in Figure 1. They are currently labeled with yellow numbers on an orange ball and this “tone in tone” labeling makes it very difficult to read the numbers. Please use a different color scheme here.

(Response)

We changed the color of the water number to black. We thank the reviewer’s suggestion for the improvement of the figure clarity.

Reviewer Reports on the First Revision:

Referees' comments:

Referee #1 (Remarks to the Author):

Authors have carefully addressed concerns of all three crystallographic reviewers. I accept almost all of their responses with one exception, which requires an additional minor revision. It corresponded to point 2 of reviewer #2: "The authors should address the criticisms raised in Wang et al. PNAS (2021) 118 e202392118 (<https://www.pnas.org/doi/full/10.1073/pnas.2023982118>) regarding assigning electron density near the Mn ions of the Mn₄CaO₅ cluster, particularly to O₆."

I believe that authors of Wang et al. have been fully aware of that the O₅ distances of 2.65 Å and 2.48 Å to Mn₁ and Mn₄ do not correspond to covalent Mn coordination and therefore directly challenge the current view of valence notion that both Mn₁ and Mn₄ are +4 oxidation states in these 2F structures. To this reviewer, crystallographic data have clearly shown that they are not. It is not surprised that Wang's re-interpretation of crystallographic data are indeed inconsistent with inference from interpretations of spectroscopic data and from theoretical calculations that were based on inference from spectroscopic data. The crystallographic interpretations must be solely based on crystallographic data only with well established crystallographic principles. How can authors argue that Wang et al.'s interpretation is not chemically improbable? In fact, Mn centers in majority of Mn metalloenzymes are often associated 5 ligands in a square pyramidal geometry with a sixth, apical coordination site available for substrate binding (Udayalaxmi et al. 2020, Structural Chemistry 31, 1057-1064). The response from the current authors and the reply from the reply authors to Wang et al., (2021) letter shares the same view to apply inference from spectroscopic data to interpret their crystallographic data.

It should be noted that the crystallographic methods used to solve the macromolecular crystal structures are usually completely independent of the information about their biochemical and biophysical properties that has been obtained by other means. When this is done so, the discovery that the crystal structure of the macromolecule is consistent with other data is powerful evidence that both are correct, and that both have been properly understood. Conversely, inconsistencies between crystal structures and other kinds of data should cause serious concern. It follows that when generating structures from crystallographic data, one should ignore whatever inferences may have been drawn about them using non-crystallographic techniques. Due to the well-known phenomenon called "model bias", it is all too easy to convince oneself of the reality of some artifactual feature of a structural model that was included in it because it was thought to "explain" other data. Once this happens, the fact that a crystal structure is consistent with some particular piece of non-crystallographic data becomes meaningless. This is a particularly big problem for the oxygen-evolving center (OEC) of PSII. The structural changes that occur as the OEC progresses through its reaction cycle are small. The resolution of the crystal structures available for it are respectable for a macromolecule, but not what one would like to have for a system this demanding. Lastly, its spectroscopic properties have been under investigation for decades, and the interpretation of that data heavily depends on whatever one believes the underlying structure might be. To allow interpretations of the spectroscopic data to be used in interpreting the crystallographic data is to invite a particularly noxious kind of feedback. Let the crystallography stand on its own feet and then proceed from there.

Furthermore, Wang et al. (2017, ACS Energy Letter, 2, 2299-2306) showed after they retrieved sA-weighted Fo-Fc maps from the PDB that the inserted O6 atom was outside the zero-sigma contouring levels for both monomers even though O6 was included in model phase calculation. Combined with the above cited Wang et al. (2020) paper, it is a real concern about the validity of crystallographic interpretations of both Okayama and Berkeley groups.

I do not believe that I should withhold the acceptance of this manuscript for publication because of my disagreement with the authors. This paper makes an important contribution to the PSII community. It is important for the authors to cite Wang et al., 2020 paper in the main text to acknowledge that there is a crystallographic debate whether or not O6/Ox exists in the 2F structures. This paper does not address this debate or if it does so, please give your justifications in the main or supporting text. This could help the authors to avoid any challenge by expert crystallographers in the future who may not agree with the authors.

Finally, according to Wang's theory (Wang et al., 2021, Biochemistry 60, 3374-3384), the change of occupancy and the change of atomic motions B-factors can be completely decomposed in the omit maps. The change of occupancy does not alter the half-width at the half-maximum of omit peaks. The half-width at the half-maximum of omit peaks is only a function of B-factors. I hope that the authors will use normalized electron density peaks to substantiate their claim that W16 becomes disordered (i.e., increased B-factor) but it does not lose its occupancy. Otherwise, it appears to this reviewer that the authors arbitrarily exclude the possibility of W16 being one of substrate molecules entering the OEC during the 1F transition while hypothesizing that O6* is one during the 2F transition.

Referee #2 (Remarks to the Author):

The authors have addressed my concerns. I especially like the modifications to Supplementary Table 1. The manuscript can be published without further change, although I recommend that the authors include their rebuttal of Wang et al. PNAS (2021) 118e202392118 as a section in their Supplementary Information so that a wide audience can see it. - Rick Debus

Referee #3 (Remarks to the Author):

Reviewer comments on the revised manuscript of Li et al "Structural dynamics of oxygen-evolving photosystem II during 0F-1F-2F transitions"

The reviewer is very satisfied with the revised version of the manuscript of Li et al. The authors have very carefully revised the manuscript based on the comments of all 3 reviewers. While all 3 reviewers were very positive and impressed about the work Reviewer 1 raised several points about the interpretation of the electron density maps, and was most critical about the question of the appearance of O6, offering alternate interpretations. The authors took this critique very seriously and discussed both the alternate interpretation by this reviewer and their own model based on their experimental electron densities. They also now calculated polder maps and these clearly lead to the conclusion that the interpretation they presented in this paper is valid and they therefore did not change the conclusions drawn. This reviewer fully agrees to the conclusions of the authors and is on

the strong opinion that the authors have addressed the concerns of reviewer 2 sufficiently.

On a side note the authors agreed to change the title of the paper from the S1-S2-S3 transition to 0F-1F-2F transition based on the comments of reviewer 1 also taking the questions of referee 2 into account who asked to include more details on the FTIR measurements that identified the %ages of S states after the 1F 2F and 3F laser excitation. This reviewer does not agree to this title change as the structures determined were from the S1, S2 and S3 state as the authors have carefully taken the S-state content for each data set into account. So, for example, after 1F 40% of the PSII molecules are in the S1 state and 60% in the S2 state. However, the authors have (based on the known S1 structure) taken the occupancy into account and therefore the electron density maps and difference density maps do NOT represent the 40:60 mixture of S1 and S2 states but represent the S2 state at time points tx after the 1st flash. Therefore this reviewer strongly recommends to revert the title back to the original title. This will be also much more consistent to the naming in previous publications on TR-SFX studies on Photosystem II from this group and the group at Berkeley.

The major critique point of this reviewer was focused on the lack of information on the cell culture of *Thermosynechococcus vulcanus* and the PSII isolation, crystallization, the dehydration process as well as information on the embedding of the crystals in viscous media and sample delivery. The reviewer is very satisfied with the revised version of the manuscript, where the authors have rewritten the methods section on isolation and crystallization completely and now provide a comprehensive description of all steps involved in this important part of their study. The reviewer highly appreciates that they describe the process now in detail and also provide the corresponding references. There is only one detail that is missing: the authors state that they further dehydrated the sample after they embedded the PSII crystals in the grease viscous media before sample delivery. This is astonishing as one would assume that a 250ul sample (200ul grease plus 50ul crystals) dries out very fast but obviously the process is slower in grease. It would be important if the authors could provide the optimal incubation time for this additional dehydration step.

The reviewer had asked several minor points that included suggestions on making the figures more readable, which the authors have all considered. The reviewer appreciates all the changes (like labelling the amino acids only in one of the panels, or the clear featuring of difference density changes which are now much more prominently marked and highlighted).

The reviewer had also criticized that the paper lacks a detailed conclusion section and suggested to move the summary and conclusion section included extended data figure 8 into the main manuscript (hoping for the journal to allow for the paper to exceed the figure and word limit). The authors explained that this was not possible to the strict page limitations but they were able to include this important Figure now into the main text of the paper and also included a short conclusion paragraph with reference to the more detailed discussion in the extended data section. The reviewer is satisfied with this solution as it allows the general reader to see the main conclusions of the paper in the main text and also will be able to find the more detailed discussion easily in the extended data file.

One question that puzzled the reviewer was the negative difference density of the "invisible" water W53'. The authors are now explaining that this water is only visible in cryogenic dark state X-ray structures, but they assume that it is there at RT but is disordered and becomes even further disordered after light excitation. The reviewer is satisfied that this interpretation is now included in the main text, as it avoids confusion of the readers.

An interesting answer was given by the authors to the reviewers question about the comparison of the two data evaluation programs CrystFEL and cctbx.xfel. The authors explained that they did the

initial data analysis including hitfinding with CrystFEL and then performed the further analysis both with CrystFEL and cctbx.xfel where they reached higher indexing rates and thereby 0.1 to 0.2 Å higher resolution with cctbx.xfel. They want, however, not draw conclusions and compare the data analysis site by site in this paper as major upgrades have been done in the meantime for both programs. The reviewer understands this reasoning and is satisfied with this answer.

The reviewer had also asked if the results of the DFT calculations could be included in the revised manuscript. The authors explained that this would be beyond the scope of the paper and the reviewer is fine with this answer and looks forward to see this included in a separate paper.

In summary, the authors have fully addressed the comments and suggestions of the reviewers.

As outlined in the original review, the reviewer finds this paper extremely important and is very enthusiastic about the exciting results presented in this paper. The reviewer is fully satisfied with the revisions and strongly recommends the publication of the paper in its revised form in Nature.

**Author Rebuttals to First Revision:
Responses to the comments of the Referees**

Referee #1 (Remarks to the Author):

Authors have carefully addressed concerns of all three crystallographic reviewers. I accept almost all of their responses with one exception, which requires an additional minor revision. It corresponded to point 2 of reviewer #2: "The authors should address the criticisms raised in Wang et al. PNAS (2021) 118 e202392118 (<https://www.pnas.org/doi/full/10.1073/pnas.2023982118>) regarding assigning electron density near the Mn ions of the Mn₄CaO₅ cluster, particularly to O₆."

I believe that authors of Wang et al. have been fully aware of that the O₅ distances of 2.65 Å and 2.48 Å to Mn₁ and Mn₄ do not correspond to covalent Mn coordination and therefore directly challenge the current view of valence notion that both Mn₁ and Mn₄ are +4 oxidation states in these 2F structures. To this reviewer, crystallographic data have clearly shown that they are not. It is not surprising that Wang's re-interpretation of crystallographic data are indeed inconsistent with inference from interpretations of spectroscopic data and from theoretical calculations that were based on inference from spectroscopic data. The crystallographic interpretations must be solely based on crystallographic data only with well established crystallographic principles. How can authors argue that Wang et al.'s interpretation is not chemically improbable? In fact, Mn centers in majority of Mn metalloenzymes are often associated 5 ligands in a square pyramidal geometry with a sixth, apical coordination site available for substrate binding (Udayalaxmi et al. 2020, Structural Chemistry 31, 1057-1064). The response from the current authors and the reply from the reply authors to Wang et al., (2021) letter shares the same view to apply inference from spectroscopic data to interpret their crystallographic data.

It should be noted that the crystallographic methods used to solve the macromolecular crystal structures are usually completely independent of the information about their biochemical and biophysical properties that has been obtained by other means. When this is done so, the discovery that the crystal structure of the macromolecule is consistent with other data is powerful evidence that both are correct, and that both have been properly understood. Conversely, inconsistencies between crystal structures and other kinds of data should cause serious concern. It follows that when generating structures from crystallographic data, one should ignore whatever inferences may have been drawn about them using non-crystallographic techniques. Due to the well-known phenomenon called "model bias", it is all too easy to convince oneself of the reality of some artifactual feature of a structural model that was included in it because it was thought to "explain" other data. Once this happens, the fact that a crystal structure is consistent with some particular piece of non-crystallographic data becomes meaningless. This is a particularly big problem for the oxygen-evolving center (OEC) of PSII. The structural changes that occur as the OEC progresses through its reaction cycle are small. The resolution of the crystal structures available for it are respectable for a macromolecule, but not what one would like to have for a system this demanding. Lastly, its spectroscopic properties have been under investigation for decades, and the interpretation of that data heavily depends on whatever one believes the underlying structure might be. To allow interpretations of the spectroscopic data to be used in interpreting the crystallographic data is to invite a particularly noxious kind of feedback. Let the crystallography stand on its own feet and then proceed from there.

Furthermore, Wang et al. (2017, ACS Energy Letter, 2, 2299-2306) showed after they retrieved sA-weighted Fo-Fc maps from the PDB that the inserted O6 atom was outside the zero-sigma contouring levels for both monomers even though O6 was included in model phase calculation. Combined with the above cited Wang et al. (2020) paper, it is a real concern about the validity of crystallographic interpretations of both Okayama and Berkeley groups.

I do not believe that I should withhold the acceptance of this manuscript for publication because of my disagreement with the authors. This paper makes an important contribution to the PSII community. It is important for the authors to cite Wang et al., 2020 paper in the main text to acknowledge that there is a crystallographic debate whether or not O6/Ox exists in the 2F structures. This paper does not address this debate or if it does so, please give your justifications in the main or supporting text. This could help the authors to avoid any challenge by expert crystallographers in the future who may not agree with the authors.

Finally, according to Wang's theory (Wang et al., 2021, Biochemistry 60, 3374-3384), the change of occupancy and the change of atomic motions B-factors can be completely decomposed in the omit maps. The change of occupancy does not alter the half-width at the half-maximum of omit peaks. The half-width at the half-maximum of omit peaks is only a function of B-factors. I hope that the authors will use normalized electron density peaks to substantiate their claim that W16 becomes disordered (i.e., increased B-factor) but it does not lose its occupancy. Otherwise, it appears to this reviewer that the authors arbitrarily exclude the possibility of W16 being one of substrate molecules entering the OEC during the 1F transition while hypothesizing that O6* is one during the 2F transition.

(Response to Reviewer #1)

We are happy that the reviewer has agreed with most of the revisions we have made. We agree with the reviewer that crystallographic data should be interpreted independent of other data such as theoretical calculations and spectroscopic data. As stated in the previous rebuttal letter and in the text, we do see a density corresponding to O6* in the omit map in the early time range, which was disappeared later with a concomitant appearance and increase of the O6 density. We believe that this is the evidence of O6 originating from O6*, instead of the movement of O5 only. On the other hand, we are aware of the different opinions regarding this interpretation, especially by Wang et al. To make this clear, we have added the following sentence and cited the paper by Wang et al. PNAS (2021) in the main text in the revised manuscript. We also added part of the rebuttal letter for the O5/O6 calculation in supplementary information based on the suggestion by reviewer #2.

This may leave some room for a hydroxyl/oxo coupling mechanism, and there is a crystallographic debate regarding the existence of O6/Ox in the S₃ structure³⁸ (Supplementary Fig. 1 and Supplementary Information).

Responses to Reviewer #2

Referee #2 (Remarks to the Author):

The authors have addressed my concerns. I especially like the modifications to Supplementary Table 1. The manuscript can be published without further change, although I recommend that the authors include their rebuttal of Wang et al. PNAS (2021) 118e202392118 as a section in their Supplementary Information so that a wide audience can see it. - Rick Debus

(Response to Reviewer #2)

We are delighted to see that the reviewer is happy with our revisions. According to the reviewer suggestion, we have included our rebuttal to Wang et al. PNAS (2021) in the Supplementary Information and Supplementary Fig. 1.

Response to Reviewer #3

Referee #3 (Remarks to the Author):

Reviewer comments on the revised manuscript of Li et al “Structural dynamics of oxygen-evolving photosystem II during 0F-1F-2F transitions”

The reviewer is very satisfied with the revised version of the manuscript of Li et al. The authors have very carefully revised the manuscript based on the comments of all 3 reviewers. While all 3 reviewers were very positive and impressed about the work Reviewer 1 raised several points about the interpretation of the electron density maps, and was most critical about the question of the appearance of O6, offering alternate interpretations. The authors took this critique very seriously and discussed both the alternate interpretation by this reviewer and their own model based on their experimental electron densities. They also now calculated polder maps and these clearly lead to the conclusion that the interpretation they presented in this paper is valid and they therefore did not change the conclusions drawn. This reviewer fully agrees to the conclusions of the authors and is on the strong opinion that the authors have addressed the concerns of reviewer 2 sufficiently.

On a side note the authors agreed to change the title of the paper from the S1-S2-S3 transition to 0F-1F-2F transition based on the comments of reviewer 1 also taking the questions of referee 2 into account who asked to include more details on the FTIR measurements that identified the %ages of S states after the 1F 2F and 3F laser excitation. This reviewer does not agree to this title change as the structures determined were from the S1, S2 and S3 state as the authors have carefully taken the S-state content for each data set into account. So, for example, after 1F 40% of the PSII molecules are in the S1 state and 60% in the S2 state. However, the authors have (based on the known S1 structure) taken the occupancy into account and therefore the electron density maps and difference density maps do NOT represent the 40:60 mixture of S1 and S2 states but represent the S2 state at time points t_x after the 1st flash. Therefore this reviewer strongly recommends to revert the title back to the original title. This will be also much more consistent to the naming in previous publications on TR-SFX studies on Photosystem II from this group and the group at Berkeley.

The major critique point of this reviewer was focused on the lack of information on the cell culture of *Thermosynechococcus vulcanus* and the PSII isolation, crystallization, the dehydration process as well as information on the embedding of the crystals in viscous media and sample delivery. The reviewer is very satisfied with the revised version of the manuscript, where the authors have rewritten the methods section on isolation and crystallization completely and now provide a comprehensive description of all steps involved in this important part of their study. The reviewer highly appreciates that they describe the process now in detail and also provide the corresponding references. There is only one detail that is missing: the authors state that they further dehydrated the sample after they embedded the PSII crystals in the grease viscous media before sample delivery. This is astonishing as one would assume that a 250ul sample (200ul grease plus 50ul crystals) dries out very fast but obviously the process is slower in grease. It would be important if the authors could provide the optimal incubation time for this additional dehydration step.

The reviewer had asked several minor points that included suggestions on making the figures more readable, which the authors have all considered. The reviewer appreciates all the changes

(like labelling the amino acids only in one of the panels, or the clear featuring of difference density changes which are now much more prominently marked and highlighted).

The reviewer had also criticized that the paper lacks a detailed conclusion section and suggested to move the summary and conclusion section included extended data figure 8 into the main manuscript (hoping for the journal to allow for the paper to exceed the figure and word limit).

The authors explained that this was not possible to the strict page limitations but they were able to include this important Figure now into the main text of the paper and also included a short conclusion paragraph with reference to the more detailed discussion in the extended data section. The reviewer is satisfied with this solution as it allows the general reader to see the main conclusions of the paper in the main text and also will be able to find the more detailed discussion easily in the extended data file.

One question that puzzled the reviewer was the negative difference density of the “invisible” water W53’. The authors are now explaining that this water is only visible in cryogenic dark state X-ray structures, but they assume that it is there at RT but is disordered and becomes even further disordered after light excitation. The reviewer is satisfied that this interpretation is now included in the main text, as it avoids confusion of the readers.

An interesting answer was given by the authors to the reviewers question about the comparison of the two data evaluation programs CrystFEL and cctbx.xfel . The authors explained that they did the initial data analysis including hitfinding with CrystFEL and then performed the further analysis both with CrystFEL and cctbx.xfel where they reached higher indexing rates and thereby 0.1 to 0.2 Å higher resolution with cctbx.xfel . They want, however, not draw conclusions and compare the data analysis site by site in this paper as major upgrades have been done in the meantime for both programs. The reviewer understands this reasoning and is satisfied with this answer.

The reviewer had also asked if the results of the DFT calculations could be included in the revised manuscript. The authors explained that this would be beyond the scope of the paper and the reviewer is fine with this answer and looks forward to see this included in a separate paper.

In summary, the authors have fully addressed the comments and suggestions of the reviewers.

As outlined in the original review, the reviewer finds this paper extremely important and is very enthusiastic about the exciting results presented in this paper. The reviewer is fully satisfied with the revisions and strongly recommends the publication of the paper in its revised form in Nature.

Response to Reviewer #3

We are very grateful for the detailed evaluation of our paper by this reviewer and for the strong recommendation of our paper for publication in Nature. Based on the reviewer's suggestions, we have reverted the title to the original one, although we have shortened it slightly to meet the word limitation of Nature. We also added the incubation time for the dehydration step in grease, which is between 30 to 60 minutes.